# EMFormer: Efficient Multi-Scale Transformer for Accumulative Context Weather Forecasting

**Hao Chen** [1]  **Tao Han** [1]  **Jie Zhang** [1]  **Song Guo** [1]  **Fenghua Ling** [2]  **Lei Bai** [2]

## Abstract

Long-term weather forecasting is critical for socioeconomic planning and disaster preparedness. While recent approaches employ finetuning to extend prediction horizons, they remain constrained by the issues of catastrophic forgetting, error accumulation, and high training overhead. To address these limitations, we present a novel pipeline across pretraining, finetuning and forecasting to enhance long-context modeling while reducing computational overhead. First, we introduce an Efficient Multi-scale Transformer (EMFormer) to extract multi-scale features through a single convolution in both training and inference. Based on the new architecture, we further employ an accumulative context finetuning to improve temporal consistency without degrading short-term accuracy. Additionally, we propose a composite loss that dynamically balances different terms via a sinusoidal weighting, thereby adaptively guiding the optimization trajectory throughout pretraining and finetuning. Experiments show that our approach achieves great performance in weather forecasting and extreme event prediction, substantially improving long-term forecast accuracy. Moreover, EMFormer demonstrates strong generalization on vision benchmarks (ImageNet-1K and ADE20K). Code: https://github.com/chenhao-zju/emformer

## 1. Introduction

**Why do we need data-driven methods in weather forecasting?** Long-term weather forecasting is a critical challenge with significant socioeconomic implications, affecting

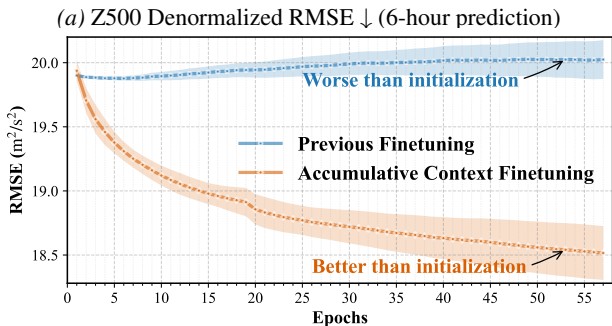

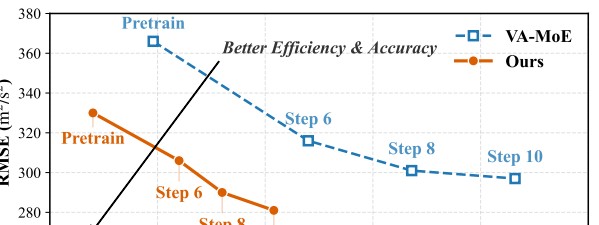

*Figure 1.* Denormalized Z500 RMSE $(m^2/s^2)$ for short-term (6-hour) and medium-term (5-day) forecasts. (a) Training convergence comparison in 6-step finetuning: accumulative context finetuning and previous finetuning. (b) Medium-term forecast performance: The proposed method consistently outperforms VA-MoE across pretraining and multi-step finetuning (6, 8, 10 steps). Two models are trained by us with A100.

sectors such as aviation, maritime navigation, and finance. Traditional Numerical Weather Prediction (NWP) generates forecasts by solving partial differential equations (Bauer et al., 2015; Lynch, 2008; Kalnay, 2002), but it suffers from cumulative errors and high computation. In contrast, data-driven models (Kurth et al., 2023; Bi et al., 2023; Chen et al., 2025b; Bodnar et al., 2025) learn atmospheric patterns directly from historical observations. By producing forecasts from learned representations rather than iterative physical integration, these approaches minimize error propagation and achieve greater computational efficiency.

**Why is an Efficient Architecture Essential for Finetuning?** While data-driven methods outperform NWP by capturing atmospheric dynamics, they remain prone to stepwise error accumulation during auto-regressive forecasting. To

[1]Department of Computer Science and Engineering, Hong Kong University of Science and Technology, Hong Kong, China [2]Shanghai AI Laboratory, Shanghai, China. Correspondence to: Song Guo <songguo@ust.hk>, Jie Zhang <csejzhang@ust.hk>.

*Proceedings of the 43$^{rd}$ International Conference on Machine Learning*, Seoul, South Korea. PMLR 306, 2026. Copyright 2026 by the author(s).

enhance long-term accuracy, many approaches employ extended finetuning on multi-step sequences. Although this improves forecast horizons, it introduces two critical limitations: (1) as inference length increases, the model gradually forgets information from earlier steps (blue line in Fig. 1a), and (2) longer finetuning sequences demand greater training time and computational resources (blue line in Fig. 1b). To address the first issue, we introduce accumulative context finetuning, which explicitly preserves historical information to ensure temporal consistency. Yet, this technique inevitably exacerbates the computational burden. Consequently, it is crucial to design an efficient framework that reduces computational cost during training and finetuning.

To address the efficiency and stability bottlenecks in long-context forecasting, we propose **a novel pipeline** comprising pretraining, finetuning, and forecasting. During pretraining, we introduce a hierarchical pruning-recovering framework coupled with an Efficient Multi-scale Transformer (EMFormer) to lower computational cost. For finetuning, an accumulative context mechanism is employed to strengthen long-horizon representations, ensuring temporal consistency across multi-step predictions. To further refine optimization, we design a variable- and geography-aware loss that adapts to the inherent heterogeneity of atmospheric variables across both physical properties and geographical regions.

We make three core contributions: (i) **Efficient Multi-Scale Architecture via Hard-aware Design.** While multi-scale transformers are effective, their training costs are often prohibitive. To address this, we propose EMFormer, which integrates a novel multi-convolution (multi-convs) layer optimized with custom CUDA kernels. Unlike standard re-parameterization methods that only accelerate inference, our method enables multi-scale feature capture via a single convolution during both training and inference. This redesign preserves representational power while accelerating forward and backward passes by $5.69\times$ compared to traditional multi-scale implementations. (ii) **Accumulative Finetuning for Long-context Consistency.** We introduce a specialized finetuning strategy tailored for long-context weather forecasting. By injecting historical Key–Value (KV) pairs into current generation steps and employing a memory-pruning mechanism, we explicitly bound memory usage while strengthening long-term temporal dependencies. This ensures sustained accuracy over extended horizons without compromising short-term performance. (iii) **Sinusoidal Weighted Optimization Objective.** We design a composite objective function featuring a sinusoidal weighting mechanism to address the heterogeneity of atmospheric data. This includes a latitude-adaptive term, which accounts for spatial distortion and a variable-adaptive term that balances learning dynamics across distinct physical variables, ensuring robust optimization across diverse atmospheric conditions.

In addition to theoretical analysis, experiments are presented in Fig. 1. Fig. 1a plots the denormalized Z500 RMSE ($m^2/s^2$) curves for 6-hour forecasts under two finetuning strategies. While conventional finetuning leads to progressively deteriorating first-step RMSE, the proposed accumulative finetuning steadily reduces the error, demonstrating superior stability in short-term forecasts. Fig. 1b compares the 5-day predictions between VA-MoE (Chen et al., 2025a) and Ours with distinct finetuning steps. Our approach achieves lower RMSE with shorter finetuning period: at Step 10, it reaches about 280 $m^2/s^2$ after 210 GPU-days, whereas VA-MoE requires 430 GPU-days to attain 295 $m^2/s^2$. Although both methods improve with extended finetuning, our method outperforms VA-MoE with fewer GPU-days. Beyond atmosphere, EMFormer also delivers competitive performance on vision tasks such as classification and segmentation, surpassing existing methods.

## 2. Methodology

This section presents a novel pipeline for weather forecasting, comprising: (1) single-step pretraining, (2) multi-step finetuning, and (3) multi-step forecasting. The forecasting task is defined and the framework is outlined in Sec. 2.1. The core contribution, EMFormer, which incorporates a multi-convolution (multi-convs) layer, is introduced in Sec. 2.2. Sec. 2.3 then describes the accumulative context finetuning with a memory module for cache management. Finally, Sec. 2.4 details a sinusoidal-weighted loss that combines variable-adaptive and latitude-weighted terms.

In addition, two propositions concerning the multi-convs layer (Theorem 2.1) and loss function (Theorem 2.2) are formulated. Proofs of the propositions and supporting experiments are provided in Sec. A and Sec. F.1, respectively.

### 2.1. Overview

As illustrated in Fig. 2, this pipeline addresses weather forecasting, in which a model $\Phi$ predicts future atmospheric states $\mathbf{X}^{t+1}$ from historical inputs $\mathbf{X}^t$, such that $\mathbf{X}^{t+1} = \Phi(\mathbf{X}^t)$. The input $\mathbf{X}^t$ comprises upper-air variables $\mathbf{P}^t \in \mathbb{R}^{H \times W \times 13 \times N}$ across 13 pressure levels and surface variables $\mathbf{S}^t \in \mathbb{R}^{H \times W \times M}$, where $N$ and $M$ denote the number of variables per level and surface, respectively.

In single-step pretraining, we employ an efficient framework to capture atmospheric patterns. The input variables $\mathbf{X}^t$ are first partitioned into patches and spatially pruned from the resolution $HW$ to $\frac{1}{16}HW$. The latent representation, $\mathbf{Z}^t \in \mathbb{R}^{C \times \frac{1}{16}HW}$, is then processed through a stack of EMFormers, where $C$ denotes dimension.

During multi-step accumulative-context finetuning, the model learns to perform iterative forecasting by leveraging a memory module that selectively prunes and propagates his-

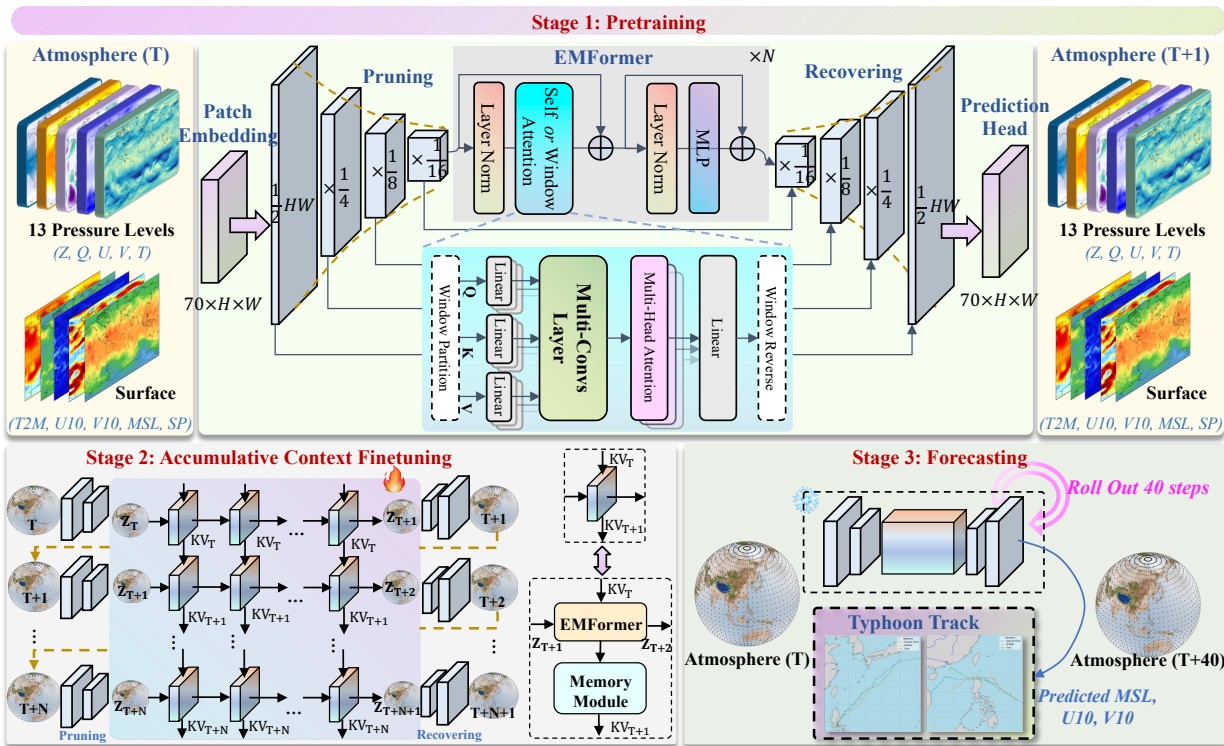

*Figure 2.* Illustration of the novel pipeline with three stages. Stage 1: EMFormer is pretrained on atmospheric variables with pruning-recovering architecture that includes a pruning module, a series of EMFormer blocks, and a recovering module; Stage 2: accumulative context finetuning; Stage 3: The forecasting stage with weather forecasting and typhoon track prediction.

torical KV states. In each step after initialization, the model conditions its prediction and the retained KV pairs from the preceding step, outputting both the prediction and an updated cache: $\mathbf{X}^{t+2}, \mathbf{KV}^{t+1} = \Phi(\mathbf{X}^{t+1}, \mathbf{KV}^t)$. In initial step, KV values are empty and thus no KV are injected.

The multi-step forecasting stage unifies two key subtasks: global deterministic forecasting $\Phi_g$ and typhoon track prediction $\Phi_{tc}$. For global forecasting, autoregressive generates iterative predictions, where each output serves as the input for the subsequent step: $\mathbf{X}^{t+N+1}, \mathbf{KV}^{t+N} = \Phi(\mathbf{X}^{t+N}, \mathbf{KV}^{t+N-1})$. The resulting forecasts provide atmospheric variables, such as mean sea-level pressure (MSL) and 10-meter wind components (U10, V10), which are subsequently utilized for typhoon track prediction.

The principal contributions are: (1) the **Efficient Multi-scale Transformer (EMFormer)**, and (2) the **Accumulative Context Finetuning** with a memory module, both of which are detailed in the subsections. In addition, we introduce a **novel loss** specifically designed for the atmosphere.

### 2.2. EMFormer

Building upon standard Transformer components, self- or window-based attention, MLPs, and residual connections, EMFormer introduces an efficient Multi-Convs Layer to capture multi-scale features from the latent space $\mathbf{Z}^t \in$

$\mathbb{R}^{3C \times \frac{1}{16}HW}$. Here, $3C$ corresponds to the query-key-value concatenation. Unlike conventional multi-scale modules that employ separate convolution layers with different kernel sizes (Fig. 3a), the proposed Multi-Convs Layer captures multi-scale information through a **single, fused convolution operation** (Fig. 3c). This design preserves the representational capacity of a multi-branch network while achieving significant acceleration. To validate equivalence, we examine both the forward and backward paths. The pseudo-code of CUDA kernel is provided in Algorithm 2.

In the forward pass, the equivalence of the fused operation to separate multi-scale convolutions follows the linearity of convolution. We consider three kernels $K_1, K_3, K_5$ of sizes $1 \times 1$, $3 \times 3$, and $5 \times 5$, all applied with stride 1. The latent embedding $\mathbf{Z}^t$ is reshaped from $\mathbb{R}^{3C \times \frac{1}{16}HW}$ to $\mathbb{R}^{3C \times H_0 \times W_0}$, where $H_0 = \frac{1}{4}H$ and $W_0 = \frac{1}{4}W$. The summed multi-scale result among three separate convolutions (Fig. 3a) can be obtained equivalently by a single convolution:

$$\mathbf{Z}'^t = \sum_{i=0}^{H_0-1} \sum_{j=0}^{W_0-1} \left( K_1 \oplus K_3 \oplus K_5 \right) \odot \mathbf{Z}^t[i, j, 5], \quad (1)$$

where $K_1 \oplus K_3 \oplus K_5$ denotes by aligning and adding $K_1, K_3, K_5$ at their centers (Fig. 3b,c), $\oplus$ denotes element-wise addition after zero-padding each kernel to the largest kernel ($5 \times 5$), and $\mathbf{Z}^t[i, j, 5]$ is the $5 \times 5$ region

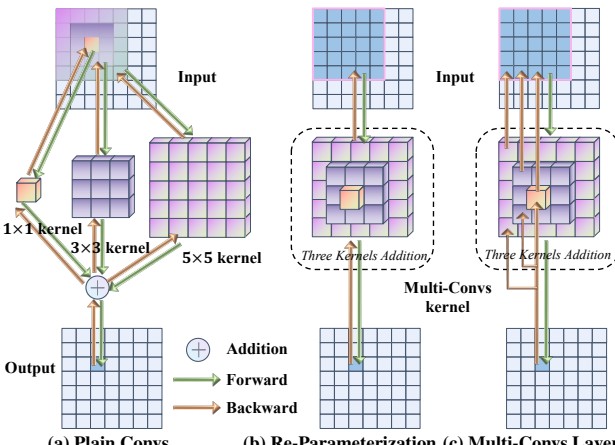

(a) Plain Convs    (b) Re-Parameterization (c) Multi-Convs Layer

*Figure 3.* Illustration of Multi-Convs Layer within EMFormer.

centered at $(i, j)$. This formulation preserves mathematical equivalence while reducing the forward pass to a single convolution, thereby lowering computation cost.

In the backward pass, gradients for the three distinct kernels are computed separately, allowing each kernel to learn scale-specific features while leveraging the fused forward operation. After the forward pass, the outputs of the architectures in Fig. 3 a, b, and c are identical, denoted $\mathbf{Z}'^t$. Unlike the standard implementation in Fig. 3 a, our multi-convs layer merges the three gradient-computation loops into one, updating the gradients of all three convolutions in a single parallelized loop. With $L$ and $\mathbf{Z}^t[i, j, r]$ denoting loss and $r \times r$ region centered at $(i, j)$, the process is formulated as:

$$\frac{\partial L}{\partial K_1}, \frac{\partial L}{\partial K_3}, \frac{\partial L}{\partial K_5} = \sum_{i=0}^{H_0-1} \sum_{j=0}^{W_0-1} \Big[ \frac{\partial L}{\partial \mathbf{Z}'^t[i,j]} \cdot \mathbf{Z}^t[i,j,1],$$

$$\frac{\partial L}{\partial \mathbf{Z}'^t[i,j]} \cdot \mathbf{Z}^t[i,j,3], \frac{\partial L}{\partial \mathbf{Z}'^t[i,j]} \cdot \mathbf{Z}^t[i,j,5] \Big], \quad (2)$$

The key difference between Fig.3 a/c in backward is thus the consolidation of the gradient updates from three sequential computations into a single, parallel computation.

The standard re-parameterization module (Fig. 3b) achieves computational efficiency during inference. However, applying this technique during training, the combined module mathematically degenerates into single convolution. Under standard differentiation, the gradients for the separate branches become coupled, making them functionally equivalent to single kernel, differing only in its initialization. To preserve the distinct optimization dynamics of each scale, we implement a custom CUDA kernel (Fig. 3c) that decouples the backward pass, maintaining independent gradient paths for each kernel and ensuring each branch retains its unique representation throughout training.

**Proposition 2.1** (Efficiency and Equivalence of Multi-Conv Layer). *Let $\mathcal{M}_{plain}$ denote a standard multi-scale module with kernels $K_{r \in \{1,3,5\}}$, and let $\mathcal{M}_{mc}$ denote the multi-convs*

*layer. Given identical input features $\mathbf{Z}^t$ and identical kernel initialization, the following properties hold:*

*1. Function equivalence: For spatial position $(i, j)$,*

$$\mathcal{M}_{plain}(\mathbf{Z}^t)[i, j] = \mathcal{M}_{mc}(\mathbf{Z}^t)[i, j]. \quad (3)$$

*2. Gradient equivalence: The gradients to each weight $K_r$ are identical:*

$$\left. \frac{\partial L}{\partial K_r} \right|_{\mathcal{M}_{plain}} = \left. \frac{\partial L}{\partial K_r} \right|_{\mathcal{M}_{mc}}, \quad \forall r \in \{1, 3, 5\} \quad (4)$$

*3. Computational efficiency: The multi-convs layer reduces the computation complexity from $\mathcal{O}(N_{kernels} \cdot H_0 \cdot W_0 \cdot r^2)$ to $\mathcal{O}(H_0 \cdot W_0 \cdot r_{\max}^2)$, where $N_{kernels} = 3$, $r_{\max} = 5$.*

**Why does EMFormer introduce Multi-Convs Layer in Transformer?** The Multi-Convs Layer is introduced to enable efficient capture of multi-scale patterns, which is essential for tasks where target structures vary widely in size. By extracting features across multiple receptive fields in a single forward pass, the layer provides a computationally efficient alternative to stacking separate convolutional branches. These multi-scale features allow the subsequent Multi-Head Attention (MHA) module to compute affinities not only between individual points, but also between regions of different spatial extents. In contrast to standard attention, EMFormer can therefore model relationships among spatially heterogeneous structures, enhancing representational flexibility while maintaining low computation.

### 2.3. Accumulative Context Finetuning

To mitigate error accumulation and enhance temporal consistency in multi-step forecasting, this work introduces accumulative finetuning within auto-regressive. As illustrated in Fig. 4, the KV pairs from previous steps of every block in EMFormer are concatenated and stored in cache. However, as the inference horizon extends, the accumulated cache may exceed memory constraints. To address this, we incorporate a memory module that dynamically prunes values while retaining information critical for maintaining consistency. The pseudo-code of KV pruning is in Algorithm 3.

The memory module updates scores and prunes the cache with three steps. **(1) First**, the query $\mathbf{Q} \in \mathbb{R}^{C \times L}$ is multiplied by the concatenated key $\mathbf{K} \in \mathbb{R}^{C \times NL}$, which includes both the current key and cached keys from previous steps. This produces an attention map $\mathbf{Attn} = (\mathbf{Q})^\top \mathbf{K}$, where $N = 5$ is the cache length and $L = \frac{1}{16} HW$ represents the spatial dimension. The resulting attention map $\mathbf{Attn} \in \mathbb{R}^{L \times NL}$ is then normalized along the key dimension using a softmax and mean function to generate the current scores, $\mathbf{S}_{cur} = \text{Mean}(\text{Softmax}(\mathbf{Attn})) \in \mathbb{R}^N$.

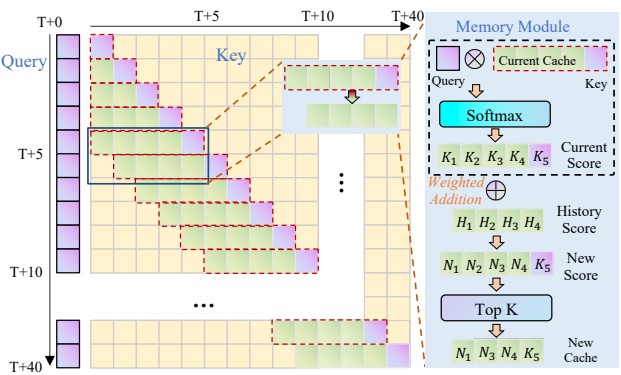

*Figure 4.* Illustration of accumulative context finetuning.

**(2) Next**, the scores are updated by blending the first $N - 1$ entries of $\mathbf{S}_{\text{cur}}$ with the historical scores $\mathbf{S}_{\text{his}}$. In the first step, since no historical scores exist, $\mathbf{S}_{\text{his}}$ is initialized as a zero vector. For subsequent steps, the combination is governed by a hyperparameter $\lambda = 0.9$, as follows: $\mathbf{S}_{\text{new}}[1 : N - 1] = \lambda \mathbf{S}_{\text{cur}}[1 : N - 1] + (1 - \lambda)\mathbf{S}_{\text{his}}$. The $N$-th token in $\mathbf{S}_{\text{cur}}$ is preserved without blending, i.e., $\mathbf{S}_{\text{new}}[N] = \mathbf{S}_{\text{cur}}[N]$.

**(3) Finally**, the cache is pruned by selecting the top $N - 2$ scores from $\mathbf{S}_{\text{new}}[1 : N - 1]$. These selected entries, together with the default $N$-th token, form the final $N - 1$ key-value pairs, passed to the next step as $\mathbf{S}_{\text{his}}$.

The accumulative context finetuning retains historical information from earlier steps, ensuring temporal consistency in long-term predictions. This method allows the model to preserve relevant features during long-term forecasting, reducing error accumulation by leveraging low-error historical states and preventing the degradation of initial forecasts. To mitigate the increasing computational demands of long-context forecasting, a memory module is employed to prune the KV pairs, retaining only the most critical information while discarding less relevant entries. This selective pruning balances efficiency and accuracy in long-term inference.

### 2.4. Loss Function

Given that some variables exhibit minimal variation over time, we introduce a variable-adaptive loss that optimizes different variables with distinct rates. In addition, we adopt a geography-adaptive loss for the latitudinal distribution. The final loss is designed to balance these two losses.

To adaptively adjust the importance of different variables during training, we adopt a dynamic weighting scheme:

$$\mathcal{L}_{\text{var}} = \text{Mean}\Big(\frac{(\hat{\mathbf{X}}^{t+1} - \mathbf{X}^{t+1})^2}{e^{\mathbf{w}}} + \mathbf{w}\Big), \quad (5)$$

where $\hat{\mathbf{X}}^{t+1}$ and $\mathbf{X}^{t+1}$ denote prediction and ground-truth states, $e$ is the natural exponential base, and $\mathbf{w} \in \mathbb{R}^{1 \times 1 \times C}$ is a learnable vector. To obtain a scalar loss value, 'Mean' denotes Latitude-Longitude Mean function. This formulation

enables the model to optimize each variable with distinct learning rates via the scaling factor $e^{\mathbf{w}}$, while the additive penalty $\mathbf{w}$ prevents excessive suppression of any variable.

Apart from dynamic loss, we introduce a latitude-weighted loss that scales prediction errors according to the area of each grid cell, a practice consistent with standard evaluation metrics in meteorology. The loss is defined as:

$$\mathcal{L}_{\text{lat}} = \text{Mean}\big(\mathbf{L} \odot (\hat{\mathbf{X}}^{t+1} - \mathbf{X}^{t+1})^2\big), \quad (6)$$

where $\mathbf{L} \in \mathbb{R}^{H \times 1}$ is a weight vector whose elements $L_i$ are repeated across the longitude, and $\odot$ denotes element-wise multiplication. The weight $L_i$ for latitude row $i$ is given by:

$$L_i = N_{\text{lat}} \times \frac{\cos \phi_i}{\sum_{j=1}^{N_{\text{lat}}} \cos \phi_j}, \quad (7)$$

where $\phi_i$ and $\phi_j$ represent the latitudes of $i$ and $j$, and $N_{\text{lat}}$ is the total number of latitudes. This formulation ensures that errors are scaled in proportion to the area represented by each grid cell, aligning with evaluation metric.

To balance the latitude-weighted loss $\mathcal{L}_{\text{lat}}$ and the variable-weighted loss $\mathcal{L}_{\text{var}}$, this work introduces a sinusoidal weighting scheme that smoothly interpolates between them during training. The total loss is defined as the convex combination:

$$\mathcal{L} = \frac{1}{2}\big(1 - \sin(\theta)\big)\mathcal{L}_{\text{lat}} + \frac{1}{2}\big(1 + \sin(\theta)\big)\mathcal{L}_{\text{var}}, \quad (8)$$

where $\theta$ is a learnable parameter. The coefficients $\frac{1}{2}(1 \mp \sin \theta)$ lie in $[0, 1]$ and sum to 1, enabling the optimization to shift adaptively from latitude-corrected toward variable-specific as training progresses.

**Proposition 2.2** (Adaptive Loss Weighting). *Consider the loss function in Equation* (8)*. With learning rate $\eta > 0$, $\theta_0 = -\pi/2$, and $\mathbf{w}_0 = \mathbf{0}$, the following properties hold:*

*1. The parameter $\theta$ evolves monotonically from $-\pi/2$ to $\pi/2$ during training;*

*2. The loss function $\mathcal{L}$ automatically transitions its emphasis from the latitude-weighted term $\mathcal{L}_{lat}$ to the variable-aware term $\mathcal{L}_{var}$ as training progresses.*

**Why do we need sinusoidal weighting?** Atmospheric variables evolve at different rates (*e.g.*, geopotential shows minimal variation over 6-hour intervals, whereas temperature exhibits pronounced changes), making it essential to adjust their learning speeds separately, especially during the low-learning-rate stage. We therefore employ a variable-adaptive loss to assign effective per-variable learning rates, alongside a latitude-weighted loss to respect geographical scaling. Simply combining these losses degrades performance, as it conflates spatial and physical adjustments prematurely. Instead, we introduce a sinusoidal weighting mechanism

*Table 1.* Performance of Ours with 4 baselines on 1.4°ERA5. Small RMSE (normalized, ↓) and bigger ACC (denormalized, ↑) indicate better. The best and second-best results are in **bold** and underline. All competitors are collected from Oneforecast (Gao et al., 2025).

| Model | Metric | | | | | | | | | |
|---|---|---|---|---|---|---|---|---|---|---|
| | 6-hour | | 1-day | | 4-day | | 7-day | | 10-day | |
| | RMSE | ACC | RMSE | ACC | RMSE | ACC | RMSE | ACC | RMSE | ACC |
| Pangu-weather(Bi et al., 2023) | 0.0826 | 0.9876 | 0.1571 | 0.9581 | 0.3380 | 0.8167 | 0.5092 | 0.5738 | 0.6215 | 0.3542 |
| Graphcast(Lam et al., 2023) | 0.0626 | 0.9928 | 0.1304 | 0.9705 | 0.2861 | 0.8705 | 0.4597 | 0.6692 | 0.6009 | 0.4275 |
| Fuxi(Chen et al., 2023b) | 0.0987 | 0.9820 | 0.1708 | 0.9511 | 0.4128 | 0.7379 | 0.5972 | 0.4446 | 0.6981 | 0.2391 |
| Oneforecast(Gao et al., 2025) | **0.0549** | **0.9943** | 0.1231 | 0.9737 | 0.2732 | 0.8825 | 0.4468 | 0.6888 | 0.5918 | 0.4457 |
| Ours (w/o finetuning) | 0.0626 | 0.9931 | **0.1219** | **0.9749** | **0.2673** | **0.8845** | **0.4327** | **0.6978** | **0.5719** | **0.4614** |
| Ours (w/ finetuning) | 0.0599 | 0.9949 | 0.1139 | 0.9775 | 0.2539 | 0.8936 | 0.4072 | 0.7223 | 0.5094 | 0.5389 |

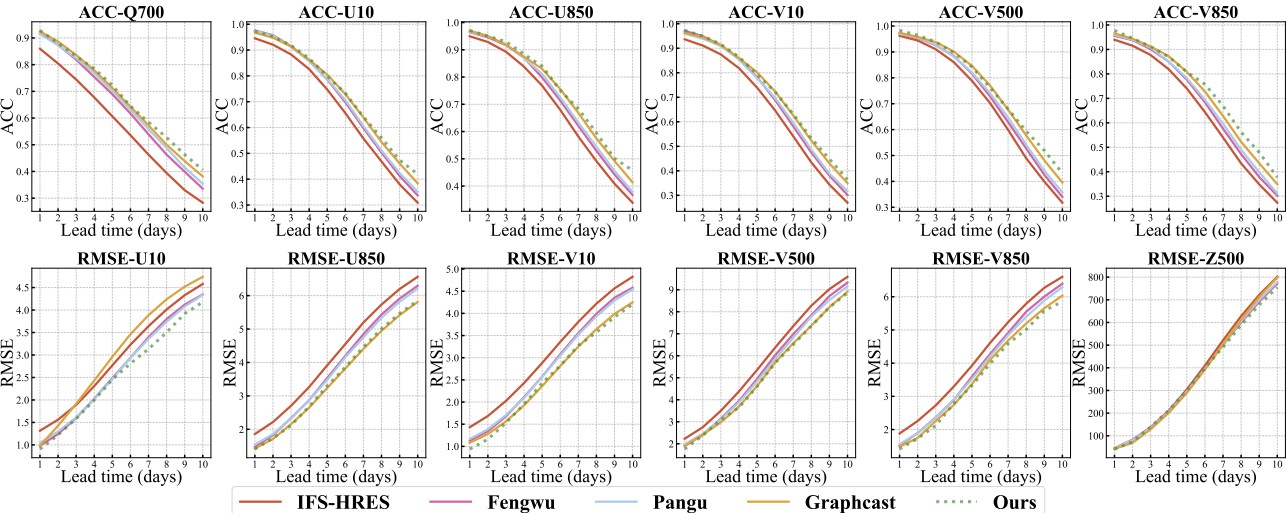

*Figure 5.* Comparison of our method with 4 competitors on denormalized RMSE ↓ and ACC ↑ in 0.25°ERA5.

that smoothly transitions the loss emphasis: initially prioritizing latitude-corrected error for stable coarse learning, and later shifting focus to variable-specific refinement for fine-grained optimization. This curriculum aligns with the multi-stage nature of training and the inherent heterogeneity of atmospheric data, yielding superior forecasts.

## 3. Experiments

### 3.1. Dataset and Implementation Details

We evaluate the proposed model with three benchmarks: ERA5 reanalysis dataset (Hersbach et al., 2020) for atmospheric forecasting, alongside the ImageNet-1K (Deng et al., 2009) and ADE20K (Zhou et al., 2017) benchmarks for classification and segmentation, respectively.

All experiments are conducted on 16 NVIDIA Tesla A100 GPUs. The codes will be released.

**Statement 1.** Related Works (Sec. B), Dataset and Implementation Details (Sec. D), Evaluation Metric (Sec. E), Computation Comparisons(Sec. F.2), Additional Results

(Sec. F.3), **Additional Ablation Study** (Sec. F.4), and Visualization (Sec. F.5) are provided in Appendix.

**Statement 2.** More competitors, ClimaX (Nguyen et al., 2023), EWMoE (Gan et al., 2025), Keisler (Keisler, 2022), FourCastNet (Kurth et al., 2023), ClimODE (Verma et al., 2024), WeatherGFT (Xu et al., 2024), FGN (Alet et al., 2025), and GenCast (Price et al., 2025), are in Sec. F.3.

### 3.2. Main Results

**Weather forecasting.** Model performance is evaluated with Root Mean Square Error (RMSE) and Anomaly Correlation Coefficient (ACC). Owing to the substantial differences in physical scales among variables, we report mean ACC and mean normalized RMSE in Tab. 1, where experiments are conducted on a downsampled 1.4°grid. Our method consistently outperforms baseline models, with especially pronounced gains in longer-term forecasts. Additional validation on the 0.25°grid with ACC and denormalized RMSE for 1–10-day predictions (Fig. 5) further confirms better results across multiple atmospheric variables. These im-

*Table 2.* Comparative performance of Ours with 8 baselines on ten typhoon track forecasts in 2024. Best results are highlighted in **bold**. All competitors are collected from CMA (`tcdata.typhoon.org.cn`).

| Model | Average Error (km) ↓ | | | | | | | | | | |
|---|---|---|---|---|---|---|---|---|---|---|---|
| | AMPIL | BEBINCA | Ewiniar | GAEMI | KONGREY | KRATHON | MANYI | SHANSHAN | YAGI | Yinxing | Average |
| AIFS | 55.0 | 213.4 | 106.0 | 98.5 | 79.6 | 187.1 | 138.7 | 161.2 | 107.3 | **44.9** | 119.17 |
| AVNO | 66.4 | 160.3 | 95.5 | 147.0 | - | 190.9 | 256.5 | 281.0 | 119.2 | 128.9 | - |
| ECMF | 87.0 | 211.5 | 51.1 | 123.7 | 78.1 | 132.4 | 263.9 | 179.7 | 99.5 | 167.5 | 139.44 |
| Fengwu | 99.3 | 214.4 | 95.3 | 119.6 | 80.0 | 195.3 | 147.7 | **90.8** | 122.5 | 87.4 | 125.23 |
| Fengqing | 122.9 | 167.6 | - | 87.9 | 62.0 | **87.5** | 195.6 | 190.0 | 103.5 | 141.1 | - |
| Fourcastnet | 163.8 | 155.6 | 111.3 | 154.9 | 114.2 | 315.4 | 252.1 | 313.0 | 285.3 | 315.5 | 218.11 |
| Pangu | 78.3 | 233.5 | **39.8** | 101.4 | 54.9 | 148.4 | 130.0 | 188.7 | 114.7 | 149.7 | 123.94 |
| Graphcast | 122.4 | 171.5 | 40.2 | 134.2 | 91.1 | 167.1 | 136.2 | 182.9 | 106.9 | 55.3 | 120.78 |
| Fuxi | **49.8** | 145.0 | 76.7 | 97.6 | 57.9 | 141.5 | 136.0 | - | 125.2 | 93.4 | - |
| Ours | 66.7 | **138.2** | 49.8 | **84.5** | **44.9** | 130.3 | **115.3** | 120.3 | **75.6** | 59.3 | **88.49** |

*Table 3.* Ablation study on finetuning strategies. 'w/o' and 'w/' denote 'without' and 'with'.

| Finetuning | 6-hour | | 1-day | | 4-day | | 7-day | | 10-day | | Params | GPU | Latency |
|---|---|---|---|---|---|---|---|---|---|---|---|---|---|
| Strategies | RMSE | ACC | RMSE | ACC | RMSE | ACC | RMSE | ACC | RMSE | ACC | (M) | (G) | Avg/(ms) |
| **w/o Accumulative Finetuning** | | | | | | | | | | | | | |
| w/o Finetuning | 0.0626 | 0.9931 | 0.1219 | 0.9749 | 0.2673 | 0.8845 | 0.4327 | 0.6978 | 0.5719 | 0.4614 | 157.9 | 0.703 | 98.3 |
| w/ Finetuning | 0.0645 | 0.9917 | 0.1189 | 0.9756 | 0.2491 | 0.8869 | 0.4261 | 0.7098 | 0.5339 | 0.4958 | 157.9 | 0.726 | 99.1 |
| w/ Lora Finetuning | 0.0626 | 0.9931 | 0.1173 | 0.9759 | 0.2473 | 0.8936 | 0.4224 | 0.6981 | 0.5278 | 0.5041 | 171.3 | 4.919 | 111.3 |
| **w/ Accumulative Finetuning (KV Cache Length=5)** | | | | | | | | | | | | | |
| w/o Pruning | 0.0599 | 0.9950 | 0.1152 | 0.9766 | 0.2466 | 0.8852 | 0.4109 | 0.7191 | 0.5153 | 0.5316 | 157.9 | 4.576 | 118.5 |
| w/ Pruning | 0.0599 | 0.9949 | 0.1139 | 0.9775 | 0.2439 | 0.8936 | 0.4072 | 0.7223 | 0.5094 | 0.5389 | 157.9 | 4.913 | 120.5 |

*Table 4.* Ablation study on the backbone's blocks. All results are gained with 1.4°ERA5.

| Block of Backbone | 6-hour | | 1-day | | 4-day | | 7-day | | 10-day | |
|---|---|---|---|---|---|---|---|---|---|---|
| | RMSE | ACC | RMSE | ACC | RMSE | ACC | RMSE | ACC | RMSE | ACC |
| Self-Attention Block | 0.0711 | 0.9912 | 0.1287 | 0.9664 | 0.2696 | 0.8711 | 0.4603 | 0.6015 | 0.5978 | 0.4499 |
| MambaVision Block | 0.0711 | 0.9912 | 0.1308 | 0.9655 | 0.2829 | 0.8619 | 0.4595 | 0.6695 | 0.5978 | 0.4365 |
| Self/Windows Attn Block | 0.0709 | 0.9915 | 0.1298 | 0.9665 | 0.2786 | 0.8679 | 0.4566 | 0.6806 | 0.6016 | 0.4486 |
| EMFormer | 0.0626 | 0.9931 | 0.1219 | 0.9749 | 0.2673 | 0.8845 | 0.4327 | 0.6978 | 0.5719 | 0.4614 |

provements stem from two core contributions: (1) the efficient multi-scale transformer, which captures patterns across varied receptive fields, and (2) the accumulative context finetuning, which enhances temporal consistency and mitigates catastrophic forgetting in extended forecasts.

**Typhoon track prediction.** To assess performance under extreme conditions, we evaluate 10 typhoons from the 2024 (Ying et al., 2014; Lu et al., 2021). Tab. 2 compares the mean 96-hour track errors (aggregated over lead times 6, 12, ..., 96 hours) against 9 baselines: AIFS (Lang et al., 2024), AVNO, ECMWF (Molteni et al., 1996), FengWu (Chen et al., 2025b), Fengqing, FourCastNet (Kurth et al., 2023), Pangu (Bi et al., 2023), Graphcast (Lam et al., 2023), and FuXi (Chen et al., 2023b). Our method obtains the lowest error on 5 typhoons and remains competitive on the remaining five. The overall mean error across all typhoons is 88.49 km, the best among all competitors and notably lower than the next-best result (119.17 km). These results, together with the consistent ability to capture cyclone evolution, con-

firm its potential for extreme event forecasting. The error of individual lead times are in Fig. 10.

**Image classification.** Beyond weather forecasting, we also evaluate EMFormer on the ImageNet-1K and compare it with recent SoTA methods. As summarized in Tab. 5, models are grouped into three parameter scales: ˜30M (tiny), ˜50M (small), and ˜90M (base). Our EMFormer achieves the highest accuracy in all three categories, 83.2%, 84.1%, and 84.4% for the tiny, small, and base models, respectively. Notably, at the base level, our model surpasses MambaOut (Yu & Wang, 2025), MambaVision (Hatamizadeh et al., 2025), and VRWKV (Duan et al., 2025) while using fewer parameters and FLOPs. These results indicate that EMFormer is not only effective for weather forecasting, but also generalizes well to vision tasks, owing to its ability to capture multi-scale patterns through varied receptive fields and to accelerate computation via the fused multi-convs layer.

**Semantic Segmentation.** To further assess the generalization of our EMFormer, we evaluate its performance on

*Table 5.* Comparison of classification benchmarks on **ImageNet-1K** dataset (Deng et al., 2009). Throughput (TP) is measured on A100 with batch size of 128. All are tested with $224 \times 224$ size.

| Model | #Params (M) | FLOPs (G) | TP (Img/Sec) | Top-1 (%) |
|---|---|---|---|---|
| **Tiny Model** | | | | |
| ConvNeXt-T (Liu et al., 2022) | 28.6 | 4.5 | 3196 | 82.0 |
| ResNetV2-50 (Wightman et al., 2021) | 25.5 | 4.1 | 6402 | 80.4 |
| Swin-T (Liu et al., 2021) | 28.3 | 4.4 | 2758 | 81.3 |
| TNT-S (Han et al., 2021) | 23.8 | 4.8 | 1478 | 81.5 |
| Twins-S (Chu et al., 2021) | 24.1 | 2.8 | 3596 | 81.7 |
| DeiT-S (Touvron et al., 2021) | 22.1 | 4.2 | 4608 | 79.9 |
| PoolFormer-S36 (Yu et al., 2022) | 30.9 | 5.0 | 1656 | 81.4 |
| CrossViT-S (Chen et al., 2021) | 26.9 | 5.1 | 2832 | 81.0 |
| NextViT-S (Li et al., 2022a) | 31.7 | 5.8 | 3834 | 82.5 |
| EfficientFormer-L3 (Li et al., 2022b) | 31.4 | 3.9 | 2845 | 82.4 |
| VMamba-T (Liu et al., 2024) | 30.0 | 4.9 | 1282 | 82.6 |
| EfficientVMamba-B (Pei et al., 2025) | 33.0 | 4.0 | 1482 | 81.8 |
| VRWKV-S (Duan et al., 2025) | 23.8 | 4.6 | - | 80.1 |
| MambaVision-T (Hatamizadeh et al., 2025) | 31.8 | 4.4 | 6298 | 82.3 |
| MambaOut-Tiny (Yu & Wang, 2025) | 27.0 | 4.0 | - | 82.7 |
| EMFormer-T | 28.7 | 5.1 | 3378 | **83.2** |
| **Small Model** | | | | |
| ConvNeXt-S (Liu et al., 2022) | 50.2 | 8.7 | 2008 | 83.1 |
| ResNetV2-101 (Wightman et al., 2021) | 44.5 | 7.8 | 4019 | 82.0 |
| Swin-S (Liu et al., 2021) | 49.6 | 8.5 | 1720 | 83.2 |
| Twins-B (Chu et al., 2021) | 56.1 | 8.3 | 1926 | 83.1 |
| PoolFormer-M36 (Yu et al., 2022) | 56.2 | 8.8 | 1170 | 82.1 |
| NextViT-L (Li et al., 2022a) | 57.8 | 10.8 | 2360 | 83.6 |
| FasterViT-1 (Hatamizadeh et al., 2024) | 53.4 | 5.3 | 4188 | 83.2 |
| VMamba-S (Liu et al., 2024) | 50.0 | 8.7 | 843 | 83.6 |
| MambaVision-S (Hatamizadeh et al., 2025) | 50.1 | 7.5 | 4700 | 83.3 |
| MambaOut-Small (Yu & Wang, 2025) | 48.0 | 9.0 | - | **84.1** |
| EMFormer-S | 45.5 | 7.4 | 2512 | **84.1** |
| **Base Model** | | | | |
| ConvNeXt-B (Liu et al., 2022) | 88.6 | 15.4 | 1485 | 83.8 |
| Twins-L (Chu et al., 2021) | 99.3 | 14.8 | 1439 | 83.7 |
| CrossViT-B (Chen et al., 2021) | 105.0 | 20.1 | 1321 | 82.2 |
| EfficientFormer-L7 (Li et al., 2022b) | 82.2 | 10.2 | 1359 | 83.4 |
| VMamba-B (Liu et al., 2024) | 89.0 | 15.4 | 645 | 83.9 |
| DeiT-B (Touvron et al., 2021) | 86.6 | 16.9 | 2035 | 82.0 |
| DeiT3-B (Touvron et al., 2022) | 86.6 | 16.9 | 670 | 83.8 |
| VRWKV-B (Duan et al., 2025) | 93.7 | 18.2 | - | 82.0 |
| MambaVision-B (Hatamizadeh et al., 2025) | 97.7 | 15.0 | 3670 | 84.2 |
| MambaOut-Base (Yu & Wang, 2025) | 85.0 | 15.8 | - | 84.2 |
| EMFormer-B | 80.6 | 12.3 | 1693 | **84.4** |

the ADE20K (see Tab. 6). Our small and base models achieve mIoU scores of 46.7 and 49.6, respectively, while requiring approximately 75% of the parameters and 25% of the FLOPs compared to other methods. These results demonstrate that the proposed framework not only delivers competitive accuracy on dense prediction tasks, but also maintains significantly lower computational overhead.

### 3.3. Ablation Study

**Multi-Convs layer.** To further validate the efficacy of the multi-convs layer, we conduct comparative experiments on ERA5 (Tab. 7) and ImageNet-1K (Tab. 8). Results show that models equipped with either the plain multi-scale module or the multi-convs layer achieve nearly identical performance on both tasks. However, the multi-convs layer yields substantially lower computational cost, reducing the required training time by approximately 25% for weather forecasting and 20% for image classification.

**Finetuning strategy.** Tab. 3 presents an ablation study on

*Table 6.* Semantic segmentation with **ADE20K** (Zhou et al., 2017). All models are trained using a crop resolution of $512 \times 512$.

| Backbone | Param (M) | FLOPs (G) | mIoU |
|---|---|---|---|
| DeiT-Small/16 (Touvron et al., 2021) | 52 | 1099 | 44.0 |
| Swin-T (Liu et al., 2021) | 60 | 945 | 44.5 |
| ResNet-101 (He et al., 2016) | 86 | 1029 | 44.9 |
| Focal-T (Yang et al., 2021) | 62 | 998 | 45.8 |
| VMamba-T (Liu et al., 2024) | 62 | 949 | 48.0 |
| EfficientVMamba-B (Pei et al., 2025) | 65 | 930 | 46.5 |
| MambaVision-T (Hatamizadeh et al., 2025) | 55 | 945 | 46.0 |
| EMFormer-S | 48 | 238 | 46.7 |
| Swin-S (Liu et al., 2021) | 81 | 1038 | 47.6 |
| Twins-SVT-B (Chu et al., 2021) | 89 | - | 47.7 |
| Focal-S (Yang et al., 2021) | 85 | 1130 | 48.0 |
| VMamba-S (Liu et al., 2024) | 76 | 1028 | 50.6 |
| MambaVision-S (Hatamizadeh et al., 2025) | 84 | 1135 | 48.2 |
| EMFormer-B | 69 | 251 | 49.6 |

*Table 7.* Comparison of plain multi-scale module and multi-convs layer on our method with weather forecasting without finetuning.

| Model | 6-hour prediction (RMSE ↓) | | | | Params | Time |
|---|---|---|---|---|---|---|
| | Z500 | T2M | T850 | U10 | (M) | (Hours) |
| Multi-scale | 18.6 | 0.531 | 0.421 | 0.525 | 157.9 | 83 |
| Multi-convs | 18.6 | 0.533 | 0.422 | 0.523 | 157.9 | 60 |

*Table 8.* Comparison of plain multi-scale module and multi-convs layer on our base method with image classification task.

| Model | Top-1 (%) | Top-5 (%) | Params (M) | Times (Hours) |
|---|---|---|---|---|
| EMFormer-B (Multi-scale) | 84.3 | 97.1 | 80.6 | 123 |
| EMFormer-B (multi-convs) | 84.4 | 96.9 | 80.6 | 107 |

three finetuning strategies: no finetuning, standard finetuning, and accumulative-context finetuning. Results show that both finetuning variants outperform the non-finetuned baseline in long-term predictions. For short-term forecasts, accumulative finetuning achieves better accuracy than the baseline, whereas standard one performs slightly worse. These findings indicate that accumulative finetuning maintains temporal consistency across forecast horizons, delivering robust performance in both short- and long-term forecasts.

**Block in backbone.** Tab. 4 presents an ablation study comparing different blocks: self-attention, Mamba, self/window attention, and EMFormer. While all variants yield only minor variations in both short- and long-term forecasts, EMFormer achieved a further reduction of 0.0083 for 6-hour and 0.0297 for 10-day predictions in RMSE. These results indicate that EMFormer delivers superior performance on forecasting, and its integrated multi-convs layer is effective at capturing atmospheric patterns across different scales.

**Ablation studies** about Loss function (Tab. 14), $\lambda$ in finetuning (Tab. 18), Cache length (Tab. 19), Finetuning steps (Tab. 20), and Window size (Tab. 21) are all in Appendix.

# 4. Conclusion

In this paper, we present a novel pipeline for weather forecasting. This work first employs an efficient framework with a multi-convs layer to capture multi-scale atmospheric patterns during training. It then adopts accumulative context finetuning to improve long-context consistency while maintaining short-term forecast quality. A sine-balanced loss is introduced to adaptively combine variable- and latitude-weighted objectives during optimization. The design is supported by theoretical analysis that validates both the efficiency of the multi-conv layer and the convergence behavior of the loss. Experiments show that the proposed method achieves great results in atmospheric field and delivers competitive performance on vision benchmarks, demonstrating its broad applicability and effectiveness.

# Acknowledgements

This research was supported by fundings from the Hong Kong RGC General Research Fund (152228/23E, 162161/24E, 162116/25E, 162180/25E), National Natural Science Foundation of China (NSFC) Key Program (No.62532005), Collaborative Research Fund (No. C1042-23GF, No. C5097-25G), NSFC/RGC Collaborative Research Scheme (Grant No. 62461160332 & CRS_HKUST602/24), Research Impact Fund (No. R5011-23F), Areas of Excellence Scheme (AoE/E-601/22-R), and the InnoHK (HKGAI).

# Impact Statement

This paper presents work whose goal is to advance the field of Machine Learning. There are many potential societal consequences of our work, none which we feel must be specifically highlighted here.

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

# Contents

## A. Proof of Proposition

### A.1. Theorem 2.1: Equivalence and Efficiency of the Multi-Conv Layer

**Proposition A.1** (Efficiency and Equivalence of Multi-Conv Layer). *Let $\mathcal{M}_{plain}$ denote a standard multi-scale convolutional module with kernels $\{K_r\}_{r \in \{1,3,5\}}$ of sizes $r \times r$, and let $\mathcal{M}_{multi\text{-}convs}$ denote the proposed multi-convolution layer with the same kernel weights. Given identical input features $\mathbf{Z}^t$ and identical kernel initialization, the following properties hold:*

1. **Functional equivalence**: *For all spatial positions $(i, j)$,*

$$\mathcal{M}_{plain}(\mathbf{Z}^t)[i, j] = \mathcal{M}_{multi\text{-}convs}(\mathbf{Z}^t)[i, j]. \tag{9}$$

2. **Gradient equivalence**: *The gradients with respect to each kernel weight $K_r$ are identical in both modules:*

$$\left.\frac{\partial L}{\partial K_r}\right|_{\mathcal{M}_{plain}} = \left.\frac{\partial L}{\partial K_r}\right|_{\mathcal{M}_{multi\text{-}convs}}, \quad \forall r \in \{1, 3, 5\}. \tag{10}$$

3. **Computational efficiency**: *The multi-conv layer reduces the computational complexity from $\mathcal{O}(N_{kernels} \cdot H_0 \cdot W_0 \cdot r^2)$ to $\mathcal{O}(H_0 \cdot W_0 \cdot r_{\max}^2)$ for both forward and backward passes, where $N_{kernels} = 3$, $r_{\max} = 5$, and $H_0 \times W_0$ is the spatial dimension of the feature map.*

*Proof.* We present a formal derivation to demonstrate the equivalence between the forward and backward computations of the proposed multi-convs layer and those of separate multi-scale convolutions, taking as an example the fusion of three convolution layers $K_1, K_3, K_5$ with kernel sizes 1×1, 3×3, and 5×5.

**Preliminaries and Notation** For clarity, the derivation focuses on the spatial dimensions (height $H_0$ and width $W_0$); the conclusions extend directly to the full tensor case including batch and channel dimensions. Let $\mathbf{Z}^t[i, j, r]$ denote the $r \times r$ input region centered at spatial location $(i, j)$. The convolution operation is defined as:

$$\mathbf{Z}'^t = \sum_{i=0}^{H_0-1} \sum_{j=0}^{W_0-1} K_r \odot \mathbf{Z}^t[i, j, r], \tag{11}$$

where $\odot$ represents element-wise multiplication followed by summation over the kernel support. The gradient of a loss function $L$ with respect to a kernel weight $K_r$ is:

$$\frac{\partial L}{\partial K_r} = \sum_{i=0}^{H_0-1} \sum_{j=0}^{W_0-1} \frac{\partial L}{\partial \mathbf{Z}'^t[i, j]} \odot \mathbf{Z}^t[i, j, r]. \tag{12}$$

**Proof of 1: Functional Equivalence (Forward Pass)** The output of a standard multi-scale module $\mathcal{M}_{plain}$ is the sum of three separate convolutions:

$$\mathbf{Z}'^t = \sum_{i=0}^{H_0-1} \sum_{j=0}^{W_0-1} \left( K_1 \odot \mathbf{Z}^t[i, j, 1] + K_3 \odot \mathbf{Z}^t[i, j, 3] + K_5 \odot \mathbf{Z}^t[i, j, 5] \right). \tag{13}$$

A key observation is that the supports of the three kernels are nested: $K_1$ occupies only the center point, $K_3$ a $3 \times 3$ region, and $K_5$ a $5 \times 5$ region. Therefore, the summation in (13) can be rewritten by aligning and adding the kernels at their common center point (denoted by operator $\oplus$, with zero-padding for alignment):

$$\mathbf{Z}'^t = \sum_{i=0}^{H_0-1} \sum_{j=0}^{W_0-1} \left( K_1 \oplus K_3 \oplus K_5 \right) \odot \mathbf{Z}^t[i, j, 5]. \tag{14}$$

Equation (14) is precisely the forward pass of the multi-convs layer $\mathcal{M}_{multi\text{-}convs}$, which performs a single convolution with the composite kernel $(K_1 \oplus K_3 \oplus K_5)$. Since (13) and (14) are mathematically equivalent, the outputs of the two modules are identical, proving property (1).

**Proof of 2: Gradient Equivalence (Backward Pass)** In $\mathcal{M}_{\text{plain}}$, gradients for the three kernels are computed independently:

$$\frac{\partial L}{\partial K_1} = \sum_{i=0}^{H_0-1} \sum_{j=0}^{W_0-1} \frac{\partial L}{\partial \mathbf{Z}'^t[i,j]} \cdot \mathbf{Z}^t[i,j,1], \tag{15}$$

$$\frac{\partial L}{\partial K_3} = \sum_{i=0}^{H_0-1} \sum_{j=0}^{W_0-1} \frac{\partial L}{\partial \mathbf{Z}'^t[i,j]} \cdot \mathbf{Z}^t[i,j,3], \tag{16}$$

$$\frac{\partial L}{\partial K_5} = \sum_{i=0}^{H_0-1} \sum_{j=0}^{W_0-1} \frac{\partial L}{\partial \mathbf{Z}'^t[i,j]} \cdot \mathbf{Z}^t[i,j,5]. \tag{17}$$

Since $\mathcal{M}_{\text{multi-convs}}$ produces the same output $\mathbf{Z}'^t$ from the same input $\mathbf{Z}^t$ and kernels $\{K_r\}$, and because gradient computation for each kernel depends only on $\partial L/\partial \mathbf{Z}'^t$ and the corresponding input region $\mathbf{Z}^t[i,j,r]$, the gradient formulas remain unchanged. The multi-convs layer computes these gradients in a fused, parallel loop:

$$\frac{\partial L}{\partial K_1}, \frac{\partial L}{\partial K_3}, \frac{\partial L}{\partial K_5} = \sum_{i=0}^{H_0-1} \sum_{j=0}^{W_0-1} \Big[ \frac{\partial L}{\partial \mathbf{Z}'^t[i,j]} \cdot \mathbf{Z}^t[i,j,1],$$

$$\frac{\partial L}{\partial \mathbf{Z}'^t[i,j]} \cdot \mathbf{Z}^t[i,j,3],$$

$$\frac{\partial L}{\partial \mathbf{Z}'^t[i,j]} \cdot \mathbf{Z}^t[i,j,5] \Big], \tag{18}$$

which is computationally consolidated but mathematically identical to evaluating (15)–(17) separately. This proves property (2).

**Proof of 3: Computational Efficiency** We analyze the computational complexity for a standard convolution operation. Let $H_0 \times W_0$ denote the spatial resolution, and let $C_{\text{in}}$ and $C_{\text{out}}$ denote the number of input and output channels, respectively. We define $\mathcal{C}_{\text{mult-add}}$ as the cost of a multiply-accumulate operation and $\mathcal{C}_{\text{mem}}$ as the cost of a memory access.

In the standard multi-scale module $\mathcal{M}_{\text{plain}}$, $N$ separate convolutions (with kernel sizes $r \in \mathcal{R}$) are executed sequentially. The total arithmetic cost involves summing the operations for each kernel:

$$\mathcal{C}_{\text{arith}}^{(\text{plain})} = \mathcal{C}_{\text{mult-add}} \cdot H_0 W_0 C_{\text{in}} C_{\text{out}} \sum_{r \in \mathcal{R}} r^2. \tag{19}$$

Assuming a standard implementation where each convolution layer triggers separate kernel launches, the memory access cost (accounting for input reads, weight reads, and output writes) is:

$$\mathcal{C}_{\text{mem}}^{(\text{plain})} = \mathcal{C}_{\text{mem}} \cdot \sum_{r \in \mathcal{R}} \big( H_0 W_0 C_{\text{in}} + r^2 C_{\text{in}} C_{\text{out}} + H_0 W_0 C_{\text{out}} \big). \tag{20}$$

In the proposed $\mathcal{M}_{\text{multi-convs}}$, the kernels are fused into a single composite kernel with size $r_{\max} = \max(\mathcal{R})$. The forward pass effectively performs a single convolution:

$$\mathcal{C}_{\text{arith}}^{(\text{fused})} = \mathcal{C}_{\text{mult-add}} \cdot H_0 W_0 C_{\text{in}} C_{\text{out}} \cdot r_{\max}^2. \tag{21}$$

Crucially, the fused operation significantly reduces memory traffic by reading the input feature map and writing the output feature map only once, rather than $N$ times. The memory cost becomes:

$$\mathcal{C}_{\text{mem}}^{(\text{fused})} = \mathcal{C}_{\text{mem}} \cdot \big( H_0 W_0 C_{\text{in}} + r_{\max}^2 C_{\text{in}} C_{\text{out}} + H_0 W_0 C_{\text{out}} \big). \tag{22}$$

Comparing the two approaches, the multi-convs layer achieves efficiency gains in two aspects: 1. **Arithmetic Reduction:** Since $\sum_{r \in \mathcal{R}} r^2 > r_{\max}^2$ (for non-trivial sets $\mathcal{R}$ containing multiple kernels), the total FLOPs are reduced. 2. **Memory Bandwidth Optimization:** The dominant term in memory cost for large feature maps, $H_0 W_0 (C_{\text{in}} + C_{\text{out}})$, is reduced by a factor of $N$ (the number of branches), as the fused kernel eliminates redundant I/O operations associated with intermediate results in the multi-branch structure.

### A.2. Theorem 2.2 : Loss Function

**Proposition A.2** (Adaptive Loss Weighting). *Consider the composite loss function*

$$\mathcal{L}(\theta, \mathbf{w}) = \frac{1}{2}\big(1 - \sin(\theta)\big)\big(\mathbf{L} \odot \mathcal{E}\big) + \frac{1}{2}\big(1 + \sin(\theta)\big)\big(\mathcal{E}e^{-\mathbf{w}} + \mathbf{w}\big), \tag{23}$$

*where $\mathcal{E} = (\hat{\mathbf{X}}^{t+1} - \mathbf{X}^{t+1})^2$ denotes the squared prediction error, $\mathbf{L}$ is a fixed latitude-weight matrix with non-negative entries, $\mathbf{w} \in \mathbb{R}^{1 \times 1 \times C}$ represents uncertainty parameters, and $\theta \in [-\pi/2, \pi/2]$ is a learnable balancing parameter. Under gradient descent optimization with learning rate $\eta > 0$, and with initial conditions $\theta_0 = -\pi/2 + \epsilon$ (where $0 < \epsilon \ll 1$) and $\mathbf{w}_0 = \mathbf{0}$, the following properties hold:*

1. *The parameter $\theta$ evolves monotonically from $-\pi/2$ to $\pi/2$ during training;*

2. *The loss function $\mathcal{L}$ automatically transitions its emphasis from the latitude-weighted term $\mathbf{L} \odot \mathcal{E}$ to the variable-aware term $\mathcal{E}e^{-\mathbf{w}} + \mathbf{w}$ as training progresses.*

*Proof.* We prove the proposition by analyzing the coupled dynamics of $\mathbf{w}$ and $\theta$ under gradient descent. For convenience, we rewrite the loss as a convex combination of two terms:

$$\mathcal{L} = \underbrace{\frac{1}{2}\big(1 - \sin(\theta)\big)}_{\alpha(\theta)} \underbrace{\big(\mathbf{L} \odot \mathcal{E}\big)}_{A} + \underbrace{\frac{1}{2}\big(1 + \sin(\theta)\big)}_{\beta(\theta)} \underbrace{\big(\mathcal{E}e^{-\mathbf{w}} + \mathbf{w}\big)}_{B}, \tag{24}$$

with $\alpha(\theta) + \beta(\theta) = 1$. Note that $A$ is the latitude-weighted error, and $B$ is the variable-aware loss with uncertainty parameter $\mathbf{w}$.

**Step 1: Optimal $\mathbf{w}$ for a fixed error $\mathcal{E}$**  We first analyze the behavior of term $B$ with respect to $\mathbf{w}$. The gradient of $\mathcal{L}$ w.r.t. $\mathbf{w}$ is

$$\frac{\partial \mathcal{L}}{\partial \mathbf{w}} = \beta(\theta) \cdot \frac{\partial B}{\partial \mathbf{w}} = \frac{1}{2}(1 + \sin\theta)\big(-\mathcal{E}e^{-\mathbf{w}} + 1\big). \tag{25}$$

Setting the gradient to zero gives the optimal $\mathbf{w}^*$ for a given $\mathcal{E}$:

$$-\mathcal{E}e^{-\mathbf{w}^*} + 1 = 0 \quad \Rightarrow \quad e^{-\mathbf{w}^*} = \frac{1}{\mathcal{E}} \quad \Rightarrow \quad \mathbf{w}^* = \ln(\mathcal{E}). \tag{26}$$

Substituting $\mathbf{w}^*$ back into $B$ yields its minimized value:

$$B^* = \mathcal{E} \cdot \frac{1}{\mathcal{E}} + \ln(\mathcal{E}) = 1 + \ln(\mathcal{E}). \tag{27}$$

Thus, as training progresses and the error $\mathcal{E}$ decreases, $B^*$ decreases and can become negative when $\mathcal{E}$ is small, whereas $A$ remains non-negative.

**Step 2: Dynamics of $\theta$**  The gradient of $\mathcal{L}$ with respect to $\theta$ is

$$\frac{\partial \mathcal{L}}{\partial \theta} = -\frac{1}{2}\cos(\theta)A + \frac{1}{2}\cos(\theta)B = \frac{1}{2}\cos(\theta)(B - A). \tag{28}$$

Under gradient descent with learning rate $\eta$, the update for $\theta$ is

$$\theta_{k+1} = \theta_k - \eta\frac{\partial \mathcal{L}}{\partial \theta} = \theta_k + \frac{\eta}{2}\cos(\theta_k)(A - B). \tag{29}$$

**Step 3: Monotonic increase of $\theta$**  To determine the sign of the gradient, we compare the magnitudes of $A$ and $B$. Here, we formally invoke an adiabatic approximation regarding the optimization dynamics.

*Assumption A.1 (Adiabatic Dynamics).* We assume a time-scale separation where the uncertainty parameters $\mathbf{w}$ converge to their local optima $\mathbf{w}^*$ significantly faster than the evolution of the balancing parameter $\theta$. This implies that for the analysis of $\theta$, we can approximate the instantaneous value $B$ with its optimized lower bound $B^*$.

Under this assumption, we compare the asymptotic scaling of $A$ and $B^*$:

The term $A$ represents the latitude-weighted error. Since the weights $\mathbf{L}$ are normalized ($\langle \mathbf{L} \rangle = 1$) and bounded, $A$ scales *linearly* with the error magnitude:

$$A = \mathbf{L} \odot \mathcal{E} \approx \mathcal{O}(\langle \mathcal{E} \rangle). \tag{30}$$

In contrast, as the model converges and the error $\langle \mathcal{E} \rangle$ becomes small (specifically, $\langle \mathcal{E} \rangle \ll 1$), the logarithmic term in Equation (27) dominates:

$$B^* \approx 1 + \ln(\langle \mathcal{E} \rangle) \to -\infty. \tag{31}$$

Consequently, in the regime of low error (typical after early training stages), we strictly have $B^* < 0 < A$, which implies $B - A < 0$, (see Fig. 6).

Substituting $A - B > 0$ into the update rule (29), and recalling that after the initial perturbation we have $\cos(\theta_k) > 0$, we obtain

$$\theta_{k+1} = \theta_k + \underbrace{\frac{\eta}{2} \cos(\theta_k)}_{>0} \underbrace{(A - B)}_{>0}. \tag{32}$$

Therefore, $\partial \mathcal{L}/\partial \theta < 0$ for all subsequent steps, causing $\theta$ to increase monotonically from its initial perturbed value ($> -\pi/2$) toward $\pi/2$.

**Step 4: Convergence and automatic loss transition** As $\theta$ increases, $\sin \theta \to 1$, which implies

$$\alpha(\theta) = \frac{1}{2}(1 - \sin \theta) \to 0, \qquad \beta(\theta) = \frac{1}{2}(1 + \sin \theta) \to 1. \tag{33}$$

Consequently, the weighting coefficient $\alpha(\theta)$ on the latitude-weighted term $A$ diminishes to zero, while $\beta(\theta)$ on the variable-aware term $B$ approaches one. This dynamic implements a soft curriculum schedule: the loss initially emphasizes the stable latitude-weighted error for coarse learning and automatically shifts focus to the uncertainty-aware term for fine-grained refinement as training progresses.

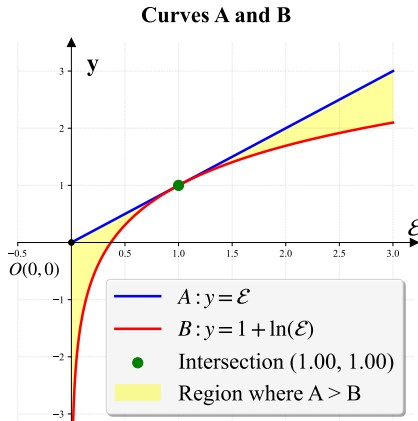

*Figure 6.* Asymptotic behavior of the terms $A$ (linear) and $B^*$ (logarithmic) as the mean error $\langle \mathcal{E} \rangle$ decreases. The intersection point where $A = B^*$ marks the threshold beyond which $\theta$ increases monotonically.

**Statement.** The experimental verification of the theoretical results in Sec. A.1 (Theorem 2.1) and Sec. A.2 (Theorem 2.2) is provided in Sec. F.1. The runtime comparison between the plain multi-scale module and the proposed multi-convs layer is detailed in Tab. 13. Furthermore, the evolution of the balancing parameter $\theta$ during training is visualized in Fig. 7.

# B. Related work

## B.1. Data-Driven Weather Forecasting

Before the advent of data-driven methods, numerical weather prediction (NWP) was the predominant approach, producing forecasts by solving partial differential equations on high-resolution global grids (Bauer et al., 2015; Lynch, 2008; Kalnay, 2002). While these methods deliver physically consistent and rigorously validated forecasts (Molteni et al., 1996; Ritchie et al., 1995; Karlbauer et al., 2024), they remain computationally prohibitive, especially at fine spatial scales.

The availability of large-scale atmospheric reanalysis datasets has since catalyzed a shift toward data-driven weather forecasting. Early exemplars such as FourCastNet (Kurth et al., 2023) and Pangu-Weather (Bi et al., 2023) employed Fourier Neural Operators (FNO) (Li et al., 2021) and 3D Swin Transformers (Liu et al., 2021), respectively, to approximate atmospheric evolution. Subsequent research has largely bifurcated into two streams: (i) neural operators, including KNO (Xiong et al., 2023), NeuralGCM (Kochkov et al., 2024), SFNO (Bonev et al., 2023), ClimODE (Verma et al.,

2024), FANO (Lin et al., 2026), SMgNO (Hu et al., 2025), and TNO (Diab & Al Kobaisi, 2025), that directly learn the temporal evolution operator; and (ii) neural network, such as FengWu (Chen et al., 2025b; Han et al., 2024; 2026; 2025), GraphCast (Lam et al., 2023), FuXi (Chen et al., 2023b), GenCast (Price et al., 2025), Stormer (Nguyen et al., 2025), ClimaX (Nguyen et al., 2023), OneForecast (Gao et al., 2025), and VAMoE (Chen et al., 2025a; Wang et al., 2025), which incorporate inductive biases tailored to atmospheric dynamics. Both categories achieve competitive forecast skill with substantially lower computational cost.

Notably, Aurora (Bodnar et al., 2025) addresses error accumulation by adopting a Low-Rank Adaptation (LoRA) finetuning strategy that assigns distinct parameters to different forecast time steps. While this approach can mitigate error drift, it introduces additional overhead from loading and saving multiple model states. In contrast, our method employs accumulative context finetuning, which shares the same parameters across all time steps and reduces error accumulation by retaining key-value pairs from previous steps, thereby avoiding unnecessary computational complexity.

## B.2. Long Context Learning

The advent of large language models has spurred extensive research into long-context learning, aiming to improve contextual coherence and accelerate inference over extended sequences (Tikochinski et al., 2025). Existing methods can be broadly divided into two categories: (1) those that use dynamic sparse attention masks to skip redundant computations (Pagliardini et al., 2023; Qu et al., 2022; Roy et al., 2021; Wang et al., 2021), and (2) those that prune the KV pairs adaptively based on input-sequence characteristics (Anagnostidis et al., 2023; Sheng et al., 2023; Ge et al., 2024). Both directions have successfully extended the effective context length while reducing computational overhead.

Inspired by these advances, recent work has adapted long-context techniques to video generation, leading to three main paradigms: training-free KV-cache compression (Yi et al., 2026; Kim et al., 2025), efficient frameworks via sparse attention matrices (Zhang et al., 2025a;b), and dual-model architectures that decouple memory from generation (Zhu et al., 2025; Shang et al., 2025). These approaches have improved both the speed and temporal consistency of long-form video synthesis.

Moving beyond language and video domains, this work introduces memory-augmented accumulative-context learning to weather forecasting. By coupling autoregressive prediction with accumulative-context modeling, the system retains relevant information from past time steps and leverages it to condition current forecasts. This design preserves long-range temporal consistency and mitigates error accumulation, thereby extending the skillful forecast horizon.

## B.3. Basic Vision Models

Since the rise of deep learning, fundamental vision tasks have been a central and driving domain in artificial intelligence (AI). Landmark contributions, such as the ImageNet dataset (Deng et al., 2009) and the AlexNet architecture (Krizhevsky et al., 2012), have profoundly shaped the field's evolution. Subsequent research has explored a wide variety of visual architectures, which can be broadly categorized into four paradigms: convolutional (Conv) networks (Wightman et al., 2021; Liu et al., 2022), vision transformers (ViTs) (Chen et al., 2021; Liu et al., 2021; Chu et al., 2021; Touvron et al., 2022; Cai et al., 2023), hybrid Conv-Transformer designs (Chen et al., 2021; Hatamizadeh et al., 2024; Li et al., 2022a; Chen et al., 2024; 2023a), and more recent Mamba-based models (Liu et al., 2024; Yu & Wang, 2025; Pei et al., 2025; Hatamizadeh et al., 2025). Each of these lines of work has achieved strong performance on core vision benchmarks and has collectively advanced deep learning at different stages of its development.

Following the hybrid Conv-Transformer paradigm, our model likewise prioritizes multi-scale representation and computational efficiency. Unlike previous works that directly employ plain multi-scale modules, we propose a novel layer capable of capturing multi-scale information through a single convolutional operation. To realize this capability, we redesign both the forward pass and the backward propagation of the standard convolution layer. Compared with conventional multi-scale modules, our proposed multi-convs layer accelerates the forward and backward computations by a factor of 5.69×.

*Comparison with structural re-parameterization methods.* Prior work on structural re-parameterization has focused on merging multi-branch modules into a single operation during inference (Ding et al., 2021). Although later extensions claim to support training (Huang et al., 2022), a fundamental limitation remains: the re-parameterized module effectively reduces to a single operation during backpropagation, losing its original multi-scale representation. As a result, such re-parameterized versions are not functionally equivalent to true multi-branch designs. In contrast, our multi-convs layer implements a CUDA-level multi-branch gradient-update mechanism that maintains independent gradient flows for each kernel throughout training. This ensures exact equivalence, in both forward and backward passes, between the conventional

---

**Algorithm 1** The Proposed Model in Weather Forecasting

---

**Input:** Atmospheric variables $\mathbf{X}^t$ at timestep $t$
**Output:** Forecasted atmospheric variables $\mathbf{X}^{t+1}$ at timestep $t+1$
 1: Apply high-stride convolution for patch embedding: $\mathbf{X}^t = \text{Conv}_{p \times p}(\mathbf{X}^t)$
 2: Add positional embedding: $\mathbf{X}^t = \mathbf{X}^t + \mathbf{P}$
 3:
 4: **Pruning**
 5: **for** each $k \in range(4)$ **do**
 6:     $\mathbf{X}^t_{\text{down}} = \mathbf{X}^t_{\text{res}} = \text{DownSample}(\mathbf{X}^t)$
 7:     $\mathbf{X}^t = \text{Cross-Attention}(\mathbf{X}^t_{\text{down}}, \mathbf{X}^t)$
 8:     $\mathbf{X}^t = \text{Self-Attention}(\mathbf{X}^t)$
 9: **end for**
10:
11: **Processor**
12: **for** each $i \in$ [blocks' number] **do**
13:     **if** $i\%2 == 0$ **then**
14:         Windows Size = (4, 4) *or* (2, 8) *or* (8, 2)
15:         Attention = Windows-EMFormer(Windows Size)
16:     **else**
17:         Attention = EMFormer()
18:     **end if**
19:     Apply multi-head attention with residual connection: $\mathbf{A}^t = \text{LN}(\text{Attention}(\mathbf{X}^t)) + \mathbf{X}^t$
20:     Apply MLP with residual connection: $\mathbf{X}^{t+1} = \text{MLP}(\mathbf{A}^t) + \mathbf{A}^t$
21: **end for**
22:
23: **Recovering**
24: **for** each $k \in range(4)$ **do**
25:     $\mathbf{X}^{t+1}_{up} = \text{UpSample}(\mathbf{X}^{t+1}) + \mathbf{X}^t_{\text{res}}$
26:     $\mathbf{X}^{t+1} = \text{Cross-Attention}(\mathbf{X}^{t+1}_{up}, \mathbf{X}^{t+1})$
27:     $\mathbf{X}^{t+1} = \text{Self-Attention}(\mathbf{X}^{t+1})$
28: **end for**
29:
30: Reconstruct atmospheric variables to longitude-latitude grids with prediction head: $\mathbf{X}^{t+1} = \text{Linear}(\mathbf{X}^{t+1})$

---

multi-scale module and our proposed layer, while preserving the representational benefits of multi-scale feature extraction.

## C. Detailed Structure

As shown in Algorithm 1, the global weather forecasting framework comprises three core components: Pruning, Processor, and Recovering, and two supplementary settings: patch embedding and prediction head. In this work, the Pruning, EMFormer, and Recovering implementations follow Flash-Attention (Dao et al., 2022). For the forecasting task, the AI model $\Phi$ predicts future atmospheric states $\mathbf{X}^{t+1}$ from historical fields $\mathbf{X}^t$ as $\mathbf{X}^{t+1} = \Phi(\mathbf{X}^t)$. Detailed configurations for all three elements are provided below.

### C.1. Pruning

At the encoding stage, atmospheric variables across pressure levels are structured as a 3D tensor $\mathbf{X}^t \in \mathbb{R}^{H \times W \times N}$, where $H$ and $W$ denote the spatial dimensions of the global grid, and $N$ is the number of variables. These variables are first embedded into a 2D patch representation $\mathbf{X}^t \in \mathbb{R}^{HW \times C}$, with $C$ being the embedding dimension.

To reduce computational cost, a series of downsampling operations are applied, each integrating cross-attention and self-attention. First, the input feature is reshaped to $\mathbf{X}^t \in \mathbb{R}^{\frac{1}{2}HW \times 2C}$ and then projected through a linear layer parameterized by a weight matrix $\mathbf{W} \in \mathbb{R}^{2C \times C}$, compressing the channel dimension to produce a downsampled feature $\mathbf{X}^t_{\text{down}} \in \mathbb{R}^{\frac{1}{2}HW \times C}$.

---

**Algorithm 2** Multi-Convs Layer

---

$\mathbf{input} \in \mathbb{R}^{B \times C_{in} \times H \times W}, \mathbf{weight1} \in \mathbb{R}^{C_{out} \times C_{in} \times 1 \times 1}, \mathbf{weight3} \in \mathbb{R}^{C_{out} \times C_{in} \times 3 \times 3}, \mathbf{weight5} \in \mathbb{R}^{C_{out} \times C_{in} \times 5 \times 5}$

1: **ForwardKernel**
2: Each thread computes one output element $(n, c_{out}, h, w)$
3: Initialize result $\leftarrow 0$
4: **for** $c_{in} = 0$ to $C_{in} - 1$ **do**
5:     **for** each spatial offset $(kh, kw)$ **do**
6:         Compute combined weight = **weight1** + **weight3** + **weight5**
7:         result $\leftarrow$ result + input $\times$ combined_weight
8:     **end for**
9: **end for**
10:
11: **WeightGradKernel**
12: Each thread processes one output gradient element
13: Load gradient value $\delta$
14: **for** $c_{in} = 0$ to $C_{in} - 1$ **do**
15:     **for** each spatial offset $(kh, kw)$ **do**
16:         Compute contribution $g \leftarrow \delta \times$ input
17:         **Parallel accumulation:** Add $g$ to **grad_weight1**, **grad_weight3**, **grad_weight5** using atomic operations
18:     **end for**
19: **end for**
20:
21: **InputGradKernel**
22: Each thread computes one input gradient element $(n, c_{in}, h, w)$
23: Initialize grad $\leftarrow 0$
24: **for** $c_{out} = 0$ to $C_{out} - 1$ **do**
25:     **for** each spatial offset $(kh, kw)$ **do**
26:         Load gradient value $\delta$
27:         Compute combined weight from three scales
28:         grad $\leftarrow$ grad + $\delta \times$ combined_weight
29:     **end for**
30: **end for**

---

This linear transformation effectively halves the channel dimension while preserving essential spatial–spectral information. The downsampled feature is also retained as a residual signal $\mathbf{X}_{\text{res}}^t$, which is later added to the corresponding upsampled features in the recovering module via a residual connection.

To preserve information that may be lost during downsampling, a cross-attention module is introduced to fuse the downsampled feature with the original input:

$$\mathbf{X}^t = \text{Cross-Attention}(Q = \mathbf{X}_{\text{down}}^t, K = \mathbf{X}^t, V = \mathbf{X}^t). \tag{34}$$

Subsequently, multiple self-attention blocks are applied to the fused feature to strengthen its internal correlations:

$$\mathbf{X}^t = \text{Self-Attention}(\mathbf{X}^t). \tag{35}$$

### C.2. EMFormer

The core component of the architecture consists of a sequence of EMFormer blocks. Each block contains a multi-head attention module, an MLP (Multi-Layer Perceptron) module, layer normalization, and residual skip connections. The operations within one block are formulated as:

$$\mathbf{A}^t = \text{LN}(\text{EMFormer/Windows-EMFormer}(\mathbf{X}^t)) + \mathbf{X}^t, \tag{36}$$
$$\mathbf{X}^{t+1} = \text{LN}(\text{MLP}(\mathbf{A}^t)) + \mathbf{A}^t, \tag{37}$$

---

**Algorithm 3** KV Values Pruning in Accumulative Context Finetuning

---

1: The maximum number of KV pairs in cache: $N$
2: Project input to key-value pairs: $Q, K_{\text{new}}, V_{\text{new}} \leftarrow \text{LinearProj}(x)$
3: Length of K, V values: $L \leftarrow \text{len}(K_{\text{new}})$
4:
5: **if** $(K \neq \text{None})$ & $(V \neq \text{None})$ **then**
6: $\quad K_{\text{cache}}, V_{\text{cache}} \leftarrow K, V$
7: $\quad T \leftarrow \text{len}(K_{\text{cache}})$
8: $\quad$ **if** $T \geq N * L$ **then**
9: $\quad\quad$ Get historical importance: scores $\leftarrow$ importance_scores
10: $\quad\quad$ Most important tokens: $\text{Index}_{topk} \leftarrow \text{TopK}(\text{scores}[0 : T - L], (N - 2) * L)$
11: $\quad\quad$ Most recent tokens: $\text{Index}_{recent} \leftarrow [T - L : T]$
12: $\quad\quad$ Combine the important and recent tokens: $\text{Index}_{keep} \leftarrow \text{Unique}(\text{Index}_{topk} \cup \text{Index}_{recent})$
13: $\quad\quad$ Preserve order: $\text{Index}_{keep} \leftarrow \text{Sort}(\text{Index}_{keep})$
14: $\quad\quad K_{\text{cache}} \leftarrow K_{\text{cache}}[:, \text{Index}_{keep}]$
15: $\quad\quad V_{\text{cache}} \leftarrow V_{\text{cache}}[:, \text{Index}_{keep}]$
16: $\quad$ **end if**
17: $\quad K \leftarrow [K_{\text{cache}}, K_{\text{new}}]$
18: $\quad V \leftarrow [V_{\text{cache}}, V_{\text{new}}]$
19: **else**
20: $\quad K \leftarrow K_{\text{new}}$
21: $\quad V \leftarrow V_{\text{new}}$
22: **end if**
23:
24: output $\leftarrow \text{Attention}(Q, K, V)$
25: attn $\leftarrow \text{Softmax}(QK^\top / \sqrt{d})$
26: Average across heads and batch: imp $\leftarrow \text{Mean}(\text{attn})$
27: old $\leftarrow$ importance_scores
28: importance_scores $\leftarrow 0.9 \times \text{old} + 0.1 \times \text{imp}$
29: Output: $(K, V)$

---

where EMformer denotes the efficient multi-scale attention mechanism with the multi-convs layer, LN is layer normalization, and MLP refers to the feed-forward module. The pseudo-code of multi-convs layer is provided in Algorithm 2.

To accommodate the multi-scale nature of atmospheric structures, we adopt a hybrid attention mechanism that interleaves window-based attention with global self-attention. This design effectively balances local feature extraction with long-range dependency modeling, allowing the network to capture both fine-grained details and broader contextual relationships within atmospheric data.

### C.3. Recovering

The recovering module mirrors the pruning module in reverse order, employing a series of upsampling stages to restore the hidden representation to its original spatial size. At each step, the hidden feature $\mathbf{X}^{t+1} \in \mathbb{R}^{\frac{1}{2}HW \times C}$ is reshaped to $\mathbf{X}_{\text{up}}^{t+1} \in \mathbb{R}^{HW \times \frac{1}{2}C}$ and then passed through a linear projection to expand the channel dimension, yielding $\mathbf{X}_{\text{up}}^{t+1} \in \mathbb{R}^{HW \times C}$. To retain fine-grained details from earlier layers, a residual connection links the corresponding stages of the pruning and recovering branches:

$$\mathbf{X}_{\text{up}}^{t+1} = \text{UpSample}(\mathbf{X}^{t+1}) + \mathbf{X}_{\text{res}}^t. \tag{38}$$

As in the pruning stage, cross-attention and multiple self-attention blocks are applied to refine the upsampled feature:

$$\mathbf{X}^{t+1} = \text{Cross-Attention}(Q = \mathbf{X}_{\text{up}}^{t+1}, K = \mathbf{X}^{t+1}, V = \mathbf{X}^{t+1}), \tag{39}$$

$$\mathbf{X}^{t+1} = \text{Self-Attention}(\mathbf{X}^{t+1}). \tag{40}$$

Finally, a prediction head, implemented as a linear layer, projects the latent representation back to the original variable space

*Table 9.* A summary of atmospheric variables. The 13 levels are 50, 100, 150, 200, 250, 300, 400, 500, 600, 700, 850, 925, 1000 hPa. 'Single' denotes the variables under earth's surface.

| Name | Description | Levels | Time |
|---|---|---|---|
| Z | Geopotential | 13 | 1979-2020 |
| Q | Specific humidity | 13 | 1979-2020 |
| U | x-direction wind | 13 | 1979-2020 |
| V | y-direction wind | 13 | 1979-2020 |
| T | Temperature | 13 | 1979-2020 |
| t2m | Temperature at 2m height | Single | 1979-2020 |
| u10 | x-direction wind at 10m height | Single | 1979-2020 |
| v10 | y-direction wind at 10m height | Single | 1979-2020 |
| msl | Mean sea-level pressure | Single | 1979-2020 |
| sp | Surface pressure | Single | 1979-2020 |

*Table 10.* Implementation details on weather forecasting.

| Category | Parameter | Value | Parameter | Value |
|---|---|---|---|---|
| Model Architecture | Input Size | $128 \times 256$ / $721 \times 1440$ | Input Channels | 70 |
| | Output Channels | 70 | Number of Blocks | 24 |
| | Dimension | 768 | Patch Size | 2/4 |
| Training Configuration | Optimizer | AdamW | LR Scheduler | CosineAnnealingLR |
| | Initial LR | $2 \times 10^{-4}$ | Minimum LR | $1 \times 10^{-7}$ |
| | Maximum Epochs | 100 | Loss Function | Hybrid Loss |
| Finetuning Configuration | Optimizer | AdamW | LR Scheduler | CosineAnnealingLR |
| | Initial LR | $5 \times 10^{-5}$ | Minimum LR | $1 \times 10^{-7}$ |
| | Maximum Epochs | 50 | Loss Function | Hybrid Loss |
| Data Settings | Input Steps | 1 | Training Output Steps | 1 |
| | Testing Output Steps | 40 | Training Period | [1979, 2020] |
| | Test Period | [2021, 2021] | Grid Resolution | 1.4°/0.25° |
| Experimental Setup | Batch Size | 1 | Global Batch Size | 16 |
| | #Data Workers | 20 | Mixed Precision | Enabled |
| | World Size | 16 | | |

to produce the forecast output.

## D. Dataset Details and Implementation Details

### D.1. Dataset Details

We evaluate the proposed model using three benchmark datasets spanning meteorological and vision tasks. For atmospheric field, this work employs the ERA5 global reanalysis dataset (Hersbach et al., 2020). For vision benchmarks, we adopt ImageNet-1K (Deng et al., 2009) for classification and ADE20K (Zhou et al., 2017) for segmentation, which allow us to assess the architecture's generalization beyond atmospheric modeling.

**ERA5.** In this work, experiments are conducted using the widely employed ERA5 reanalysis dataset[1] (Hersbach et al., 2020), produced by the European Centre for Medium-Range Weather Forecasts (ECMWF). As detailed in Tab. 9, ERA5 provides a comprehensive atmospheric record from 1979 onward at a spatial resolution of 0.25°(global grid dimensions of $721 \times 1440$). The selected variables comprise 5 upper-air quantities, geopotential (Z), specific humidity (Q), zonal wind (U), meridional wind (V), and temperature (T), across 13 pressure levels, together with 5 surface variables: 2-m temperature (T2M), 10-m zonal wind (U10), 10-m meridional wind (V10), mean sea-level pressure (MSL), and surface pressure (SP). This results in a total of 70 input variables ($5 \times 13 + 5$). The model is trained on data spanning 1979–2020 and evaluated on the held-out year 2021, both at the native 0.25°resolution.

**Imagenet-1K** (Deng et al., 2009) consists of approximately 1.28 million high-resolution training images and 50,000

---

[1]https://cds.climate.copernicus.eu/

*Table 11.* Implementation details for ImageNet-1K dataset.

| Configuration | Value | Configuration | Value |
|---|---|---|---|
| **Data Configuration** | | | |
| Dataset | ImageNet-1K | Batch size (per GPU) | 128 |
| Number of workers | 8 | Image size schedule | 128, 160, 192, 224 |
| Training interpolation | Random | Validation interpolation | Bicubic |
| Validation crop ratio | 1.0 | | |
| **Data Augmentation** | | | |
| RandAugment | $N = 2, M = 5$ | Random erase | $p = 0.2$ |
| Mixup | $\alpha = 0.2$, probability=1.0 | CutMix | $\alpha = 0.2$, probability=1.0 |
| Label smoothing | 0.1 | BCE loss | Enabled |
| **Training Schedule** | | | |
| Total epochs | 300 | Base learning rate | 1.5e-4 |
| Warmup epochs | 20 | Warmup learning rate | 0.0 |
| Learning rate schedule | Cosine decay | Optimizer | AdamW |
| Optimizer parameters | $\beta_1 = 0.9, \beta_2 = 0.999, \epsilon = 1e-8$ | Weight decay | 0.1 |
| Weight decay exclusion | Norm layers, bias terms | Gradient clipping | 2.0 |
| EMA decay | 0.9998 | | |
| **Model Configuration** | | | |
| Input resolution | 224 | Drop path rate | 0.1 (linear decay) |
| Stochastic depth | Enabled | Dropout rate | 0.0 |
| **Additional Techniques** | | | |
| BatchNorm reset | Enabled | Reset batch size | 16,000 samples |
| MESA threshold | 0.25 | Reset batch size per iteration | 100 |
| MESA ratio | 2.0 | | |

validation images, manually annotated into 1,000 distinct object categories.

**ADE20K** (Zhou et al., 2017) is a challenging task for semantic segmentation, which contains 150 categories, including 25574 natural images for training and 2000 images for validation.

### D.2. Data Processing

To account for differences in scale and distribution across variables, all model inputs are standardized. Using the training data from 1979–2019, we compute the mean and standard deviation for each variable. Each input is then normalized by subtracting its mean and dividing by its standard deviation.

### D.3. Implementation Details

#### Weather forecasting.

Our architecture builds upon the Flash Attention backbone (Dao et al., 2022). For weather forecasting, the entire model is trained end-to-end. Detailed training hyperparameters are provided in Tab. 10.

#### Image classification.

Our implementation adopts the FlashAttention architecture as its core backbone (Dao et al., 2022). For the ImageNet-1K image-classification task, the full model is trained in an end-to-end manner. Comprehensive training hyperparameters (e.g., learning rates, schedules, regularization) are listed in Tab. 11. The configurations of the three model variants, EMFormer-T (Tiny), EMFormer-S (Small), and EMFormer-B (Base), are detailed in Tab. 12.

## E. Evaluation Metric

### E.1. Weather Forecasting

We evaluate forecasting performance using three standard metrics: Root Mean Square Error (RMSE), Normalized RMSE (NRMSE), and Anomaly Correlation Coefficient (ACC). To account for the convergence of meridians toward the poles, all

*Table 12.* Architecture configurations of EMFormer backbone models.

| | Output Size (Downs. Rate) | EMFormer-T | EMFormer-S | EMFormer-B |
|---|---|---|---|---|
| Stem | $112{\times}112$ $(\times\frac{1}{2})$ | Conv-BN-Act, C:32, S:2 (ResBlock, C:32) $\times$ 2 | Conv-BN-Act, C:32, S:2 (ResBlock, C:32) $\times$ 2 | Conv-BN-Act, C:32, S:2 (ResBlock, C:32) $\times$ 2 |
| Stage 1 | $56{\times}56$ $(\times\frac{1}{4})$ | Pruning, C:64, S:2 (ResBlock, C:64) $\times$ 2 | Pruning, C:64, S:2 (ResBlock, C:64) $\times$ 2 | Pruning, C:64, S:2 (ResBlock, C:64) $\times$ 6 |
| Stage 2 | $28{\times}28$ $(\times\frac{1}{8})$ | Pruning, C:128, S:2 (ResBlock, C:128) $\times$ 2 | Pruning, C:128, S:2 (ResBlock, C:128) $\times$ 4 | Pruning, C:128, S:2 (ResBlock, C:128) $\times$ 6 |
| Stage 3 | $14{\times}14$ $(\times\frac{1}{16})$ | Pruning, C:256, S:2 (EMFormer, C:256) $\times$ 2 (Windows-EMFormer, C:256) $\times$ 2 | Pruning, C:256, S:2 (EMFormer, C:256) $\times$ 5 (Windows-EMFormer, C:256) $\times$ 5 | Pruning, C:256, S:2 (EMFormer, C:256) $\times$ 6 (Windows-EMFormer, C:256) $\times$ 6 |
| Stage 4 | $7{\times}7$ $(\times\frac{1}{32})$ | Pruning, C:512, S:2 (EMFormer, C:512) $\times$ 1 | Pruning, C:512, S:2 (EMFormer, C:512) $\times$ 2 (Windows-EMFormer, C:512) $\times$ 2 | Pruning, C:512, S:2 (EMFormer, C:512) $\times$ 6 (Windows-EMFormer, C:512) $\times$ 6 |

metrics incorporate a latitude-dependent weight $L_i$. The definitions are:

$$\text{RMSE}(t) = \sqrt{\frac{\sum_{i=1}^{N_{\text{lat}}} \sum_{j=1}^{N_{\text{lon}}} L_i \big(\hat{X}_{i,j}^t - X_{i,j}^t\big)^2}{N_{\text{lat}} \times N_{\text{lon}}}}, \tag{41}$$

$$\text{NRMSE}(t) = \sqrt{\frac{\sum_{i=1}^{N_{\text{lat}}} \sum_{j=1}^{N_{\text{lon}}} L_i \big(\hat{Z}_{i,j}^t - Z_{i,j}^t\big)^2}{N_{\text{lat}} \times N_{\text{lon}}}}, \quad \text{with} \quad Z_{i,j}^t = \frac{X_{i,j}^t - \mu}{\sigma}, \quad \hat{Z}_{i,j}^t = \frac{\hat{X}_{i,j}^t - \mu}{\sigma}, \tag{42}$$

$$\text{ACC}(t) = \frac{\sum_{i=1}^{N_{\text{lat}}} \sum_{j=1}^{N_{\text{lon}}} L_i \, \hat{X}_{i,j}^t \, X_{i,j}^t}{\sqrt{\sum_{i=1}^{N_{\text{lat}}} \sum_{j=1}^{N_{\text{lon}}} L_i \big(\hat{X}_{i,j}^t\big)^2 \times \sum_{i=1}^{N_{\text{lat}}} \sum_{j=1}^{N_{\text{lon}}} L_i \big(X_{i,j}^t\big)^2}}, \tag{43}$$

where $\hat{X}_{i,j}^t$ and $X_{i,j}^t$ denote the predicted and ground-truth values at grid point $(i, j)$ and forecast time $t$; $N_{\text{lat}}$ and $N_{\text{lon}}$ are the numbers of grid cells in latitude and longitude. For NRMSE, $\mu$ and $\sigma$ are the temporal mean and standard deviation of the ground-truth variable over the training period, used to normalize both predictions and targets to a common scale. The latitude weight $L_i$ is defined as:

$$L_i = N_{\text{lat}} \times \frac{\cos \phi_i}{\sum_{j=1}^{N_{\text{lat}}} \cos \phi_j}, \tag{44}$$

with $\phi_i$ representing the latitude of the $i$-th grid row. To obtain a single scalar assessing overall performance across multiple variables, we also report the **Mean Normalized RMSE (MNRMSE)**, calculated as the arithmetic mean of the NRMSE values over all predicted variables.

### E.2. Typhoon Tracking

This study evaluates the proposed forecasting model using ten tropical cyclones that occurred across the 2024 season, selected as test cases for their intensity and representative tracks in the Western Pacific and East Asia region. The cyclones, listed with their international names and approximate formation periods, are: AMPIL (August 2024), BEBINCA (September 2024), EWINIAR (May 2024), GAEMI (July 2024), KONG-REY (October 2024), KRATHON (September 2024), MAN-YI (November 2024), SHANSHAN (August 2024), YAGI (September 2024), and YINXING (November 2024). All systems developed within the Western Pacific or adjacent East Asian maritime basins. The initial condition for each forecast is set at 00:00 UTC on its respective formation date. Ground-truth and the corresponding forecasts from all comparison models are

obtained from the publicly accessible CMA Tropical Cyclone Data Archive (TCData)[2].

Following previous AI-based approaches (Bi et al., 2023; Magnusson et al., 2021), we identify the tropical-cyclone eye as the location of the local minimum in mean sea-level pressure (MSL). The tracking algorithm employs a multi-constraint sequential method that combines MSL minimization with physical consistency checks to ensure robust and meteorologically plausible trajectory estimation.

The tracking procedure consists of the following steps: (1) initialize the current position with the given coordinates; (2) extract the latitude–longitude grid and relevant atmospheric variables (MSL, U10, V10); (3) locate the cyclone center by finding the minimum MSL within a 278 km search radius; (4) compute the central pressure, maximum wind speed, and displacement distance; (5) evaluate termination criteria against thresholds for pressure (101 200 Pa), wind speed ($10.2 m/s$), and displacement (400 km); (6) if all criteria are satisfied, update the current position and append the point to the trajectory.

To ensure the robustness of the tracking results, we conduct a comprehensive sensitivity analysis on the key threshold parameters. The choice of the 278 km search radius is based on the typical diameter of tropical cyclones, while the pressure (101 200 Pa) and wind-speed (10.2 $m/s$) thresholds follow the World Meteorological Organization definition of a tropical depression. We systematically vary each threshold within a plausible range (pressure: 101 000–102 000Pa; wind speed: 9.0–11.5$m/s$; search radius: 200–400 km) and evaluate the resulting track errors.

For typhoon track evaluation, we employ the Mean Distance Error (MDE), which averages the great-circle (Haversine) distance between predicted and observed cyclone centers over all forecast steps:

$$\text{MDE} = \frac{1}{N} \sum_{k=1}^{N} d\big(P_{\text{pred}}^{(k)}, P_{\text{obs}}^{(k)}\big), \tag{45}$$

$$d(P_1, P_2) = 2R \cdot \arcsin\big(\sqrt{a}\,\big), \tag{46}$$

$$a = \sin^2\Big(\frac{\Delta\phi}{2}\Big) + \cos\phi_1 \cdot \cos\phi_2 \cdot \sin^2\Big(\frac{\Delta\lambda}{2}\Big), \tag{47}$$

where $R = 6371$ km is the Earth's mean radius, $P_1 = (\phi_1, \lambda_1)$ and $P_2 = (\phi_2, \lambda_2)$ are the latitude–longitude pairs of two points, and $\Delta\phi = \phi_2 - \phi_1$, $\Delta\lambda = \lambda_2 - \lambda_1$.

### E.3. Image Classification on Imagenet-1K

Top-1 accuracy represents the most stringent evaluation metric for image classification models on the ImageNet-1K dataset. It measures the proportion of test images for which the model's single highest-confidence prediction matches the ground-truth label. Formally, given a validation set $\mathcal{D} = \{(\mathbf{x}_i, y_i)\}_{i=1}^{N}$ where $\mathbf{x}_i \in \mathbb{R}^{H \times W \times 3}$ denotes the input image, $y_i \in \{1, 2, \ldots, K\}$ is the true class label ($K = 1000$ for ImageNet-1K), and $N = 50,000$ is the size of the validation set, the top-1 accuracy is computed as:

$$\text{Accuracy}_{\text{top-1}} = \frac{1}{N} \sum_{i=1}^{N} \mathbb{I}\left(\arg\max_{k \in \{1,\ldots,K\}} f_k(\mathbf{x}_i; \theta) = y_i\right) \tag{48}$$

where:

- $f_k(\mathbf{x}_i; \theta)$ denotes the predicted probability for class $k$ given input $\mathbf{x}_i$ and model parameters $\theta$

- $\arg\max_k f_k(\mathbf{x}_i; \theta)$ identifies the class index with the highest prediction probability

- $\mathbb{I}(\cdot)$ is the indicator function that returns 1 when the condition is true and 0 otherwise

The evaluation protocol follows strict standardization: all images are center-cropped to $224 \times 224$ pixels after resizing the shorter edge to 256 pixels. No test-time augmentation is applied during top-1 accuracy computation, ensuring consistent comparison across different architectures. This metric provides a direct measure of a model's ability to make precise single-label predictions under real-world deployment scenarios where only one classification decision is permitted per input.

---

[2]https://tcdata.typhoon.org.cn

*Table 13.* Comprehensive performance comparison between Plain Multi-scale Module and Multi-Convs Layer. All metrics are measured on identical hardware and input conditions with NVIDIA A100 GPU.

| Metric | Plain Multi-scale Module | Multi-Convs Layer | |
|---|---|---|---|
| | Output Statistics | | Difference |
| Range (min/max) | -154.778732/155.702896 | -154.780792/155.698761 | |
| Mean | 0.001675 | 0.001684 | 0.000009 |
| Std Dev | 32.543854 | 32.543770 | 0.000084 |
| | Weight Gradients | | |
| Weight 1 / Layer 1 | 2.657446 | 2.657445 | 0.0000001 |
| Weight 2 / Layer 2 | 6.396124 | 6.396096 | 0.000028 |
| Weight 3 / Layer 3 | 8.638949 | 8.638924 | 0.000025 |
| | Time (s) | | Speedup Factor |
| Performance | 0.227616 | 0.039987 | 5.69× |

## E.4. Semantic Segmentation on ADE20K

The mean Intersection over Union (mIoU) serves as the primary evaluation metric for semantic segmentation performance on the ADE20K dataset. This metric quantifies the spatial overlap accuracy between predicted segmentation masks and ground-truth annotations across all semantic classes. For a segmentation model $f : \mathbb{R}^{H \times W \times 3} \to \{1, 2, \ldots, C\}^{H \times W}$ where $C = 150$ represents the number of semantic classes in ADE20K validation set, the mIoU is computed through the following formal procedure.

Given a validation image $\mathbf{x}$ with ground-truth segmentation mask $\mathbf{y} \in \{1, \ldots, C\}^{H \times W}$ and predicted segmentation $\hat{\mathbf{y}} = f(\mathbf{x})$, the Intersection over Union for class $c$ is defined as:

$$\text{IoU}_c = \frac{|\{p : y_p = c \wedge \hat{y}_p = c\}|}{|\{p : y_p = c \vee \hat{y}_p = c\}| + \epsilon} \tag{49}$$

where:

- $p$ indexes spatial positions in the segmentation map

- $y_p$ and $\hat{y}_p$ denote the ground-truth and predicted class labels at position $p$

- $\epsilon = 10^{-6}$ is a small constant to prevent division by zero when both sets are empty

- The numerator computes the true positive pixels for class $c$

- The denominator represents the union of ground-truth and predicted pixels for class $c$

The mean IoU across all $C$ semantic classes is then calculated as:

$$\text{mIoU} = \frac{1}{C} \sum_{c=1}^{C} \text{IoU}_c \tag{50}$$

For the ADE20K benchmark, evaluation follows strict protocol specifications: predictions are resized to match the original image resolution before computing mIoU, and the metric is computed exclusively on the official 2,000-image validation set. Notably, the background class (class 0) is excluded from evaluation, focusing solely on the 150 foreground semantic categories. This implementation adheres to the standard evaluation code provided by the MIT Scene Parsing Benchmark, ensuring consistent and comparable results across different segmentation architectures. The mIoU metric provides a robust measure of a model's ability to accurately delineate object boundaries and correctly classify regions across diverse indoor and outdoor scene categories.

*Table 14.* Ablation study of the loss function on weather forecasting tasks. A lower RMSE (denormalized, ↓) indicates better performance. **All experiments are conducted on a 6-year subset of ERA5 data spanning 2015–2020.** Consequently, the resulting error values are notably worse than those reported in earlier experiments, which used different evaluation periods.

| Loss Function | 6-hour (RMSE ↓) | | | 4-day (RMSE ↓) | | | 10-day (RMSE ↓) | | | Equation |
|---|---|---|---|---|---|---|---|---|---|---|
| | Z500 | T2M | U10 | Z500 | T2M | U10 | Z500 | T2M | U10 | |
| L2Loss | 34.4 | 0.731 | 0.651 | 236.42 | 1.63 | 2.21 | 851.34 | 3.65 | 4.82 | $\mathcal{E} = (\hat{\mathbf{X}}^{t+1} - \mathbf{X}^{t+1})^2$ |
| Variable-weighted Loss | 31.8 | 0.704 | 0.652 | 221.15 | 1.54 | 2.12 | 806.21 | 3.38 | 4.54 | $\frac{\mathcal{E}}{e^{\mathbf{w}}} + \mathbf{w}$ |
| Lat-weighted Loss | 31.5 | 0.707 | 0.659 | 219.08 | 1.49 | 2.08 | 794.55 | 3.26 | 4.45 | $\mathbf{L} \odot \mathcal{E}$ |
| Lat- and Variable- weighted Loss | 32.5 | 0.713 | 0.643 | 217.96 | 1.46 | 2.05 | 789.68 | 3.21 | 4.41 | $\mathbf{L} \odot \left(\frac{\mathcal{E}}{e^{\mathbf{w}}} + \mathbf{w}\right)$ |
| Hybrid Loss ($\lambda_{init} = 0.2$) | 31.1 | 0.796 | 0.651 | 215.82 | 1.31 | 2.01 | 764.51 | 2.82 | 4.27 | |
| Hybrid Loss ($\lambda_{init} = 0.3$) | 32.1 | 0.707 | 0.649 | 221.57 | 1.30 | 2.08 | 802.67 | 2.81 | 4.41 | $\mathbf{L} \odot \mathcal{E} + \lambda * \left(\frac{\mathcal{E}}{e^{\mathbf{w}}} + \mathbf{w}\right), \lambda$ is trainable |
| Hybrid Loss ($\lambda_{init} = 0.4$) | 35.7 | 0.731 | 0.654 | 227.74 | 1.36 | 2.18 | 833.6 | 2.92 | 4.61 | |
| Hybrid Loss (Sinusoidal Weighting) | 31.6 | 0.696 | 0.641 | 213.85 | 1.30 | 2.02 | 762.14 | 2.75 | 4.28 | $\frac{1}{2}\big(1 + \sin(\theta)\big)\big(\mathbf{L} \odot \mathcal{E}\big) + \frac{1}{2}\big(1 - \sin(\theta)\big)\big(\frac{\mathcal{E}}{e^{\mathbf{w}}} + \mathbf{w}\big)$ |
| Hybrid Loss (Ours) | 30.3 | 0.684 | 0.626 | 212.34 | 1.28 | 1.99 | 756.67 | 2.72 | 4.22 | $\frac{1}{2}\big(1 - \sin(\theta)\big)\big(\mathbf{L} \odot \mathcal{E}\big) + \frac{1}{2}\big(1 + \sin(\theta)\big)\big(\frac{\mathcal{E}}{e^{\mathbf{w}}} + \mathbf{w}\big)$ |

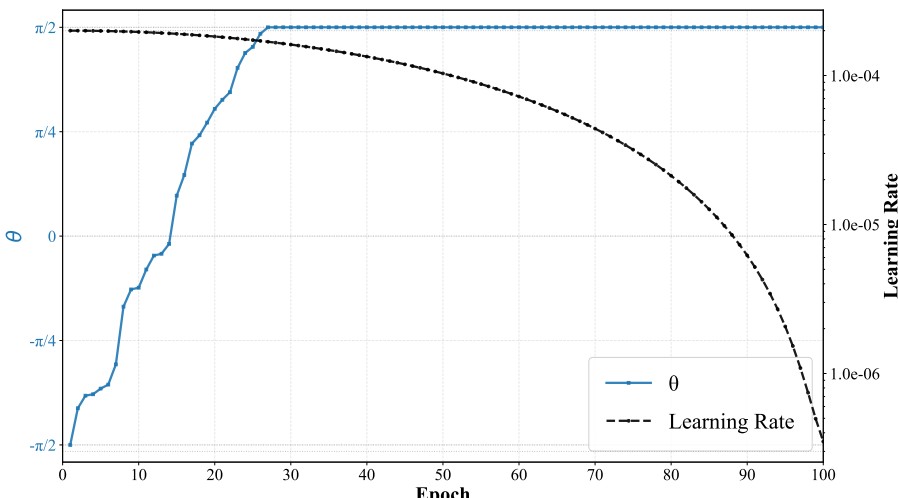

*Figure 7.* Training curves of learning rate and $\theta$ in 100 epochs.

## F. Additional Experiments

### F.1. Experiments of Proposition

In this section, we present experimental results to validate our theoretical claims.

**Experiment on Proposition 1.** The performance comparison between the plain multi-scale module and our proposed multi-convs layer is summarized in Tab. 13. Given identical input features and initialization, the differences in output statistics and weight gradients between the two modules remain below $10^{-4}$, confirming the functional equivalence of their forward and backward passes in Sec. A.2. At the same time, the forward and backward runtimes of the multi-convs layer are reduced by a factor of 5.69× relative to the plain multi-scale module.

**Experiment on Proposition 2.** To validate the proof of Proposition 2 given in Sec. A.2, we visualize the evolution of $\theta$ alongside the learning rate during training (Fig. 7). The curve shows that $\theta$ starts from $-\pi/2$ and stabilizes near $\pi/2$ after approximately 30 epochs.

To further assess the effectiveness of the proposed loss, we conduct an ablation study on a 6-year subset of ERA5 data, comparing eight different loss formulations (Tab. 14). The results indicate that both the variable-weighted loss and the latitude-weighted loss individually outperform the plain L2 loss. Interestingly, simply combining the two weighting strategies without adaptive balancing yields worse performance than either alone. In contrast, the hybrid loss with sine-based balancing achieves the best results, improving Z500, T2M, T850, and U10 by 0.8, 0.02, 0.024, and 0.017, respectively, over

*Table 15.* Comparative Analysis of Training Times and Hardware Specifications for Deep Learning Models. ∗ denotes that the reported time includes only the training duration, excluding finetuning. GPU cost and Latency are all computed by us with one A100(40G) GPU.

| Model | Params (M) | MACs (G) | GPUs | Training Time | GPU Cost (Inference, M) | Latency (ms) |
|---|---|---|---|---|---|---|
| $721 \times 1440$ (0.25°) | | | | | | |
| Fengwu (Chen et al., 2025b) | 153.49 | 132.83 | 32 A100 | 17 days | 1363.1 | 227 |
| FourCastNet (Kurth et al., 2023) | 79.6 | 111.62 | 64 A100 | 16 hours | 868.0 | 164.0 |
| Graphcast (Lam et al., 2023) | 28.95 | 1639.26 | 32 TPUv4 | 4 weeks | >40G | - |
| Pangu-Weather (Bi et al., 2023) | 23.83 | 142.39 | 192 V100 | 64 days | 997.1 | 275 |
| Ours | 208.70 | 124.89 | 16 A100 | 9 days | 962.2 | 177.6 |
| $128 \times 256$ (1.4°) | | | | | | |
| VA-MoE (Chen et al., 2025a) | 665.37 | - | 16 A100 | 12 days∗ | 688.3 | 269 |
| OneForecast (Gao et al., 2025) | 24.76 | 509.27 | 16 A100 | 8 days∗ | 225.3 | 663 |
| STCast (Chen et al., 2026) | 654.82 | 436.12 | 16 A100 | 5 days∗ | 546.7 | 115 |
| Ours | 157.9 | 102.54 | 16 A100 | 60 hours∗ | 703.3 | 98.3 |

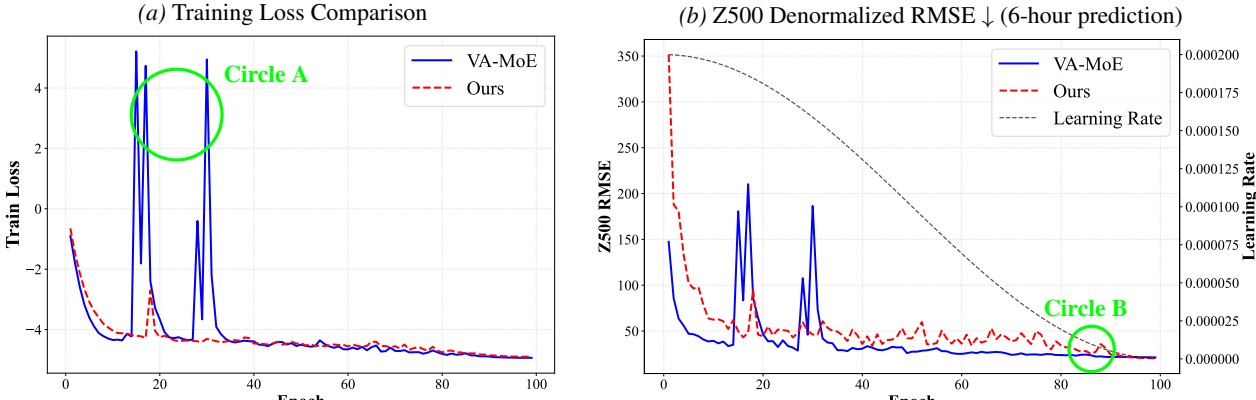

*Figure 8.* Comparison between the training processes of VA-MoE (Chen et al., 2025a) and the proposed method. (a) Training loss comparison: Ours and VA-MoE. (b) Short-term forecast performance: evolution of Z500 RMSE during training for Ours and VA-MoE.

the best single-weighted variant. These experiments confirm that our hybrid loss effectively guides the model to balance variable-wise and geographic information, leading to superior forecast accuracy.

### F.2. Training Time

As shown in Tab. 15, we compare the number of parameters, multiply–accumulate operations (MACs), GPU usage, training duration, GPU cost in inference, and Latency across eight baseline models. While our model has more parameters and MACs than GNN-based methods such as GraphCast and OneForecast, its overall computational cost remains substantially lower than that of large-scale predecessors, notably FengWu, Pangu-Weather, and GraphCast. These results show that our approach attains better performance while retaining competitive efficiency.

**How does the model achieve better performance at lower cost?** We examine two training metrics in Fig. 8 and highlight two observations. (1) Fig. 8a compares training losses between VA-MoE (Chen et al., 2025a) and our model. Although VA-MoE converges more rapidly in early epochs, its loss curve displays sharp, irregular spikes (marked in Circle A), whereas our loss decreases smoothly throughout training. (2) Fig. 8b compares Z500 RMSE during training. For the first 90 epochs (under a higher learning rate), our model lags behind VA-MoE. However, in the final 10 epochs, our model exhibits a sharp drop in RMSE when a much lower learning rate is applied (see Circle B). As noted in Sec. 2.4, lower learning rates favor the optimization of atmospheric variables. Our training schedule aligns with this principle, thereby harmonizing effectively with the characteristics of atmospheric data.

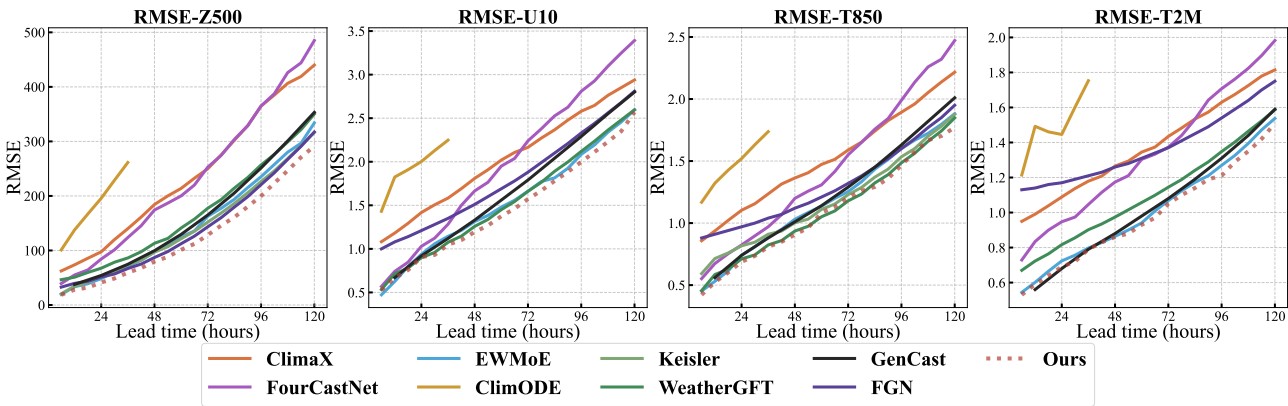

*Figure 9.* 120-hour comparative analysis of **RMSE** ↓ across 10 data-driven models for four variables, including Z500, T850, T2M, and U10. Results are collected from EWMoE (Gan et al., 2025), WeatherGFT (Xu et al., 2024) and WeatherBench (Rasp et al., 2024) in https://sites.research.google/gr/weatherbench/deterministic-scores.

*Table 16.* Efficiency and Accuracy Comparison. **(a)** Module-level efficiency comparison (average of 50 inferences). **(b)** End-to-end forecasting performance (Denormalized RMSE on 1.4° ERA5, 6-year quick training) alongside overall latency and MACs.

**(a) Single Module Efficiency Comparison**

| Single Module | Params (M) | Latency (ms) | GPU Memory (MB) |
|---|---|---|---|
| RepVGG (Ding et al., 2021) | 9.17 | 14.3 | 324 |
| RepLKNet (Ding et al., 2022) | 9.17 | 14.5 | 318 |
| Plain Multi-Scale Module | 9.17 | 31.2 | 301 |
| Multi-Convs Layer (Ours) | 9.17 | 8.9 | 195 |

**(b) End-to-end Denormalized RMSE and Efficiency**

| Method | 6-hour | | | 4-day | | | 10-day | | | Latency (ms) | MACs (G) |
|---|---|---|---|---|---|---|---|---|---|---|---|
| | Z500 | T2M | U10 | Z500 | T2M | U10 | Z500 | T2M | U10 | | |
| RepVGG (Ding et al., 2021) | 32.7 | 0.715 | 0.664 | 319.37 | 2.81 | 2.49 | 871.08 | 4.32 | 4.66 | 110.6 | 119.3 |
| RepLKNet (Ding et al., 2022) | 32.3 | 0.719 | 0.671 | 229.72 | 1.37 | 2.20 | 835.87 | 2.93 | 4.64 | 113.7 | 122.1 |
| EMFormer (Ours) | 30.3 | 0.684 | 0.626 | 212.34 | 1.28 | 1.99 | 764.51 | 2.72 | 4.27 | 98.3 | 102.5 |

### F.3. Additional Results

**Additional weather forecasting analysis.** As shown in Fig. 9, we compare the proposed model with several baselines over forecast horizons from 6 to 120 hours on four key atmospheric variables. Results indicate that our model performs on par with WeatherGFT (Xu et al., 2024) and Keisler (Keisler, 2022) in predicting T850 and U10, while surpassing all compared methods in Z500 and T2M. These outcomes underscore the effectiveness of the proposed approach and its integrated multi-scale architecture, demonstrating a consistent ability to capture atmospheric patterns across different spatial regions and diverse meteorological variables.

**Additional typhoon track prediction.** In addition to the aggregate errors reported in Tab. 2, we provide more detailed error visualizations for individual lead times in Fig. 10. These plots show the Mean Distance Error (MDE, in kilometers) across nine forecasting systems at each forecast step. The results indicate that our model achieves competitive accuracy in the short term while delivering superior performance at longer lead times, particularly for Typhoons Bebinca, Kong-rey, and Yagi (see Figs. 10b, 10e and 10i). Such detailed comparisons further confirm the model's strong capability in extreme-weather forecasting.

To illustrate the trajectory forecasts concretely, Fig. 11 visualizes the predicted tracks of Typhoon Ampil (Fig. 11a) and Typhoon Yinxing (Fig. 11b) alongside those of nine baseline methods. Additional visualizations in Figs. 11c to 11e further dissect the evolution of Typhoon Yinxing by contrasting the initial conditions, model predictions, and ground-truth observations. Collectively, these results substantiate the robustness of the proposed approach for extreme-event assessment

*Table 17.* Ablation study on the accumulative context finetuning strategy with VA-MoE (Chen et al., 2025a). All results are trained with 1.4°ERA5. The model is trained by us followed official code.

| VA-MoE | Metric | | | | | | | | | |
|---|---|---|---|---|---|---|---|---|---|---|
| | 6-hour | | 1-day | | 4-day | | 7-day | | 10-day | |
| | RMSE | ACC | RMSE | ACC | RMSE | ACC | RMSE | ACC | RMSE | ACC |
| w/o finetuning | 0.0613 | 0.9938 | 0.1248 | 0.9715 | 0.2699 | 0.8737 | 0.4436 | 0.6899 | 0.5855 | 0.4481 |
| w/ Finetuning | 0.0630 | 0.9925 | 0.1205 | 0.9734 | 0.2603 | 0.8849 | 0.4293 | 0.6966 | 0.5673 | 0.4701 |
| w/ Accumulative finetuning | 0.0608 | 0.9941 | 0.1161 | 0.9761 | 0.2451 | 0.8896 | 0.4101 | 0.7194 | 0.5128 | 0.5351 |

*Table 18.* Ablation study on the balancing hyperparamter $\lambda$ in the memory module of accumulative context finetuning. All results are gained with 1.4°ERA5 on 10-step accumulative context finetuning with 5 cache length.

| $\lambda$ | Metric | | | | | | | | | |
|---|---|---|---|---|---|---|---|---|---|---|
| | 6-hour | | 1-day | | 4-day | | 7-day | | 10-day | |
| | RMSE | ACC | RMSE | ACC | RMSE | ACC | RMSE | ACC | RMSE | ACC |
| 0.7 | 0.0599 | 0.9948 | 0.1139 | 0.9774 | 0.2444 | 0.8924 | 0.4099 | 0.7190 | 0.5121 | 0.5253 |
| 0.8 | 0.0598 | 0.9949 | 0.1140 | 0.9775 | 0.2441 | 0.8932 | 0.4087 | 0.7211 | 0.5109 | 0.5316 |
| 0.9 | 0.0599 | 0.9949 | 0.1139 | 0.9775 | 0.2439 | 0.8936 | 0.4072 | 0.7223 | 0.5094 | 0.5389 |
| 1.0 | 0.0599 | 0.9949 | 0.1139 | 0.9775 | 0.2462 | 0.8901 | 0.4139 | 0.7173 | 0.5204 | 0.5218 |

and demonstrate its clear advantage in accurately predicting tropical-cyclone tracks over extended forecast horizons.

### F.4. Additional Ablation Study

**Efficiency and Accuracy Comparison.** Tab. 16 comprehensively evaluates the efficiency and accuracy of our proposed approach against existing structural re-parameterization methods, such as RepVGG (Ding et al., 2021) and RepLKNet (Ding et al., 2022). While conventional methods implement multi-branch structures at the framework (*e.g.*, PyTorch) level, meaning the training phase still incurs the full multi-branch computational cost and speedups are restricted primarily to inference, our approach pushes this optimization to the CUDA level. By rewriting the low-level execution path, the Multi-Convs Layer efficiently executes multi-scale convolutions while preserving the original representation, enabling acceleration in both training and inference.

As demonstrated in Table 16a, under an identical parameter budget (9.17M), the Multi-Convs Layer achieves the lowest latency (8.9 ms) and GPU memory footprint (195 MB), significantly outperforming the plain multi-scale module and Rep-based baselines. Furthermore, Tab. 16b demonstrates that this module-level efficiency directly translates into superior end-to-end forecasting performance. When integrated into the overall architecture, EMFormer not only records the lowest inference latency (98.3 ms) and computational cost (102.5G MACs), but also consistently yields the best denormalized RMSE scores across all tested forecast horizons (6-hour, 4-day, and 10-day) for key meteorological variables including Z500, T2M, and U10.

**Finetuning strategy on VA-MoE.** Tab. 17 presents an ablation study comparing the VA-MoE model (Chen et al., 2025a) without finetuning and with the proposed accumulative context finetuning strategy. Results show that accumulative finetuning consistently improves forecast accuracy, increasing ACC by 0.0046, 0.0159, 0.0295, and 0.087 for 1-day, 4-day, 7-day, and 10-day predictions, respectively. These gains demonstrate that the accumulative context finetuning strategy is not only effective in our architecture, but also generalizes well to other frameworks such as VA-MoE.

**Balancing hyperparameter $\lambda$ in accumulative context finetuning.** Tab. 18 compares different values of the balancing hyperparameter $\lambda$ during the accumulative context finetuning stage. Experimental results indicate that the choice of $\lambda$ has only a minor effect on short-term forecast accuracy. Even for long-term predictions, a noticeable performance decline occurs only when $\lambda = 1.0$, *i.e.*, when the updated scores rely exclusively on the current token without blending historical information. These observations suggest that completely disregarding historical token information leads to degraded forecast consistency. Therefore, we select $\lambda = 0.9$ as an effective trade-off, balancing the contributions of current and historical tokens in the cache-update process.

**Cache length in accumulative context finetuning.** Tab. 19 presents an ablation study on the cache length during

*Table 19.* Ablation study on the cache length in the accumulative context finetuning with 10 steps. All results are gained with 1.4°ERA5.

| Cache length | Metric | | | | | | | | | |
|---|---|---|---|---|---|---|---|---|---|---|
| | 6-hour | | 1-day | | 4-day | | 7-day | | 10-day | |
| | RMSE | ACC | RMSE | ACC | RMSE | ACC | RMSE | ACC | RMSE | ACC |
| 3 | 0.0599 | 0.9948 | 0.1186 | 0.9748 | 0.2511 | 0.8834 | 0.4293 | 0.6991 | 0.5461 | 0.4875 |
| 4 | 0.0598 | 0.9949 | 0.1149 | 0.9760 | 0.2451 | 0.8921 | 0.4101 | 0.7201 | 0.5147 | 0.5219 |
| 5 | 0.0599 | 0.9949 | 0.1139 | 0.9775 | 0.2439 | 0.8936 | 0.4072 | 0.7223 | 0.5094 | 0.5389 |
| 6 | 0.0598 | 0.9950 | 0.1135 | 0.9778 | 0.2431 | 0.8941 | 0.4067 | 0.7229 | 0.5081 | 0.5399 |

*Table 20.* Ablation study on the number of steps in the finetuning stage with cache length 5. All results are gained with 1.4°ERA5.

| #Steps | Metric | | | | | | | | | |
|---|---|---|---|---|---|---|---|---|---|---|
| | 6-hour | | 1-day | | 4-day | | 7-day | | 10-day | |
| | RMSE | ACC | RMSE | ACC | RMSE | ACC | RMSE | ACC | RMSE | ACC |
| 0 | 0.0626 | 0.9931 | 0.1219 | 0.9749 | 0.2673 | 0.8845 | 0.4327 | 0.6978 | 0.5719 | 0.4614 |
| 5 | 0.0624 | 0.9945 | 0.1210 | 0.9752 | 0.2483 | 0.8871 | 0.4235 | 0.7081 | 0.5302 | 0.5098 |
| 6 | 0.0615 | 0.9947 | 0.1201 | 0.9756 | 0.2481 | 0.8875 | 0.4202 | 0.7101 | 0.5236 | 0.5168 |
| 7 | 0.0610 | 0.9947 | 0.1168 | 0.9766 | 0.2468 | 0.8906 | 0.4156 | 0.7168 | 0.5194 | 0.5229 |
| 8 | 0.0610 | 0.9948 | 0.1164 | 0.9767 | 0.2445 | 0.8926 | 0.4103 | 0.7197 | 0.5185 | 0.5227 |
| 9 | 0.0603 | 0.9949 | 0.1152 | 0.9771 | 0.2441 | 0.8928 | 0.4088 | 0.7221 | 0.5109 | 0.5314 |
| 10 | 0.0599 | 0.9949 | 0.1139 | 0.9775 | 0.2439 | 0.8936 | 0.4072 | 0.7223 | 0.5094 | 0.5389 |

accumulative context finetuning. As expected, increasing the cache length from 3 to 6 yields improved forecast performance. However, the gain between cache lengths 5 and 6 is marginal, with RMSE improvements of only 0.0005 and 0.0013 for 7-day and 10-day predictions, respectively. Meanwhile, computational cost grows nearly linearly with cache length. We therefore select a cache length of 5 as a practical trade-off between predictive accuracy and computational overhead, and adopt this value for pruning the KV values in our experiments.

**Number of steps in finetuning.** Tab. 20 presents an ablation study on the number of steps used in the accumulative context finetuning stage, with step counts ranging from 5 to 10. As expected, forecast performance improves as the number of steps increases. After reaching 9 steps, however, the weather forecasting accuracy stabilizes and shows no further gains. Given that computational cost grows with the number of steps, we select 10 steps as a practical compromise between performance and efficiency for the experiments in this work.

**Window sizes in window attention.** Tab. 21 presents an ablation study comparing a fixed $4 \times 4$ window size with a hybrid configuration. Previous experiments indicate that using a single window size can introduce grid-like artifacts in the predictions, which is inconsistent with the smooth spatial distribution of atmospheric variables on a global scale. In contrast, hybrid window sizes yield better performance and effectively mitigate these gridding artifacts.

### F.5. Visualization

We present visualizations of semantic segmentation results on ADE20K, comparing ground-truth annotations with the predictions generated by EMFormer small and base models in Fig. 12.

We also provide more visualization about global weather forecasting of 6-hour, 1-day, 2-day, 3-day, 5-day, 7-day, and 10-day in Fig. 13, Fig. 14, Fig. 15, Fig. 16, Fig. 17, Fig. 18, and Fig. 19.

## G. Supplementary Materials

To further ensure the transparency and reproducibility of our experimental results, we have included supplementary materials containing: partial test codes for the implemented models, complete training logs for both weather prediction and image classification tasks, and the raw latitude-longitude coordinates used for visualizing typhoon trajectories.

*Table 21.* Ablation study on the windows size of the windows-attention. All results are gained with 1.4°ERA5.

| Windows size | Metric | | | | | | | | | |
|---|---|---|---|---|---|---|---|---|---|---|
| | 6-hour | | 1-day | | 4-day | | 7-day | | 10-day | |
| | RMSE | ACC | RMSE | ACC | RMSE | ACC | RMSE | ACC | RMSE | ACC |
| Only $4 \times 4$ | 0.0641 | 0.9922 | 0.1228 | 0.9735 | 0.2679 | 0.8819 | 0.4395 | 0.6885 | 0.5878 | 0.4465 |
| Hybrid Sizes ($4 \times 4, 8 \times 2, 2 \times 8$) | 0.0626 | 0.9931 | 0.1219 | 0.9749 | 0.2673 | 0.8845 | 0.4327 | 0.6978 | 0.5719 | 0.4614 |

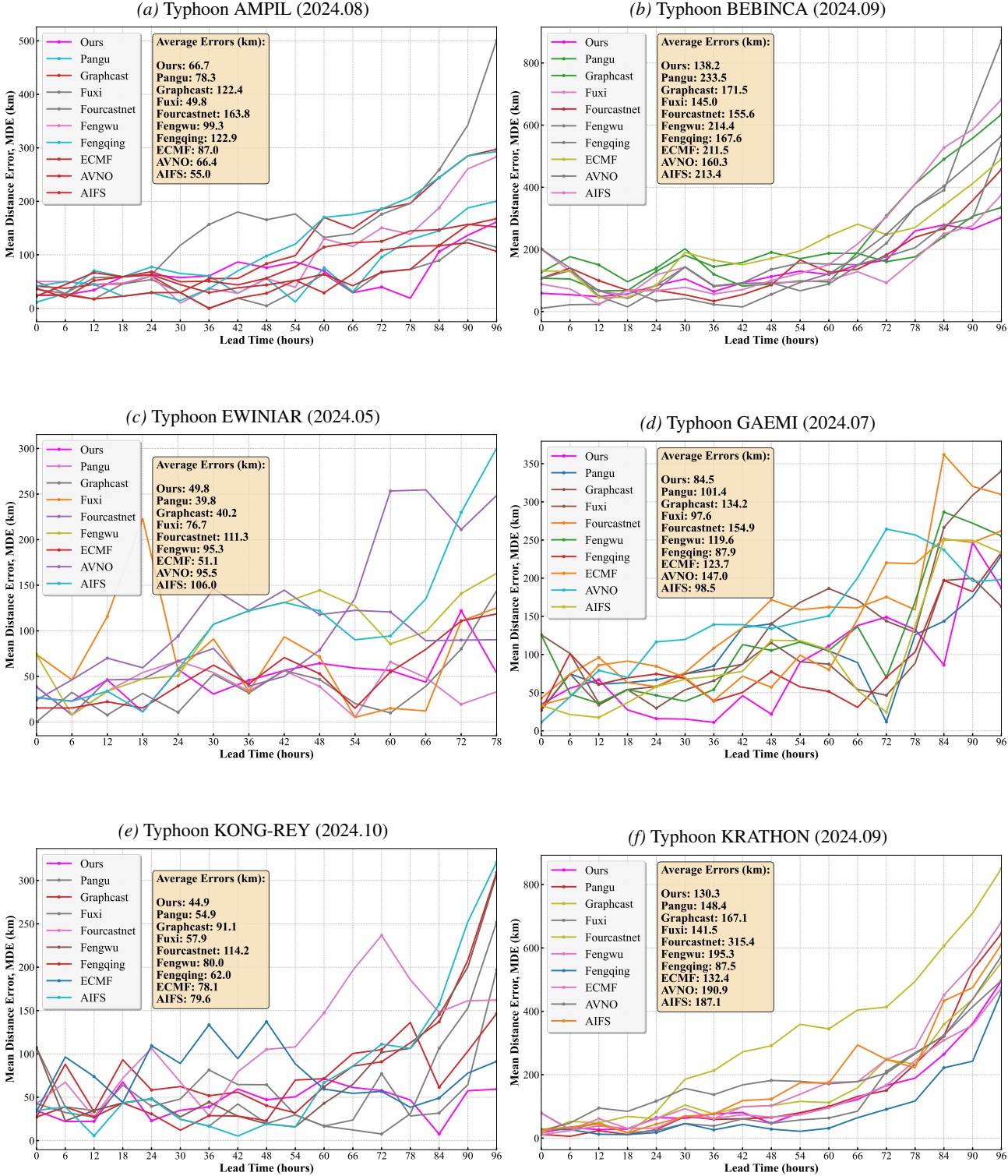

*Figure 10.* Typhoon Track Assessment (Part 1 of 2). Comparative analysis of Mean Distance Error (MDE, in kilometers ↓) for six typhoons. (a)-(f) show the prediction accuracy across different lead times.

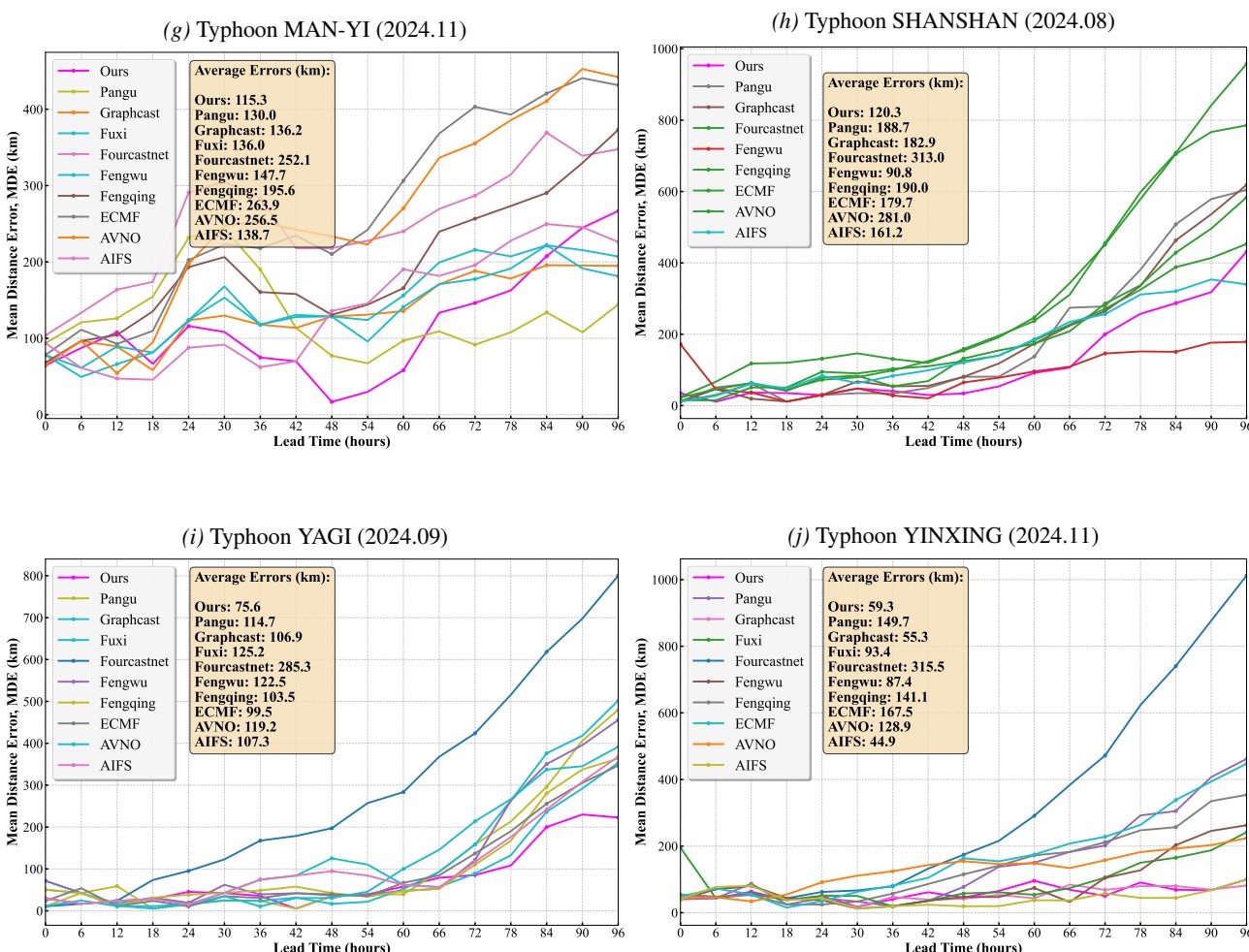

*Figure 10.* (Continued) Typhoon Track Assessment (Part 2 of 2). (g)-(j) continue the comparative analysis for the remaining four typhoons. The complete assessment demonstrates consistent performance patterns across different weather conditions and seasons.

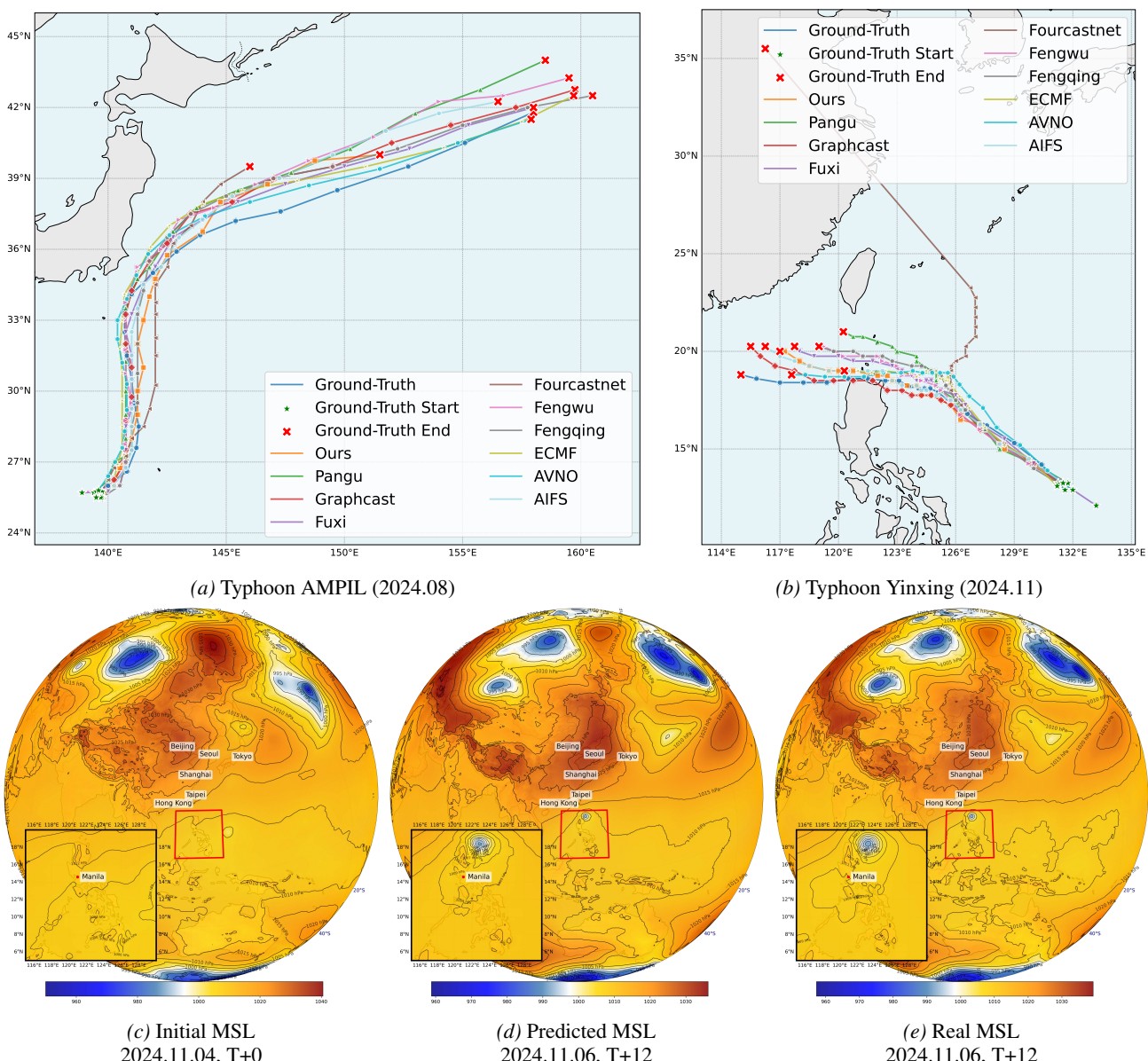

*(a)* Typhoon AMPIL (2024.08)

*(b)* Typhoon Yinxing (2024.11)

*(c)* Initial MSL
2024.11.04, T+0

*(d)* Predicted MSL
2024.11.06, T+12

*(e)* Real MSL
2024.11.06, T+12

*Figure 11.* Typhoon Track Assessment. (a) and (b) present a five-day comparative analysis of Mean Distance Error (MDE, in kilometers ↓) for Typhoons AMPIL and Yinxing, respectively. (c), (d), and (e) visualize the evolution of Mean Sea-Level Pressure (MSL) across three temporal stages: initial conditions, 60-hour predictions, and corresponding ground-truth observations.

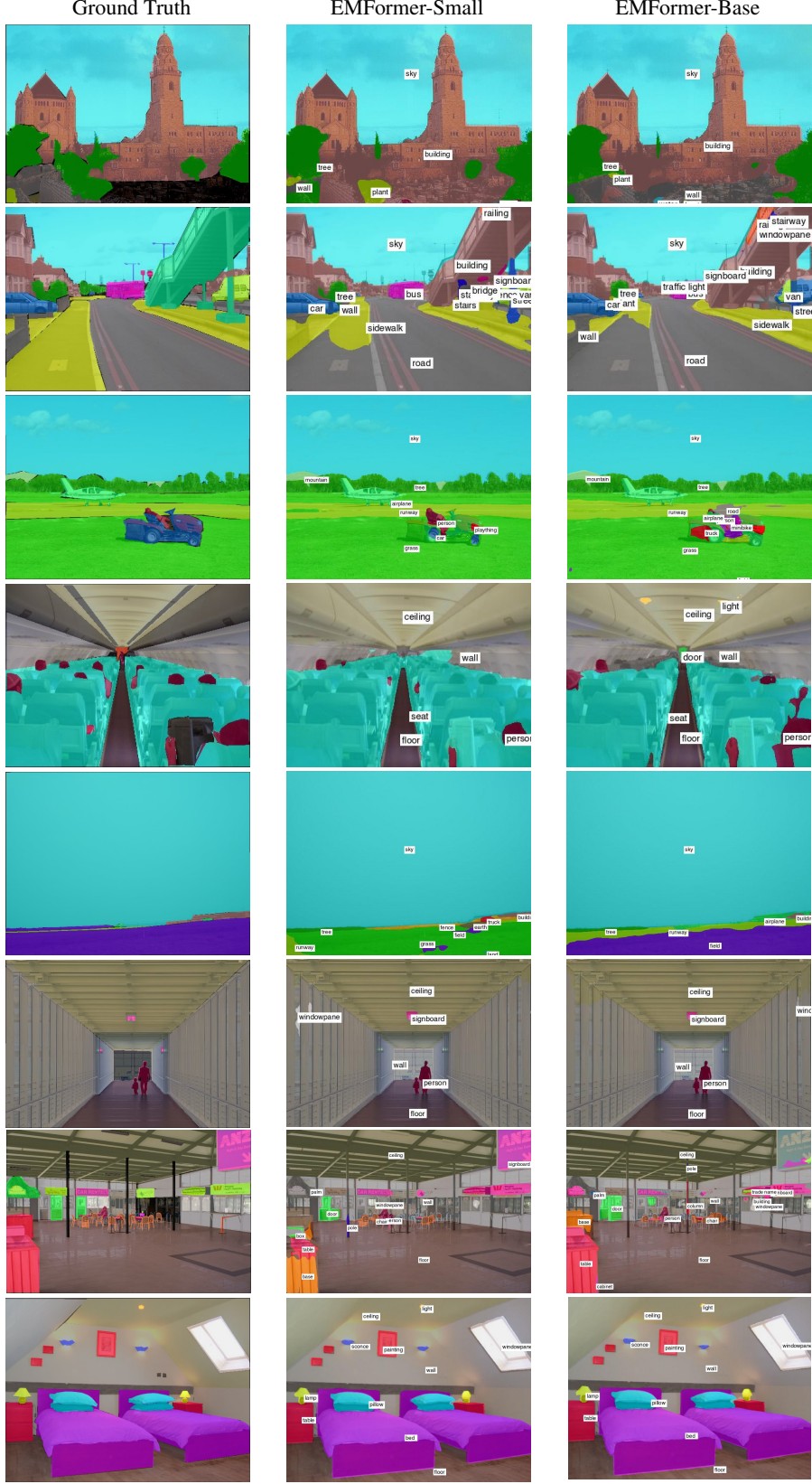

*Figure 12.* Visualization of segmentation results on ADE20K, comparing ground-truth annotations with predictions from our small and base models.

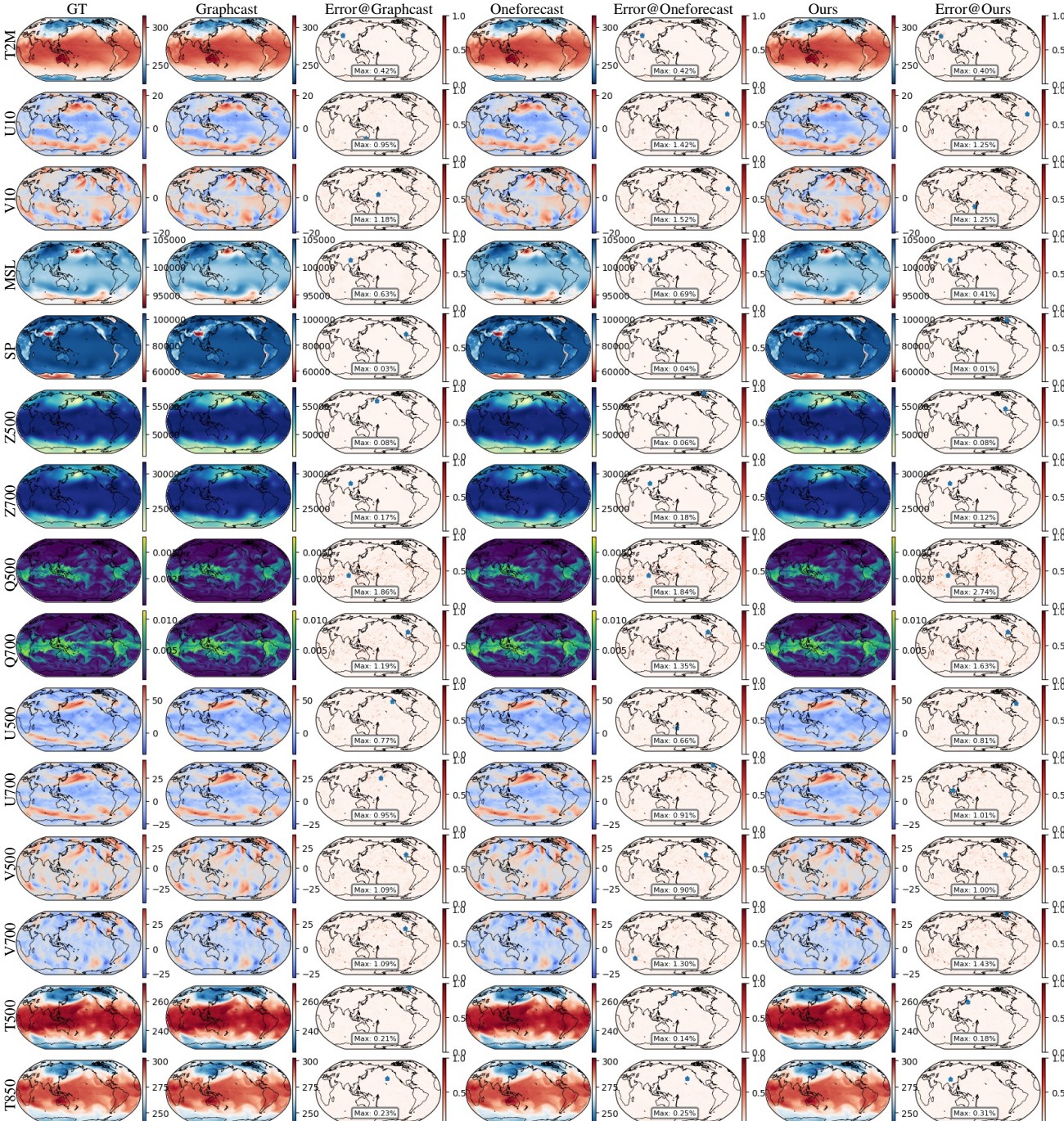

*Figure 13.* 6-hour forecast results of global weather among different models.

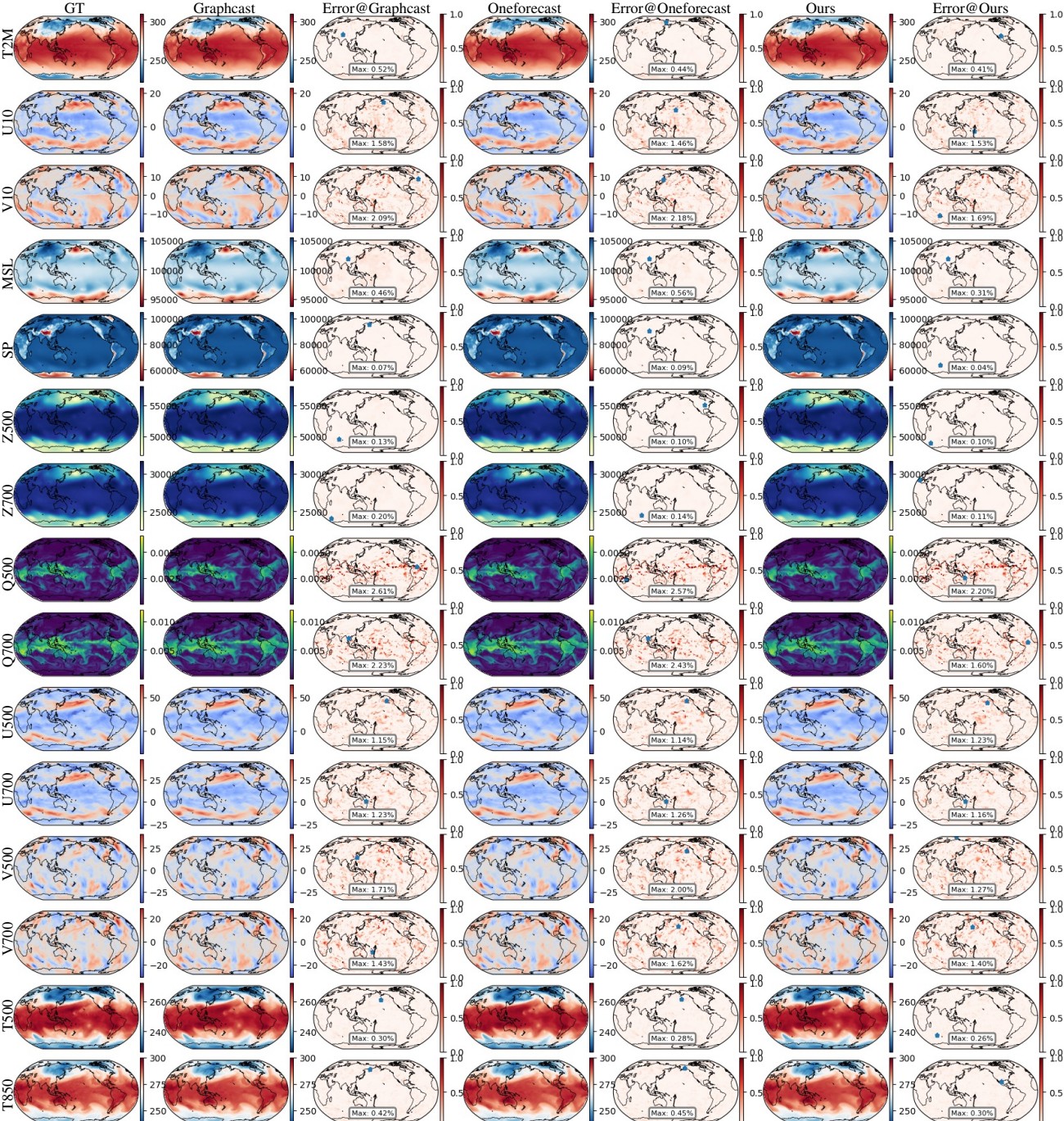

*Figure 14.* 1-day forecast results of global weather among different models.

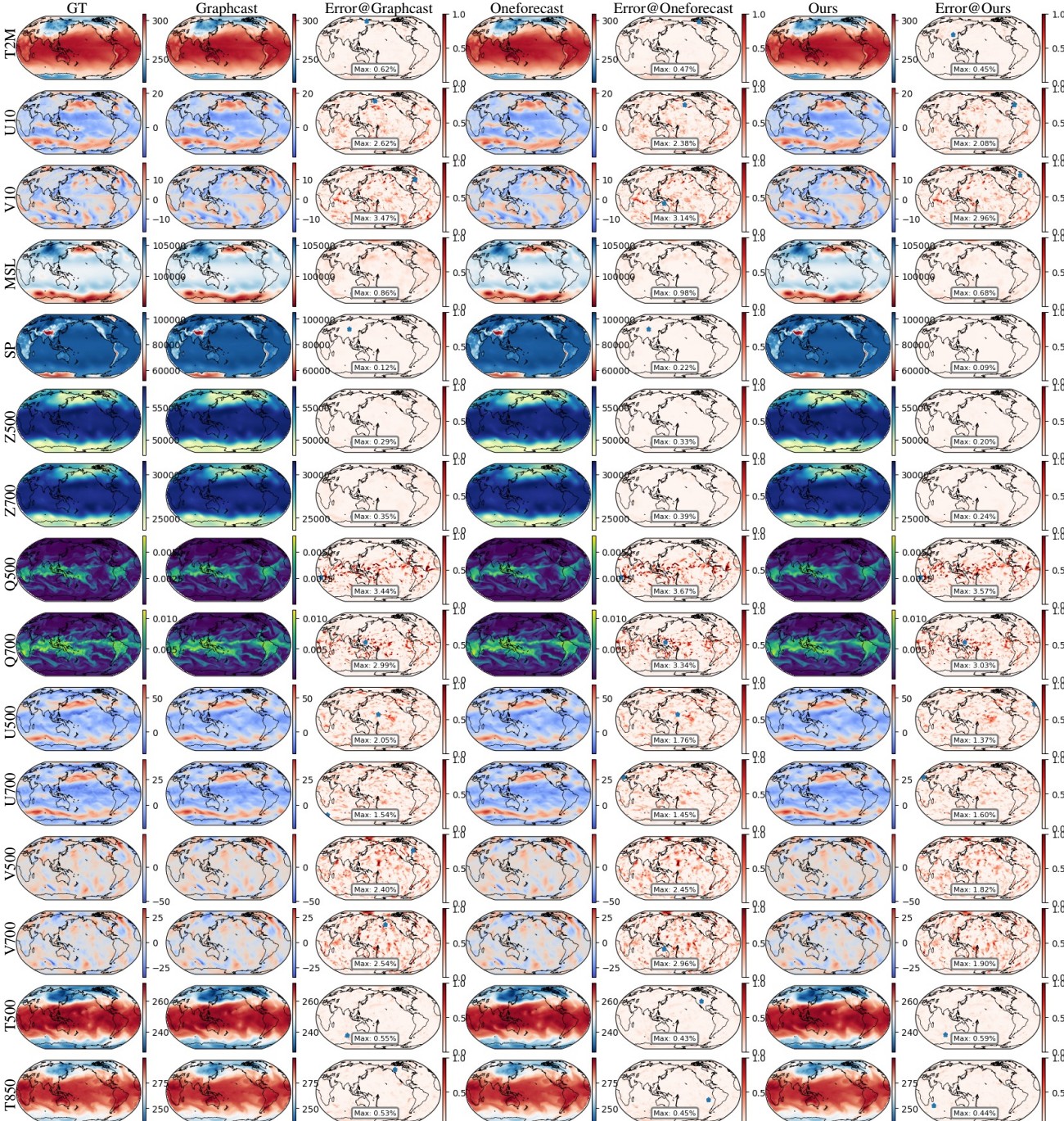

*Figure 15.* 2-day forecast results of global weather among different models.

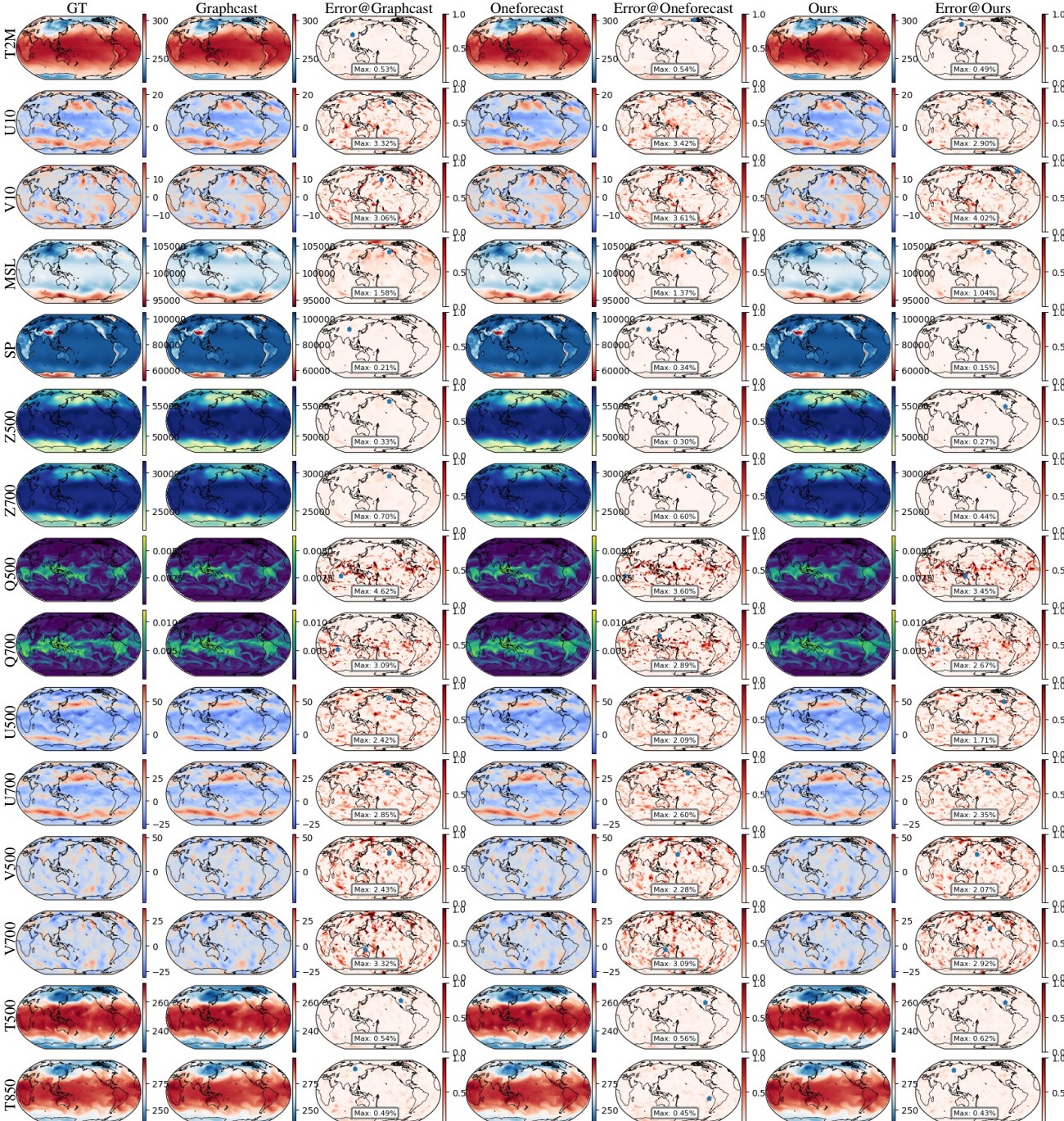

*Figure 16.* 3-day forecast results of global weather among different models.

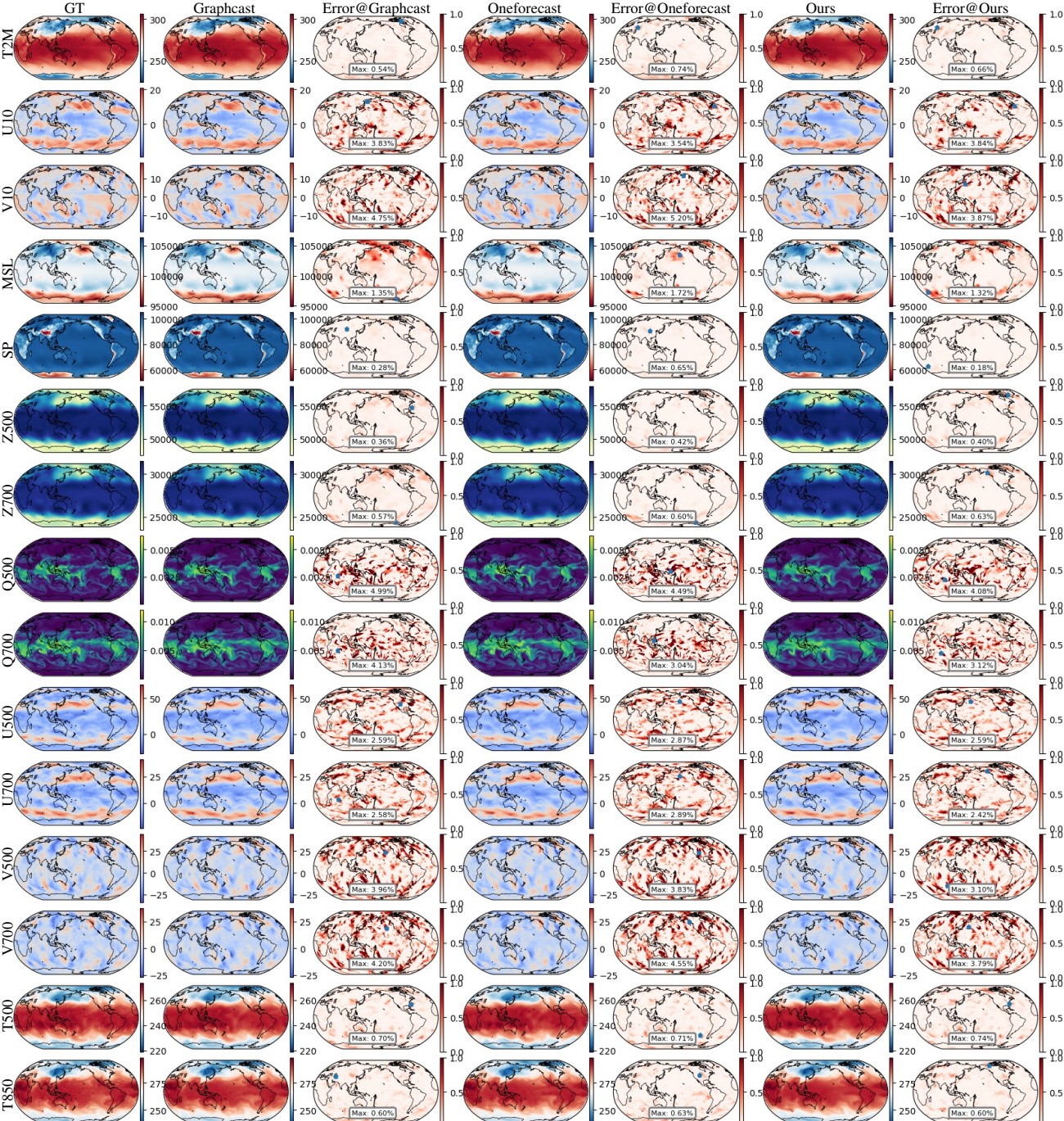

*Figure 17.* 5-day forecast results of global weather among different models.

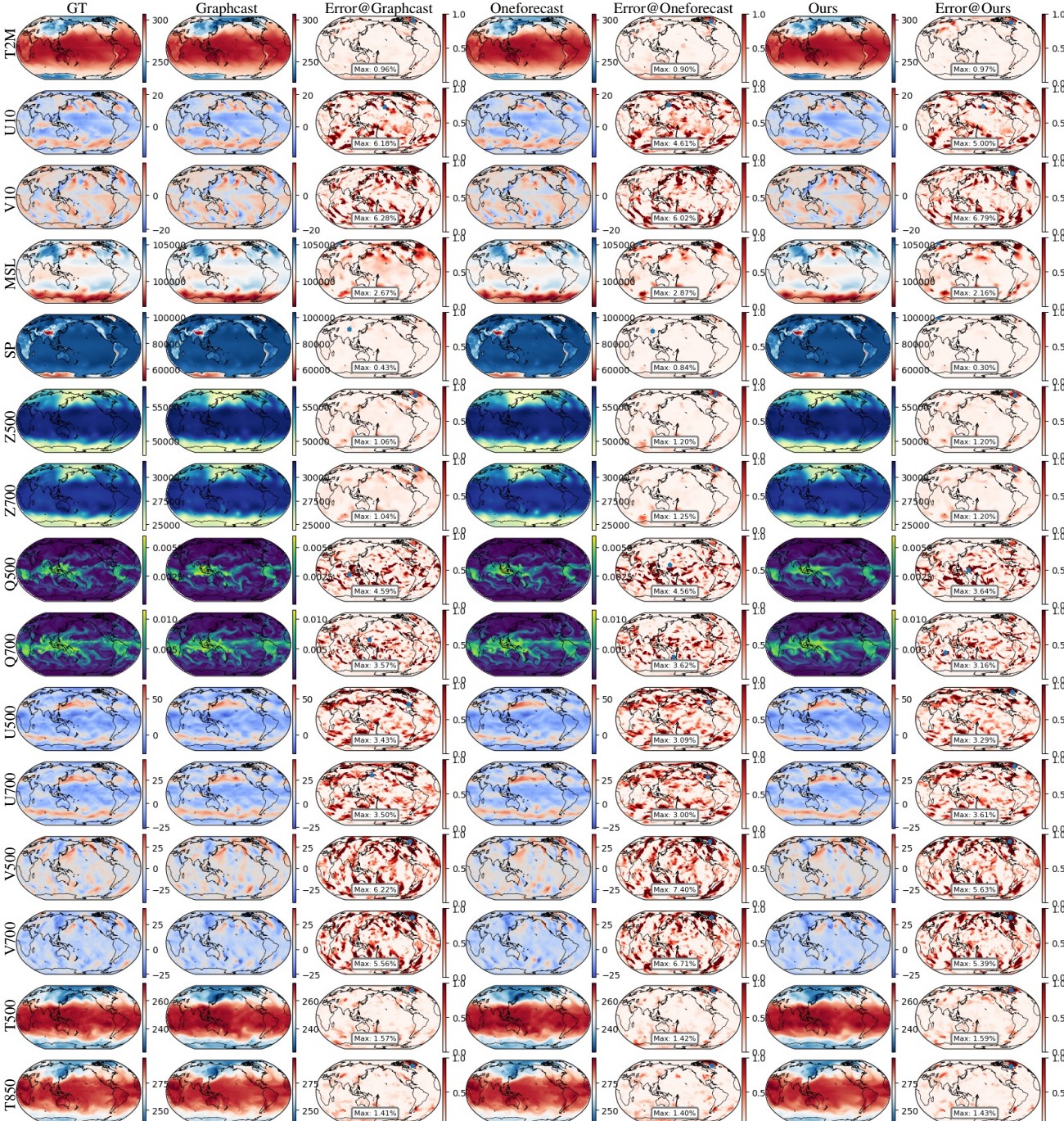

*Figure 18.* 7-day forecast results of global weather among different models.

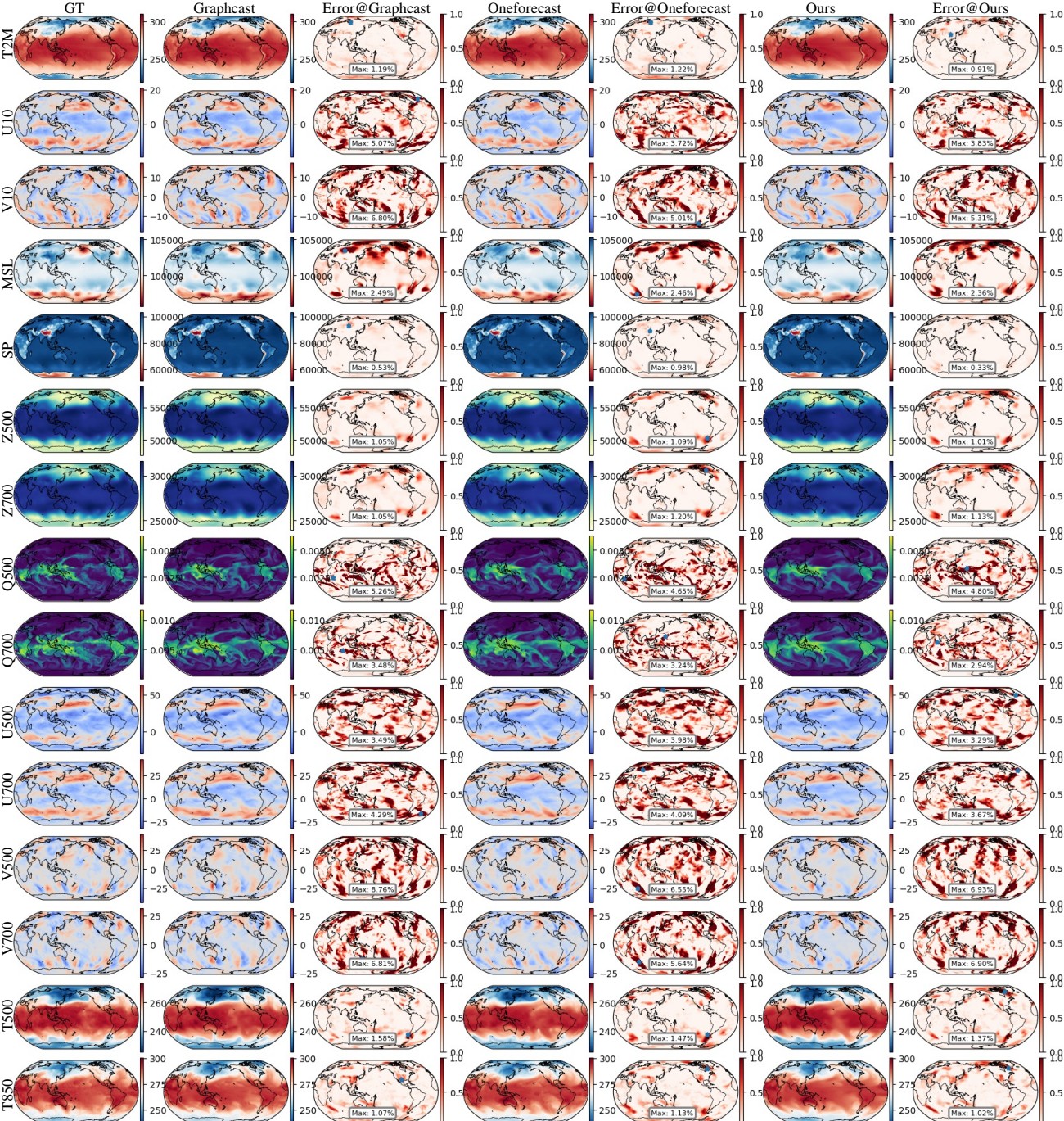

*Figure 19.* 10-day forecast results of global weather among different models.

