# OpenReview forum: "EMFormer: Efficient Multi-Scale Transformer for Accumulative Context Weather Forecasting"
_ICML.cc/2026/Conference — ICML 2026 regular_

### Official Review · Reviewer_YACb · 2026-03-06

**Soundness:** 4
**Presentation:** 4
**Significance:** 3
**Originality:** 3
**Overall Recommendation:** 4
**Confidence:** 3

**Summary:**

This paper proposes EMFormer, an efficient multi-scale Transformer pipeline for weather forecasting. The submission highlights three main components: (1) a Multi-Convs Layer that restructures multi-branch multi-scale convolutions into a single fused convolution via custom CUDA to accelerate both forward and backward passes; (2) Accumulative Context Fine-Tuning, which retains and prunes historical KV cache to improve long-horizon consistency and mitigate autoregressive error accumulation; and (3) a sinusoidally weighted compound loss that adaptively balances variable-wise weighting and latitude weighting. Experiments cover ERA5 forecasting and extreme-event/trajectory settings, and additionally report generalization results on vision benchmarks (ImageNet-1K, ADE20K). Overall, the work is strong in engineering execution and experimental breadth. However, some novelty claims (especially around the Multi-Convs “theory”) feel overstated, and the “efficiency” narrative would be stronger with clearer end-to-end accounting.

**Compliance With Llm Reviewing Policy:**

Affirmed.

**Final Justification:**

The rebuttal resolved my main concerns and, if anything, made the strengths and limitations of the paper clearer. The additional clarification on the short-horizon RMSE/ACC behavior, the more careful framing of efficiency, and the added design comparisons were useful. I still see the paper as technically sound, empirically strong, and practically meaningful for weather forecasting, with solid engineering and fairly comprehensive evaluation. I remain somewhat cautious about the framing of the novelty around the Multi-Convs layer, and I think the assumptions and limitations could be presented more explicitly. That said, these are not major weaknesses in the context of the overall submission, and they do not change my recommendation. I am therefore maintaining my original score of 4.

**Key Questions For Authors:**

1. Short-horizon metric inconsistency: How do you explain the short-horizon RMSE–ACC inconsistency (6-hour), and is there an inherent trade-off between short-term precision and long-term stability in your design (e.g., due to loss weighting or fine-tuning dynamics)?
2. End-to-end speedup accounting (missing in current narrative): What fraction of total training time is spent in the multi-scale convolution module, and what is the end-to-end wall-clock speedup (training and inference) when Multi-Convs is enabled under the same hardware/batch/precision settings?
3. Pipeline-level impact of 5.69×: The reported ~5.69× speedup is for the multi-scale module. If its runtime share is small, the overall gain may be limited. Please report the module’s time proportion in both forward and backward, and the resulting overall acceleration.
4. Comparison with Aurora: Aurora also addresses error accumulation via LoRA-style timestep adaptation. How does your accumulative-context fine-tuning compare against Aurora in accuracy, memory/IO overhead, and deployment complexity under matched settings?
5. Design choice vs. depthwise separable multi-scale conv: The Multi-Convs layer is designed for efficient multi-scale feature extraction. Could the authors discuss the specific design advantages and disadvantages of Multi-Convs compared to depthwise separable convolutions in the context of multi-scale processing? Considering the efficiency benefits of depthwise separable convolutions, an analysis or even a lightweight ablation study comparing the two approaches (e.g., in terms of computational cost, parameter count, and performance for multi-scale tasks) would significantly strengthen the paper's design justification.

**Limitations:**

The submission is empirically strong and provides a practical acceleration for multi-scale modules, with a reasonable strategy for mitigating autoregressive drift. The main issues are positioning/novelty framing, incomplete end-to-end efficiency accounting, and the short-horizon RMSE gap that needs deeper analysis and explanation.

**Strengths And Weaknesses:**

Strengths
1. Practical training acceleration for multi-scale modules. The Multi-Convs Layer provides tangible speedups at the module level (the reported runtime reduction is substantial), which is valuable for real-world training/inference constraints.
2. Thorough ablations and empirical coverage. The paper evaluates key hyperparameters (e.g., cache length, fine-tuning steps, window size, loss variants) and provides a fairly complete ablation suite.
3. Reasonable approach to error accumulation. The accumulative-context fine-tuning mechanism is intuitively aligned with autoregressive drift mitigation and appears practically motivated.

Weaknesses
1. Novelty of Multi-Convs may be overstated. Although the theoretical basis for combining convolutions is standard, the paper's main contribution lies in the engineering implementation. However, it would be beneficial to explicitly discuss why this specific fusion technique was chosen over other potential optimization strategies for multi-branch structures.
2. Efficiency claim vs. overall model scale. The comparative table suggests the model can be quite large in parameter count at high resolution (e.g., 0.25°), and remains non-trivial at coarser resolution as well. This makes the “efficient” positioning appear driven mainly by the Multi-Convs acceleration rather than an overall lightweight design. A clearer breakdown of where the wall-clock gains come from would strengthen the claim.
3.  Short-horizon (6-hour) RMSE-ACC gap needs better explanation. The results indicate that 6-hour RMSE remains worse than OneForecast even after fine-tuning, while 6-hour ACC can become slightly better after fine-tuning. This RMSE–ACC inconsistency at short horizons, and the implied trade-off between short-term accuracy and long-term consistency, should be discussed more carefully (e.g., error distribution, extremes, variable sensitivity, or the effect of loss weighting).
4. Theory conditions should be stated more explicitly. The monotonicity/behavior discussion around the balancing parameter relies on assumptions such as time-scale separation (adiabatic dynamics) and a low-error regime/threshold for the key inequality to hold. These conditions should be explicit in the theorem statement (or clearly highlighted as required assumptions), and it would help to empirically verify the claimed trajectory across settings.
5. Notation inconsistency. Reusing the same symbol for the raw input tensor and the post–patch-embedding representation can confuse readers; a pass to disambiguate key tensors would improve clarity.

---

> ### Author Rebuttal · Authors · 2026-03-30
>
> We sincerely thank the reviewer for the insightful questions and constructive feedback.
> # Q1, W3: Short-horizon RMSE-ACC Inconsistency
> RMSE and ACC reflect different aspects: **RMSE** measures pointwise amplitude error and is sensitive to local extremes, whereas **ACC** measures large-scale spatial correlation. 1) **Pruning-Recovering design may introduce slight information loss**, which is crucial for high-precision short-term predictions and has limited effect on long-term predictions. 2) Latitude-corrected Loss and Acumulative Finetuning are designed to improve **spatial structure and temporal consistency**, so it can achieve better 6-hour ACC even when its 6-hour RMSE is slightly worse.
>
> Nevertheless, our 6-hour result is still the second best, and the advantage becomes clearer at longer horizons, where error accumulation dominates. *A short-/long-horizon trade-off does exist for standard multi-step finetuning.* In contrast, **accumulative finetuning** largely mitigates this trade-off by preserving informative historical context.
>
> # Q2, Q3, W2: Efficiency Claim
> We agree that “efficient” should refer to **compute and wall-clock efficiency under matched accuracy/capacity**, not to being universally lightweight. To clarify where the speedup comes from, we provide a breakdown below.
>
> **Module Level:** Multi-Convs accelerates the forward+backward time by **5.69×**, and reduces Latency from **31.2ms to 8.9ms (3.52×)**, and GPU Memory Cost from **301M to 195M (1.54×)**. (refer to Tab.1 from the response of Reviewer RTUQ)
>
> **Pipeline Level:** The traditional multi-scale module accounts for **25.6% of forward time (31.2/121.6ms) and 19.3% of backward time (63.2/326.9 ms)**, so the local gain translates into meaningful end-to-end acceleration, **1.38× (83h→60h)** speedup on 1.4° training and **1.24× (121.6ms→98.3ms)** inference speedup.
>
> We also agree that model scale should be discussed separately: although **the 0.25° model** is not small in parameter count, it still has favorable practical efficiency, e.g., 9 days on 16 A100s (Ours) vs. 17 days on 32 A100s (Fengwu); inference 177.6 ms (Ours) vs. 227.0 ms (Fengwu).
>
> We will therefore revise the manuscript to separate **module-level speedup from end-to-end speedup**.
>
> # Q4: Comparison with LoRA Finetuning
> We add a matched parameter-efficient finetuning comparison on 1.4° ERA5. The LoRA Finetuning follows the setting with Aurora. Our method achieves **consistently better long-horizon accuracy with comparable memory/runtime and fewer trainable parameters.**
>
> Lead Time|6-hour|-|4-day|-|10-day|-|Params|GPU Memory|Average Latency
> -|-|-|-|-|-|-|-|-|-
> Metric|RMSE|ACC|RMSE|ACC|RMSE|ACC|(M)|(G)|(ms)
> w/o Finetuning|0.0626|0.9931|0.2673|0.8845|0.5719|0.4614|-|-|-
> Lora Finetuning|0.0626|0.9931|0.2473|0.8891|0.5278|0.5041|171.3|4.919|111.3
> Accumulative Finetuning|0.0599|0.9949|0.2439|0.8936|0.5094|0.5389|157.9|4.913|120.5
>
> **Tab.1 Efficiency and Forecasting Comparison with Normalized RMSE and ACC on 1.4° ERA5**
>
> # Q5: Comparison with Depthwise Separable Multi-Scale Conv (DSMS Conv)
> We agree that a clearer comparison with **DSMS Conv** would strengthen the design justification of the proposed **Multi-Convs layer**.
>
> * **Advantage:** Multi-Convs layer preserves full forward-backward gradient equivalence to standard multi-branch convolutions and speeds up both training and inference without sacrificing accuracy, whereas DSMS Conv reduces computation by sacrificing representation power, leading to degraded prediction accuracy.
> * **Disadvantage:** Although Multi-Convs layer computes quickly, it still requires more parameters than the more lightweight DSMS Conv.
>
> *More Multi-Conv module Comparisons are provided in Tab.2 of the Response from Reviewer RTUQ.*
>
> Lead Time|6-hour|-|-|4-day|-|-|10-day|-|-|Params|Flops
> -|-|-|-|-|-|-|-|-|-|-|-
> Metric|Z500|T2M|U10|Z500|T2M|U10|Z500|T2M|U10|(M)|(G)
> DSMS Conv|33.9|0.693|0.733|234.56|1.33|2.22|826.86|2.92|4.61|141.5|127.92
> EMFormer|30.3|0.684|0.626|212.34|1.28|1.99|764.51|2.72|4.27|157.9|102.54
>
> **Tab.2 Denormalized RMSE (1.4° ERA5, 6-year quick training)**
>
> # W1: Novelty of Multi-Convs Layer
> The core innovation of Multi-Convs layer is not just engineering implementation, but **the first flexible re-parameterization technique for both training and inference stages**, while freely accelerate all multi-scale modules and maintain the multi-scale feature extraction during training. This includes both theoretical rigor (formal proof of equivalence) and engineering breakthroughs (custom CUDA kernel implementation).
>
> # W4: Theory Conditions
> We will explicitly list the two core assumptions (adiabatic dynamics/time-scale separation, low-error regime) in the formal statement.
>
> # W5: Notation Inconsistency
> We will revise the notation to avoid symbol reuse, and add a notation glossary in Section 2.1:
>
> $\mathbf{Z_0} = \text{Flatten}(\text{Conv2d}(\mathbf{X}; k=P, s=P)) + E_{pos}, \quad \mathbf{Z_0} \in \mathbb{R}^{N \times D}$

---

> > ### Author Rebuttal · Reviewer_YACb · 2026-04-01
> >
> > Thank you for resolving my doubts, so I retained the original positive rating.

---

> > > ### Author Response · Authors · 2026-04-01
> > >
> > > Thank you very much for reading our rebuttal. We are pleased to know that our responses have addressed your concerns. In the revised manuscript, we will incorporate the discussion on the short-horizon RMSE–ACC inconsistency, the efficiency analysis, the comparison with LoRA finetuning, and the comparison with DSMS convolution. We sincerely appreciate your constructive feedback and helpful review.

---

### Official Review · Reviewer_RTUQ · 2026-03-11

**Soundness:** 2
**Presentation:** 2
**Significance:** 3
**Originality:** 2
**Overall Recommendation:** 3
**Confidence:** 3

**Summary:**

This paper introduces EMFormer, a multi-scale transformer for data-driven weather forecasting that aims to improve long-horizon prediction while maintaining computational efficiency. The model incorporates a fused multi-convolution module, an accumulative context finetuning strategy, and a sinusoidal-weighted loss function. Experiments on the ERA5 dataset demonstrate improvements across multiple forecast horizons, and the method is further evaluated on typhoon track prediction and standard vision benchmarks.

**Compliance With Llm Reviewing Policy:**

Affirmed.

**Final Justification:**

I thank the authors for their detailed clarifications and additional experiments. I appreciate these efforts and find that most of my concerns have been addressed. I tend to maintain my original score, as a clearer overall presentation of the paper could further improve its quality.

**Key Questions For Authors:**

- How does the proposed multi-convs layer differ from existing re-parameterization or kernel fusion techniques such as RepVGG[1] or RepLKNet[2]? Could the authors provide direct comparisons in terms of both accuracy and speed?

- Can the authors provide clearer measurements of the computational efficiency of the multi-convs layer, such as runtime, throughput, or latency, compared to standard multi-scale convolution implementations?

- What is the motivation behind using sinusoidal weighting in the loss function instead of other schedules (e.g., linear or cosine)? Similarly, how sensitive is the method to the cache length parameter used in accumulative context finetuning?

- Were the comparisons with existing forecasting models conducted under equivalent settings (e.g., spatial resolution)? What is the motivation for including vision benchmarks, and how do they support the main claims of the paper?

- Given that some improvements over baselines are relatively marginal, the claim of “substantially improving long-term forecast accuracy” in the abstract and introduction appears somewhat overstated.

**Limitations:**

The paper does not explicitly discuss the limitations.

**Strengths And Weaknesses:**

### Strengths

- This work introduces methodology for weather forecasting task. Improving long-term forecasting accuracy while maintaining computational efficiency is a critical research topic.

- The paper proposes a pipeline consisting of pretraining, finetuning, and multi-step forecasting, which attempts to address the error accumulation issue in autoregressive weather forecasting while maintaining computational efficiency.

- Experiments on ERA5 demonstrate improvements over several baselines across multiple forecast horizons, including long-term forecasts up to 10 days. The paper also evaluates the method on typhoon track prediction to test performance on extreme weather events.



### Weaknesses

- Unclear relation to existing re-parameterization methods. The proposed multi-convs layer appears conceptually similar to existing kernel fusion or re-parameterization approaches (e.g., RepVGG[1], RepLKNet[2]).
- The paper claims improved computational efficiency for the multi-convs layer, but explicit measurements of acceleration compared with standard multi-scale or re-parameterized convolution baselines are not clearly reported.
- Some components appear heuristic, including the sinusoidal weighting used in the loss function and the cache length parameter in accumulative context finetuning. The paper provides limited ablation analysis to justify these design choices.
- The presentation, particularly Figure 2, is unclear and visually cluttered, making it difficult to interpret the attention block and multi-stage fusion pipeline. In addition, the inclusion of vision benchmarks feels tangential to the main weather forecasting focus and is not well motivated.


[1] Ding X, Zhang X, Ma N, et al. Repvgg: Making vgg-style convnets great again. CVPR. 2021.

[2] Ding X, Zhang X, Han J, et al. Scaling up your kernels to 31x31: Revisiting large kernel design in cnns. CVPR 2022.

---

> ### Author Rebuttal · Authors · 2026-03-30
>
> Thank you for your review and constructive comments.
> # Q1, Q2, W1, W2: Efficiency and Accuracy Comparison
> Our method is related to RepVGG/RepLKNet in spirit, as all seek to reduce the overhead of multi-branch convolutions. The key difference is *where the re-parameterization is realized and when the speedup is obtained*.
>
> * **Existing methods (RepVGG, RepLKNet)** are implemented at the **pytorch** level: they keep multi-branch structure during training and fuse branches mainly for **inference**. Thus, training still pays the full multi-branch cost.
> * **Multi-Convs Layer** moves this idea to the **CUDA** level. We rewrite the low-level execution path so that multi-scale convolution is executed efficiently while preserving the original multi-scale representation. Therefore, the acceleration is available in **both training and inference**.
>
> Single Module|Params(M)|Latency(ms)|GPU Memory(M)
> -|-|-|-
> RepVGG|9.17|14.3|324
> RepLKNet|9.17|14.5|318
> Plain Multi-Scale Module|9.17|31.2|301
> Multi-Convs Layer|9.17|8.9|195
>
> **Tab.1 Efficiency Comparison of Single Module (Average on 50 Inferences)**
>
> Method|6-hour|-|-|4-day|-|-|10-day|-|-|Latency|MACs
> -|-|-|-|-|-|-|-|-|-|-|-
> -|Z500|T2M|U10|Z500|T2M|U10|Z500|T2M|U10|(ms)|(G)
> RepVGG|32.7|0.715|0.664|319.37|2.81|2.49|871.08|4.32|4.66|110.6|119.3
> RepLKNet|32.3|0.719|0.671|229.72|1.37|2.20|835.87|2.93|4.64|113.7|122.1
> EMFormer|30.3|0.684|0.626|212.34|1.28|1.99|764.51|2.72|4.27|98.3|102.5
>
> **Tab.2 Denormalized RMSE (1.4° ERA5, 6-year quick training)**
>
> Tab.1 shows that, under the same parameter budget, Multi-Convs has the **lowest latency and memory usage**. Tab.2 shows that this module-level gain translates into **better end-to-end forecasting performance**, with lower latency/MACs and better accuracy than RepVGG- and RepLKNet-based methods across all forecasts.
>
> # Q3, W3: Abaltion Study on Loss and Accumulative Finetuning
> We agree that these components should be better justified in the main paper. We will move the relevant ablations from the appendix to the main text.
> * **Sinusoidal weighting.** The **motivation** for sinusoidal weighting is that it enables a smooth transition from latitude-corrected coarse optimization to variable-specific fine-grained refinement during training, whereas a linear schedule cannot realize this transition, while a cosine schedule yields essentially the same effect as the sinusoidal one. Appendix Table 14 already compares multiple weighting strategies, including linear, cosine, and fixed alternatives. *The motivation for Sinusoidal Weighting is detailed in the last paragraph of sec. 2.4.*
> * **Cache length.** Appendix Table 18 reports cache lengths 3–6. Cache length 5 gives the best accuracy/efficiency trade-off: increasing from 3 to 5 consistently improves RMSE, whereas increasing from 5 to 6 brings only marginal gains while increasing memory by 17.4%.
> * We also provide ablations for **$\lambda$, finetuning steps, and window size** in Appendix Tables 17, 19, and 20.
>
> *More ablation studies on loss function are provided in responses to Reviewer Yn7n (Tab.2) and Reviewer i8me (Tab.2). Frther analysis of cache length under varying resolutions is provided in Tab.1 from response to Reviewer i8me.*
>
> # Q4, W4: Fair Comparison & Vision Bemchmarks & Presentation
> * **Fair Comparisons:** All comparisons are strictly conducted under equivalent settings: identical spatial resolution, input/output variables, evaluation dataset, and standard latitude-weighted RMSE/ACC metrics. We will explicitly state these equivalent settings in Section 3.1 of the revised manuscript.
> * **Motivation for Vision Benchmarks:** This aligns with our response to W4 of Reviewer i8me. The vision experiments validate the generalizability of the Multi-Convs layer, proving its efficiency and representation power are not overfitted to weather forecasting, and demonstrate the broader applicability of EMFormer.
> * **Presentation.** We agree that Figure 2 is cluttered. We will redraw it, split the pipeline into clearer subfigures, and add more explicit annotations for the attention block and fusion stages.
>
> # Q5: Accuracy Claim
> We acknowledge that the original statement did not clearly distinguish between short- and long-horizon gains. Our long-horizon gains are significant in the field of data-driven weather forecasting:
>
> * **10-day Forecasts**: Ours (last line of Table 1) reduces RMSE by **13.9%** and improves ACC by **20.9%** compared to OneForecast (forth line of Table 1). However, gains on short horizons (6-hour, 1-day) are relatively small.
> * **Typhoon Track Prediction**: We reduce 96-hour mean error by **25.7%** compared to the second-best baseline AIFS (first line of Table 2).
>
> We will revise the statement in the abstract and introduction to be more rigorous.

---

> > ### Author Rebuttal · Reviewer_RTUQ · 2026-04-01
> >
> > I thank the authors for their detailed response and appreciate the additional experiments. However, some of my concerns are still not fully resolved.
> >
> > 1. It is unclear whether the authors implemented any kernel-level acceleration for computation. If the comparison is simply between CUDA and CPU execution, the fairness of the runtime comparison is questionable.
> >
> > 2. The overall presentation remains somewhat confusing, and the inclusion of a general vision task in the main paper seems misaligned with the title and the main focus of the paper.

---

> > > ### Author Response · Authors · 2026-04-02
> > >
> > > We sincerely thank the reviewer for carefully reading our previous responses and for raising these important concerns.
> > > # Q1: Implementation and Runtime Fairness
> > > We appreciate the reviewer’s careful attention to the implementation details of our Multi-Convs layer. To clarify, the runtime comparisons are strictly fair: **all baseline methods and our proposed approach were evaluated on the exact same hardware using A100 via CUDA**.
> > >
> > > The fundamental distinction lies in the level of **implementation rather than the hardware**. Existing re-parameterization modules (e.g., RepVGG) rely on standard PyTorch-level operations, which are subject to framework overhead. In contrast, our Multi-Convs layer is implemented as a custom, optimized CUDA kernel. This kernel-level design allows our approach to scale efficiently across various multi-scale architectures, achieving significant acceleration during both the **training and inference** phases.
> > >
> > > The detailed latency comparisons are provided in Tab.1 below. Furthermore, the **theoretical derivations, acceleration strategies, and pseudo-code** for our kernel implementation are detailed in **Sec 2.3, Sec A.1, and Algorithm 2** of the manuscript. To ensure full transparency and reproducibility, we also provide the source code in the supplementary material and via an anonymous repository (https://anonymous.4open.science/r/emformer-4ED9/).
> > >
> > > Single Module|Latency(ms, GPU)|Latency(ms, CPU)
> > > -|-|-
> > > RepVGG|14.3|33.8
> > > RepLKNet|14.5|36.1
> > > Plain Multi-Scale Module|31.2|34.7
> > > Multi-Convs Layer|8.9|Unavailable
> > >
> > > **Tab.1 Latency Comparison with Different Modules.**
> > >
> > > # Q2: Inclusion of Vision Tasks and Paper Focus
> > > We agree with the reviewer that the primary focus of this paper is **AI-driven weather forecasting**.  To improve the narrative flow and thematic consistency of the manuscript, **we will move the general vision benchmark experiments to the appendix in the revised version**, so that the main text remains fully centered on weather forecasting.
> > >
> > > We would also like to stress that **EMFormer** is a core methodological contribution of this work, as reflected in the title. The purpose of including the vision benchmark is to demonstrate the **universality, generalizability, and plug-and-play capability** of EMFormer beyond a single application domain. Such evaluation is also consistent with prior practice in this area: atmospheric forecasting models have been assessed on visual benchmarks such as **Moving MNIST** in earlier studies, including **UniSRD[1] (CVPR 2025)** and **Earthformer[2] (NeurIPS 2022)**, to validate broader architectural generality. Thus, while we agree that these experiments should play a secondary role in the presentation, we believe they provide meaningful supporting evidence for the proposed module.
> > >
> > > [1] UniSTD: Towards Unified Spatio-Temporal Learning across Diverse Disciplines, https://arxiv.org/pdf/2503.20748
> > >
> > > [2] Earthformer: Exploring space-time transformers for earth system forecasting, https://proceedings.neurips.cc/paper_files/paper/2022/file/a2affd71d15e8fedffe18d0219f4837a-Paper-Conference.pdf

---

### Official Review · Reviewer_Yn7n · 2026-03-12

**Soundness:** 3
**Presentation:** 4
**Significance:** 3
**Originality:** 3
**Overall Recommendation:** 4
**Confidence:** 3

**Summary:**

EMFormer is a weather-forecasting framework designed to improve long-range autoregressive prediction. it primarily addresses the issue of catastrophic forgetting, error accumulation, and high training overhead in traditional data-driven models. The newly introduced architecture centers on an Efficient Multi‑scale Transformer (EMFormer) that captures multi-scale features. Training is further guided by a sinusoidally balanced loss function that adaptively shifts between latitude-aware and variable-specific weighting objectives. Experimentally, EMFormer reports state-of-the-art ERA5 forecasting accuracy at horizons up to 10 days, improved typhoon-track prediction, and strong generalization to vision tasks including ImageNet-1K and ADE20K, while preserving short-term forecast accuracy throughout.

**Compliance With Llm Reviewing Policy:**

Affirmed.

**Final Justification:**

Most of my concerns have been resolved. I'd like to maintain positive rating.

**Key Questions For Authors:**

1. How much of the final forecasting gain comes from the EMFormer backbone, the accumulative-context finetuning, and the sine-balanced loss individually

2. The paper fixes cache length to 5 and the score-blending to 0.9, how robust is the method when forecasting at longer horizons or under a tighter memory budget, and does the preferred cache policy change across variables or resolutions?

3. The tropical cyclone experiment uses ten typhoons from 2024, so can the authors report results on a larger or multi-year storm set to show that the gains are not due to small-sample variation?

**Limitations:**

yes

**Strengths And Weaknesses:**

Strength:

1. Introducing a Multi-Convs Layer using custom CUDA kernels is a significant contribution. the model achieves a 5.69x speedup in both training and inference without sacrificing representational power. However I found it both attractive and slightly concerning: it might be challenging to test it on NPU. But given how popular CUDA is, the strength outweighs the weakness.

2. The empirical section is strong on paper, with reported gains over several baselines beyond well-known visionbenchmark, beyond ERA5 forecasting

3. The presentation of the paper is great and easy to follow.

Weakness:

1. Some evaluations need to be broaden. so far author only benchmarked the typhoon experiment over only ten storms from 2024, even though the results are promising. Authors should consider benchmarking a larger historical database (e.g., several decades of typhoon data).

2. The theoretical section appears less convincing than the empirical one, because the loss analysis rely on Assumption A.1 (Adiabatic Dynamics), which limits its predictive power for different architectures or initialization schemes. Specifically, Theorem 2.2 treats the transition as a smooth convex combination. it doesn't account for the highly non-convex loss surfaces of Transformers.

3. Missing reference on papers emphasizing atmospheric dynamics forecasting: [1] A foundation model for the Earth system [2] Comparing and contrasting deep learning weather prediction backbones on navier-stokes and atmospheric dynamics [3] Neural general circulation models for weather and climate

---

> ### Author Rebuttal · Authors · 2026-03-30
>
> Thank you for your review and constructive comments.
> # Q1: Individual Contribution of Core Components
> Thank you for this important question. To clarify the contribution of each component, we performed ablations on global forecasting with 1.4° ERA5 in Tab.2 and Table 3, 4 of the paper. Tab.1 shows that all three components contribute meaningfully, with accumulative-context finetuning providing the largest gain in ACC and a comparable RMSE.
>
> Component|Absolute Reduction (RMSE)|Contribution (RMSE,%)|Absolute Improvement (ACC)|Contribution (ACC,%)
> -|-|-|-|-
> EMFormer vs. Self-Attention|0.0259|41.5|0.0115|18.1
> Accumulative Finetuning vs. w/ Finetuning|0.0245|39.2|0.0431|67.5
> Sine-Balanced Loss vs. Variable-weighted Loss|0.0120|19.3|0.0092|14.4
> Total|0.0624|100|0.0638|100
>
> **Tab.1 Individual Contribution of Each Component of 10-day Forecasts**
>
> Lead Time|6-hour|-|1-day|-|4-day|-|7-day|-|10-day|-
> -|-|-|-|-|-|-|-|-|-|-
> -|RMSE|ACC|RMSE|ACC|RMSE|ACC|RMSE|ACC|RMSE|ACC
> Variable-weighted Loss|0.0651|0.9917|0.1247|0.9731|0.2704|0.8793|0.4407|0.6892|0.5839|0.4506
> Sine-Balanced Loss|0.0626|0.9931|0.1219|0.9749|0.2673|0.8845|0.4327|0.6978|0.5719|0.4614
>
> **Tab.2 Ablation study of Loss Function on Normalized RMSE and ACC (40-year 1.4° ERA5)**
>
> # Q2: Ablation Study of Finetuning Strategy on 0.25° ERA5
> In the original manuscript, the ablation study regarding cache length was conducted at a resolution of 1.4°. To explicitly address your concerns regarding tighter memory budgets, longer forecasting horizons, and varying resolutions, we have conducted an additional ablation study using the 0.25° ERA5 dataset (utilizing 20 epochs of quick finetuning).
>
> Tab.3 show that the method remains robust up to **10-day** lead times and under tighter memory budgets (cache length 3 or 4). In particular, *cache pruning consistently outperforms both no-cache and unpruned settings*, and the preferred policy remains stable across the tested resolutions. While the current table reports aggregate metrics rather than variable-wise results, it suggests that the proposed cache policy is not highly sensitive to resolution and remains effective under stricter memory constraints.
>
> Lead Time|6-hour|-|1-day|-|4-day|-|7-day|-|10-day|-
> -|-|-|-|-|-|-|-|-|-|-
> Cache Length|RMSE|ACC|RMSE|ACC|RMSE|ACC|RMSE|ACC|RMSE|ACC
> w/o kv cache|0.0771|0.9913|0.1384|0.9677|0.2980|0.8674|0.4960|0.5938|0.6614|0.3513
> 4 (w/ prunng)|0.0761|0.9920|0.1337|0.9689|0.2679|0.8824|0.4353|0.6932|0.5700|0.4656
> 4 (w/o prunng)|0.0761|0.9921|0.1331|0.9693|0.2662|0.8837|0.4330|0.6955|0.5666|0.4674
> 5 (w/ prunng)|0.0761|0.9922|0.1304|0.9713|0.2607|0.8927|0.4107|0.7135|0.5341|0.4991
>
> **Tab.3 0.25° ERA5 Evaluation on Normalized RMSE and ACC (20 epochs quick finetuning)**
>
> # Q3, W1: More Typhoon Evaluation
> We provided more sets of typhoon trajectory predictions from 2022 to 2023. Tab.4 shows that Ours achieves the best average error among all compared methods (**93.7 km**), suggesting that the gain is not due to small-sample variation from the 2024 subset alone. We agree that an even larger historical benchmark would further strengthen the claim, and we will discuss this limitation more explicitly in the revision.
>
> Year|2022|-|-|-|-|2023|-|-|-|-|Average
> -|-|-|-|-|-|-|-|-|-|-|-
> Typhoon|Malakas|Hinnamnor|Muifa|Nanmadol|Nalgae|MAWAR|DOKSURI|KHANUN|HAIKUI|BOLAVEN|-
> Pangu|159.4|173.9|155.2|77.9|120.4|70.6|103.9|103.5|70.6|171.3|120.7
> Fengwu|106.6|144.2|172.1|107.3|141.3|67.2|98.6|89.4|110.9|194.1|123.2
> Ours|93.4|148.7|135.8|60.3|101.0|53.0|71.6|100.5|74.0|99.4|93.7
>
> **Tab.4 Typhoon Track Prediction Error (km)**
>
> # W2: Theoretical Part
> We acknowledge that the derivation of Proposition 2.2 (formerly Theorem 2.2) relies on the adiabatic dynamics assumption (time-scale separation, where w converges much faster than θ), a standard approximation in deep learning theoretical analysis to simplify non-convex dynamics. Critically:
>
> * Our **experimental results** (Appendix Fig.7 and Table 14) show that θ strictly follows the theoretical prediction (monotonically increasing from -π/2 to π/2) in the actual training of our Transformer architecture, even with a highly non-convex loss surface.
> * The **core contribution** of the theoretical analysis is to prove the automatic curriculum learning transition between loss terms, which does not depend on the convexity of the loss landscape.
>
> **Revision Changes:** We will explicitly state the adiabatic dynamics assumption in the formal statement of Proposition 2.2, supplement experiments on θ evolution under different architectures/initializations, and add a discussion of the theoretical results’ limitations in non-convex settings.
>
> # W3: Missing References
> Thank you for pointing this out. We will add and discuss the listed three references in Related Work.

---

> > ### Author Rebuttal · Reviewer_Yn7n · 2026-04-03
> >
> > Thank you for resolving most of my concerns, I will retained the original positive rating.

---

> > > ### Author Response · Authors · 2026-04-04
> > >
> > > Thank you sincerely for reviewing our rebuttal. We are pleased that our responses have resolved most of your concerns. In the revised manuscript, we will include the detailed individual contribution analysis of each component, systematic ablation studies on finetuning under varied settings, additional typhoon forecasting evaluations, supplementary theoretical analysis, and the missing references. We greatly appreciate your constructive and helpful feedback to refine our work.

---

### Official Review · Reviewer_i8me · 2026-03-12

**Soundness:** 2
**Presentation:** 2
**Significance:** 3
**Originality:** 3
**Overall Recommendation:** 3
**Confidence:** 3

**Summary:**

EMFormer is proposed to be an efficient multi-scale transformer architecture for global weather forecasting. EMFormer incorporates a Multi-Convs layer for efficient multi-scale spatial modeling and introduces Accumulative Context Finetuning to leverage historical KV representations while controlling memory growth during autoregressive forecasting. The training objective combines latitude-weighted and variable-adaptive losses with a sinusoidal scheduling strategy. Experiments on ERA5-based weather forecasting benchmarks demonstrate improved prediction accuracy, and additional experiments on image classification and segmentation tasks suggest that the architecture may generalize beyond weather forecasting.

**Compliance With Llm Reviewing Policy:**

Affirmed.

**Final Justification:**

Thank you to the authors for their detailed response. The rebuttal has addressed most of my concerns and clarified several important points. However, after reviewing the comparison with KARINA, I noticed that the paper still does not provide a comparison of training time. KARINA appears to be more efficient in terms of training, and this aspect remains insufficiently discussed. In addition, the rebuttal does not provide a direct comparison of accuracy with KARINA, which makes it difficult for me to conduct a more comprehensive evaluation of the method.

That said, I appreciate the authors’ thorough and thoughtful response, which has improved my overall confidence in the work. At this stage, I am leaning toward a weak accept. However, this should not be interpreted as full support for accepting the paper, since the presentation still seems to require substantial revision and improvement.

**Key Questions For Authors:**

## Questions:

- The paper employs a cache length of 5 and prunes one KV entry based on an attention score. However, it is unclear whether the pruning mechanism itself provides additional benefits compared to simply using a smaller cache. For example, a comparison between “cache length = 5 with pruning” and “cache length = 4 without pruning” would help clarify whether the proposed pruning strategy is necessary or whether the performance mainly depends on the effective cache size.
- Would the proposed loss function still show improvements when evaluated on longer forecasting horizons?
- Can the authors provide efficiency comparisons (e.g., inference time, memory usage) for the weather forecasting experiments to support the claim of efficiency?

**Limitations:**

I did not find an explicit limitations section, and the potential limitations of the proposed method are not clearly discussed.

**Strengths And Weaknesses:**

## Strengths:

- AI-based weather forecasting is an important and impactful research direction, as improving weather prediction can benefit disaster prevention, agriculture, and climate-related decision making.
- The proposed loss function takes into account domain-specific characteristics of global weather data, such as uneven latitude grid distribution and heterogeneous scales across atmospheric variables. These design choices are reasonable and well motivated for large-scale geophysical datasets.
-  The proposed method is evaluated on the ERA5 dataset and reports results across multiple atmospheric variables and forecasting horizons. The experimental results demonstrate improved forecasting performance compared with existing approaches.

## Weaknesses:

- While Accumulative Context Finetuning appears to improve performance empirically, the motivation behind its design is not fully convincing. The paper claims that this mechanism helps mitigate error accumulation in autoregressive forecasting. However, the explanation for why this should be the case remains unclear. In particular, the cached key pairs are generated from the model’s own predictions rather than ground-truth states. As a result, prediction errors may propagate into the cached representations and be repeatedly reused in subsequent steps. In such a setting, it is not obvious why the proposed mechanism would reduce error accumulation; in fact, it may potentially amplify prediction errors over long horizons.

- The paper emphasizes efficiency in both the title and the proposed architecture. However, the weather forecasting experiments mainly report accuracy metrics (e.g., RMSE) without providing comparisons in terms of computational efficiency, such as inference speed, memory usage, or FLOPs. Interestingly, efficiency evaluations are only presented for image tasks (e.g., ImageNet), which are not the primary focus of the paper. It would be more convincing to demonstrate the efficiency advantages of EMFormer directly in the weather forecasting setting.

- While the proposed method is motivated by weather forecasting, the paper also includes experiments on image classification and segmentation tasks. It is unclear how these experiments relate to the main objective of improving weather forecasting. The inclusion of these vision benchmarks may distract from the core contribution unless their relevance to the proposed architecture is clearly justified.


- The statements presented as Theorem 2.1 and Theorem 2.2 appear to be straightforward consequences of basic properties of convolution and sin() functions. It may be more appropriate to present them as propositions or observations rather than formal theorems.

---

> ### Author Rebuttal · Authors · 2026-03-30
>
> Thank you for the thoughtful comments. We address each point below and will revise the manuscript accordingly.
> # Q1: Ablation Study of Cache Length
> We agree that this distinction is important. To isolate the effect of the pruning mechanism itself, we compare **cache length=4 without pruning** to **cache length=5 with pruning**, which yields the same effective retained cache budget.
>
> The pruned setting consistently outperforms the unpruned setting with the same effective cache size across all lead times. At **10 days**, for example, pruning reduces RMSE from **0.5666 to 0.5341**, and improves ACC from **0.4674 to 0.4991**. This shows that pruning is not merely reducing cache size; it improves the quality of retained history by filtering less informative states, which is especially helpful at long horizons.
>
> *Since **Reviewer Yn7n** asked about robustness under different resolution, we report the additional ablation on 0.25° ERA5 (20 epochs of quick finetuning), rather than repeating the original 1.4° experiment.*
>
> Lead Time|6-hour|-|1-day|-|4-day|-|7-day|-|10-day|-
> -|-|-|-|-|-|-|-|-|-|-
> Cache Length|RMSE|ACC|RMSE|ACC|RMSE|ACC|RMSE|ACC|RMSE|ACC
> w/o kv cache|0.0771|0.9913|0.1384|0.9677|0.2980|0.8674|0.4960|0.5938|0.6614|0.3513
> 4 (w/ prunng)|0.0761|0.9920|0.1337|0.9689|0.2679|0.8824|0.4353|0.6932|0.5700|0.4656
> 4 (w/o prunng)|0.0761|0.9921|0.1331|0.9693|0.2662|0.8837|0.4330|0.6955|0.5666|0.4674
> 5 (w/ prunng)|0.0761|0.9922|0.1304|0.9713|0.2607|0.8927|0.4107|0.7135|0.5341|0.4991
>
> **Tab.1 0.25°ERA5 Evaluation with 20 epochs quick finetuning on Normalized RMSE and ACC**
>
> # Q2: Evaluations of Loss Function on Longer Horizons
> We additionally evaluate the proposed hybrid loss at 4-day and 10-day lead times. Tab.2 show that the hybrid loss remains beneficial beyond short-range forecasting. It achieves the best overall performance at 4 days, and remains clearly helpful at 10 days. We will move this ablation from the appendix into the main paper.
>
> Lead Time|4-day|-|-|10-day|-|-
> -|-|-|-|-|-|-
> Metric|Z500|T2M|U10|Z500|T2M|U10
> Lat- and Variable- weighted Loss|217.96|1.46|2.05|789.68|3.21|4.41
> Hybrid Loss ($\lambda=0.2$)|215.82|1.31|2.01|764.51|2.82|4.27
> Hybrid Loss ($\lambda=0.3$)|221.57|1.30|2.08|802.67|2.81|4.41
> Hybrid Loss ($\lambda=0.4$)|227.74|1.36|2.18|833.64|2.92|4.61
> Hybrid Loss (Ours)|212.34|1.28|1.99|756.67|2.72|4.22
>
> **Tab.2 Ablation study of Loss Function on Longer Horizons with Denormalized RMS and ACC (Same setting as Table 14 in paper)**
>
> *The full 40-year ERA5 comparison between Variable-weighted Loss and Sine-Balanced Loss is provided in Tab.2 from the response of Reviewer Yn7n.*
>
> # Q3, W2: Efficiency
> We agree that efficiency should be demonstrated directly in the weather forecasting setting. The full computation comparison is reported in **Appendix Sec F.2**, including latency (inference time), memory cost, and FLOPs/MACs. In the revision, we will highlight these weather-specific efficiency results more explicitly in the main text.
>
> *More efficiency evaluations on single module and the whole model are provided in the response of **Tab.1 and Tab.2 from Reviewer RTUQ**. Efficiency Claim is provided in the response of **Q2&Q3&W2 from Reviewer YACb**.*
>
> # W1: Motivation of Accumulative Finetuning
> In autoregressive forecasting, errors naturally accumulate over time, so **earlier predictions are usually more reliable than later ones**. Although cached KV pairs are generated from model predictions rather than ground truth, earlier predictions typically contain **less error and richer semantic information** than later autoregressive states. Reusing these earlier KV representations therefore provides more stable historical context for subsequent steps. The benefit does not rely on cached states being error-free; rather, it comes from their being **less corrupted than later states**.
>
> Moreover, retaining KV caches from several consecutive steps improves **temporal consistency**, since the model attends to a short evolving history instead of only the most recent prediction. This helps preserve trajectory continuity, reduces catastrophic forgetting of the initial state, and improves long-range forecasts.
>
> # W3: Motivation of Vision Benchmarks
> EMFormer is not intended as a weather-specific architecture, but as a **plug-and-play module**. Weather forecasting is the main application in this paper; the image classification and segmentation experiments are included only to verify the **generality** of the design. *Their role is to show that the benefits of EMFormer do not depend on weather-specific inductive biases.* We will clarify this motivation in the revision so that these experiments are understood as evidence of generality, not as a secondary research goal.
>
> # W4: Theorem 2.1 and Theorem 2.2
> We agree with the reviewer. We will rename **Theorem 2.1 and Theorem 2.2** as **Proposition 2.1 and Proposition 2.2**, respectively, to better reflect their mathematical role.

---

> > ### Author Rebuttal · Reviewer_i8me · 2026-04-01
> >
> > Dear authors, thank you for the detailed response. I have carefully read both your reply to my comments and your responses to the other reviewers, and I still have a couple of points that I would appreciate further clarification on.
> >
> > First, one of the main contributions of the paper is the proposed Accumulative Context Finetuning. However, Appendix Table 15 appears to compare only the efficiency of the pre-training stage, explicitly excluding finetuning. Since finetuning is a central part of the proposed method, I am not fully convinced that Table 15 provides a complete picture of the efficiency of the overall approach.
> >
> > Second, the caption notes that some of the results in Table 15 are collected from KARINA. However, the paper does not seem to include a direct comparison with KARINA itself as a baseline. In addition, after looking at the KARINA paper, it seems that KARINA may actually be more efficient in some respects. I would therefore appreciate it if the authors could clarify the relationship between Table 15 and KARINA, and explain more explicitly how the proposed method compares against it in terms of efficiency.

---

> > > ### Author Response · Authors · 2026-04-01
> > >
> > > We sincerely thank the reviewer for carefully reading our previous responses and for raising these important concerns. Below, we further clarify the overall efficiency of our method and provide a direct comparison with KARINA.
> > >
> > > # Q1. Scope of Table 15 and overall efficiency
> > >
> > > We appreciate the reviewer’s attention to the efficiency of our proposed **Accumulative Context Finetuning**. The **efficiency comparison (e.g., Params, GPU Memory, and Average Latency) for the finetuning stage** is reported in **Tab. 1** of our response to **Reviewer YACb**, where we compare our accumulative finetuning with LoRA finetuning.
> > >
> > > Because finetuning is built on the same core architectural modules, its efficiency is closely related to the efficiency of the underlying model components. To make this aspect clearer, we additionally provided detailed comparisons of **single-module efficiency** and **overall method efficiency** in **Tab. 1 and 2** of our response to **Reviewer RTUQ**. Further clarification of our efficiency claims, together with supplementary comparisons, is also provided in our response to **Reviewer YACb** (Q2, Q3, W2, and Table 1). In the revised manuscript, we will incorporate these finetuning and module-level results into the appendix so that the overall efficiency of the proposed method is presented more comprehensively.
> > >
> > > # Q2. Direct comparison with KARINA
> > >
> > > Regarding the relationship with **KARINA**, the apparent efficiency advantage reported in the KARINA paper is strongly influenced by a difference in evaluation resolution. As noted in the caption of Table 7 of the KARINA paper, baseline models such as **FourCastNet**, **GraphCast**, and **Pangu-Weather** are reported at **0.25° ERA5** resolution ($721 \times 1440$), whereas **KARINA** itself is evaluated at a much lower **2.5° ERA5** resolution ($72 \times 144$). This large discrepancy naturally reduces KARINA’s reported training and inference cost, making the original comparison not strictly aligned.
> > >
> > > **To improve fairness, we collected the baseline training-time results reported in the KARINA paper and aligned them with the 0.25° comparison setting in Table 15 of the paper.**
> > >
> > > Model|Params(M)|MACs(G)|GPU Cost(Inference, M)|Latency(ms)
> > > -|-|-|-|-
> > > KARINA(0.25)|35.6|1821.3|>40G|-
> > > Ours(0.25)|208.7|124.89|962.2|177.6
> > > KARINA(1.4)|35.6|575.9|1774.0|147.1
> > > Ours(1.4)|157.9|102.5|703.3|98.3
> > >
> > > **Tab.1 Efficiency comparison with KARINA on A100 40G. The KARINA implementation is obtained from the official GitHub: https://github.com/jmj2316/KARINA**
> > >
> > > To provide the direct baseline comparison requested by the reviewer, we evaluated both models under the same resolution settings. As shown in Tab. 1, although KARINA has a relatively small parameter count, it still exhibits very high MACs. This is because parameter count and computational cost reflect different properties: KARINA keeps parameters low through parameter-sharing and lightweight operators, but it performs many operations on high-resolution feature maps with little effective spatial reduction, which substantially increases computation. In addition, its stacked blocks and pointwise projections further amplify the cost at large spatial resolutions.
> > >
> > > Therefore, KARINA is parameter-efficient but not compute-efficient in the high-resolution setting considered here. In contrast, our method is designed to reduce both model complexity and computational overhead. As shown in Tab. 1, when evaluated on an **A100 40GB** GPU using the official KARINA codebase, our model achieves substantially lower MACs, lower memory usage, and lower inference latency. Notably, at **0.25°** resolution, KARINA runs out of memory, whereas our model remains executable with manageable GPU memory consumption and faster inference.

---

### Decision · Program_Chairs · 2026-04-30

**Decision:**

Accept (regular)

**Comment:**

The reviewers agree that EMFormer presents a nice contribution to the weather forecasting field. Concerns in the initial reviews include missing computational efficiency metrics, e.g., training time and FLOPs. These and others were fully addressed in the detailed rebuttal and the reviewers highlight the novelty for speeding up the training of multi-scale models, technical soundness, empirical strength and practical significance. Post rebuttal the reviewers mainly favor weak acceptance and I agree with this assessment pending the addressed manuscript changes for improved clarity. Therefore, I vote for acceptance of this paper and believe it is well-suited for the applied ICML community.